# TVNet: A novel time series analysis method based on dynamic convolution and 3D-Variation

**Chenghan Li**[*1], **Mingchen Li**[*2] **& Ruisheng Diao**[†3]

[1]Shenzhen International Graduate School(SIGS),Tsinghua University
[2]The Hong Kong University of Science and Technology (Guangzhou)
[3][†]ZJU-UIUC Institute, Zhejiang University
`lich24@mails.tsinghua.edu.cn; mli736@connect.hkust-gz.edu.cn;`
`ruishengdiao@intl.zju.edu.cn`

## Abstract

At present, the research on time series often focuses on the use of Transformer-based and MLP-based models.Conversely, the performance of Convolutional Neural Networks (CNNs) in time series analysis has fallen short of expectations, diminishing their potential for future applications. Our research aims to enhance the representational capacity of Convolutional Neural Networks (CNNs) in time series analysis by introducing novel perspectives and design innovations. To be specific, We introduce a novel time series reshaping technique that considers the inter-patch, intra-patch, and cross-variable dimensions. Consequently, we propose **TVNet**, **a dynamic convolutional network leveraging a 3D perspective to employ time series analysis**. TVNet retains the computational efficiency of CNNs and achieves state-of-the-art results in five key time series analysis tasks, offering a superior balance of efficiency and performance over the state-of-the-art Transformer-based and MLP-based models. Additionally, our findings suggest that TVNet exhibits enhanced transferability and robustness. Therefore, it provides a new perspective for applying CNN in advanced time series analysis tasks.

## 1 Introduction

Time series analysis plays a crucial role in many domains(Alfares & Nazeeruddin, 2002), such as anomaly detection(Pang et al., 2021) in industry signal and generation predication of renewable energy sources(Zhang et al., 2014). In recent years, remarkable progress has been made in this area(Wu et al., 2022a),(Luo & Wang, 2024). In particular, the MLP-based and Transformer-based architecture has been fully explored in the field of time series analysis (Wu et al., 2021),(Zhou et al., 2022),(Zhou et al., 2021),(Wu et al., 2022b),(Li et al., 2023a),(Challu et al., 2023). However, Transformers exhibit relatively low computational resource efficiency in time series tasks. MLP-based models often struggle to effectively capture the complex dependencies among variables and RNN-based models face challenges in modeling global temporal correlations. **In previous studies(Wang et al., 2024a), CNNs have been shown to be a neural network architecture that can balance efficiency and effectiveness. However, in the past, CNN architecture was often focused on video and images, and its application and research in time series predication and analysis are relatively limited.**

Time series data constitute records of continuous temporal changes, analogous to how video footage captures continuous visual transformations frame by frame - both capturing temporal continuity in their respective domains. Unlike certain sequential data types such as language(Lim & Zohren, 2021), time series data inherently reflect continuous phenomena. Much as a single video frame typically fails to convey the complete narrative of a scene, an isolated timestamp in a sequence usually lacks sufficient context for comprehensive analysis. This connection inspires us to leverage CNNs to

---

[*]Equal contribution.
[†]Corresponding author.

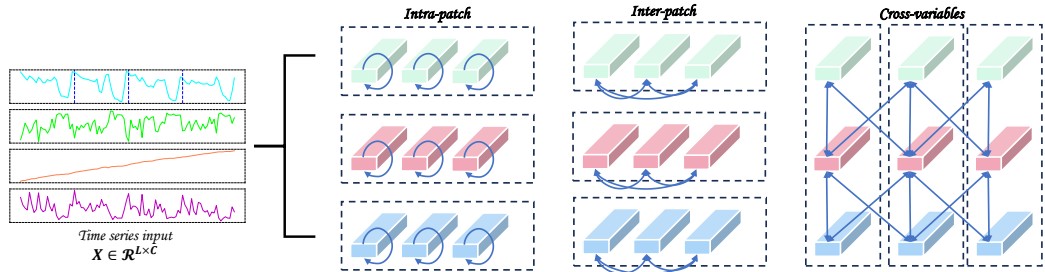

Figure 1: In the context of time series analysis, the three-dimensional variation encompasses **intra-patch**, **inter-patch**, and **cross-variable** interactions.

capture multiple dependencies(Lai et al., 2018; Luo & Wang, 2024), particularly the dynamic convolutions frequently employed in video processing, to attempt comprehensive capture of complex temporal patterns in time series data.

Modeling one-dimensional time variations can be a complex task due to the intricate patterns involved. Real-world time series often exhibit complex characteristics that require thorough analysis to interpret short-term fluctuations, long-term trends, and temporal pattern changes. Concurrently, the cross-variable dependencies within multivariate time series must be considered. To extract these crucial relationships from time series data, we specifically select intra-patch, inter-patch, and cross-variable(Figure 1) dependencies, derived from time series characteristics. These three types of embeddings encompass many significant information flows present in time series data.(Liang, 2014; Hlaváčková-Schindler et al., 2007)

Motivated by the above considerations, we introduce a novel analytical approach within the realm of time series analysis by integrating dynamic convolution. Specifically, we introduce a 3D-Embedding technique to convert one-dimensional time series tensors into three-dimensional tensors. Subsequently, we propose a network architecture, termed **TVNet (Time-Video Net)**, which leverages dynamic convolution to effectively capture intra-patch, inter-patch, and cross-variable dependencies for comprehensive time series analysis. We assessed the performance of TVNet across five benchmark analytical tasks: long-term and short-term forecasting, imputation, classification, and anomaly detection. Notably, despite being a purely convolution-based model, TVNet consistently achieves state-of-the-art results in these domains. Importantly, TVNet retains the efficiency inherent to dynamic convolution models, thereby achieving a superior balance between efficiency and performance, providing a practical and effective solution for time series analysis. **Our contributions are as follows:**

- A novel modeling approach that captures intra-patch,inter-patch and cross-variables features by converting 1D time series data into 3D shape tensor is proposed.
- TVNet implements consistent state-of-the-art performance time series analysis tasks across multiple mainstreams, demonstrating excellent task generalization.
- TVNet offers a better balance between efficiency and performance. It maintains the efficiency benefits of convolution based models while competing or even better with the most advanced base models in terms of performance.

## 2 RELATED WORK

### 2.1 TIME SERIES ANALYSIS

Traditional time series analysis methods, such as ARIMA (Anderson, 1976) and Holt-Winters models (Hyndman, 2018), despite their solid theoretical underpinnings, are insufficient for datasets exhibiting intricate temporal patterns. In recent years, deep learning approaches, particularly MLP-based and Transformer-based models, have achieved significant advancements in this domain (Li et al., 2023a; Challu et al., 2023; Liu et al., 2021; Zhang & Yan, 2023; Zhou et al., 2022; Wu

et al., 2021; Liu et al., 2023). MLP-based models have garnered increasing attention in time series analysis due to their low computational complexity. For instance, the Dlinear model (Li et al., 2023a) achieves efficient time series prediction by integrating decomposition techniques with MLP. However, current MLP models still struggle with capturing multi-period characteristics of time series and the relationship between different variables. Concurrently, Transformer-based models have demonstrated formidable performance in time series. Their self-attention mechanisms enable the capture of both long-term dependencies and critical global information. Nevertheless, their scalability and efficiency are constrained by the quadratic complexity of attention mechanisms. To mitigate this, researchers have introduced techniques to diminish the computational load of Transformers. For example, Pyraformer (Liu et al., 2021) enhances the model architecture with a pyramid attention design, facilitating connections across different scales. The recent PatchTST (Nie et al., 2022) employs a block-based strategy, enhancing both local feature capture and long-term prediction accuracy. Despite these innovations, existing Transformer-based methods, which predominantly model 1D temporal variations, continue to encounter substantial computational challenges in long-term time series analysis.

## 2.2 Convolution In Time Series Analysis

As deep learning techniques advance, Transformer-based and MLP-based models have achieved notable success in time series analysis, overshadowing traditional convolutional methods. Yet, innovative research is revitalizing convolution for time series analysis. The MICN model (Wang et al., 2023) integrates local features and global correlations by employing a multi-scale convolution structure. SCINet (Liu et al., 2022a) innovates by discarding causal convolution in favor of a recursive downsample-convolution interaction architecture to handle complex temporal dynamics. While these models have made strides in capturing long-term dependencies, they are challenged in the generality of time series tasks. TimesNet (Wu et al., 2022a) uniquely transforms one-dimensional time series into 2D forms, leveraging two-dimensional convolutional techniques from computer vision to enhance information representation. ModernTCN (Luo & Wang, 2024) extends the scope of convolutional technology by utilizing large convolution kernels to capture global time series features. However, these models predominantly focus on feature analysis within a single time window, neglecting the comprehensive integration of global, local, and cross-variable interactions.

## 2.3 Dynamic Convolution In Video

Dynamic convolution, characterized by their adaptive weight or module adjustments, exhibit structural flexibility in response to content variations. These networks incorporate dynamic filtering or convolution mechanisms ((Jia et al., 2016), (Yang et al., 2019)). They demonstrate superior processing capabilities and enhanced performance relative to conventional static networks. In the domain of video understanding, dynamic networks have also proven effective. For instance, dynamic filtering techniques and temporal aggregation methods adeptly capture temporal dynamics within videos, thereby augmenting the models' capacity to represent time series data.((Meng et al., 2021), (Li et al., 2020))

## 3 TVNet

Based on the inter-patch, intra-patch, and cross-variable features of the time series mentioned above, this paper proposes TVNet to capture the features of the time series. Specifically, we first design the 3D-embedding module so that the 1D time series can be converted to 3D time series tensor, and a 3D-block is designed to model the three types of properties (inter-patch,intra-patch, and cross-variable) through dynamic convolution. Finally, time series representations are extracted by 3D-blocks to get different task outputs through the linear layer. Figure 2 shows the structure of TVNet.

## 3.1 3D-Embedding

In this section, we propose a unique embedding way(**3D-Embedding**). This method takes into account the inter-patch, intra-patch and cross-variable for time series.We denote $X_{\text{in}} \in \mathbb{R}^{L \times C}$,$L$ means the length of time series,$C$ means the dimensions of time series.

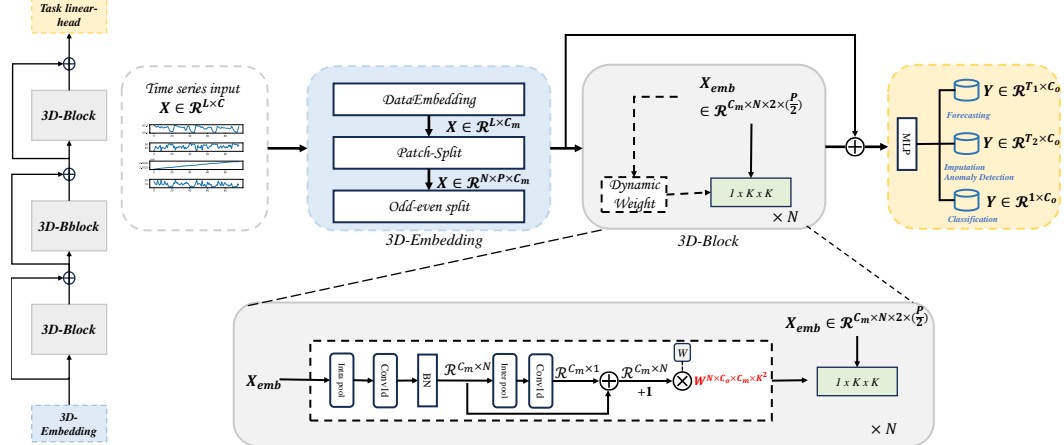

Figure 2: The overarching architecture of **TVNet** is constructed by stacking 3D-blocks in a residual way, which enables the capture of inter-patch, intra-patch, and cross-variable features from the time series 3D tensor.

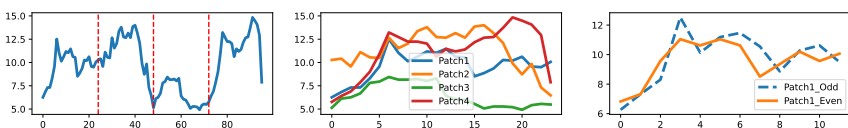

Figure 3: A univariate time series example is segmented into four distinct patches and the odd and even components specifically for Patch1 (ETTh1).

We first embed the input along the feature dimensions $C$ to $C_m$ and get $X_{\text{in}} \in \mathbb{R}^{L \times C_m}$, where $C_m$ as the embedding dimensions. Embedding dimension $C_m$ is a crucial hyperparameter that dictates the dimensionality of the representation space into which each time point is projected. After embedding the input data along the feature dimension to obtain $X_{\text{in}} \in \mathbb{R}^{L \times C_m}$, we proceed to segment the embedded data into patches using a one-dimensional convolutional layer (Conv1D). Specifically, we configure the Conv1D layer with a kernel size $P$, which defines the length of each patch. This configuration allows the Conv1D layer to divide the input sequence into $N$ patches, where $N$ is calculated based on the total length $L$, the patch length $P$(In the absence of overlapping patches, $N = L/P$). The output of this Conv1D operation is a tensor $X_{\text{emb}} \in \mathbb{R}^{N \times P \times C_m}$, with each $N$ representing the number of patches, $P$ representing the length of each patch, and $C_m$ representing the embedding dimensions. Then we use odd-even split(Liu et al., 2022a) on the patch length demisions to get $X_{odd}$ and $X_{even} \in \mathbb{R}^{N \times (P/2) \times C_m}$, then stacking $X_{odd}$ and $X_{even}$ to get $X_{\text{emb}} \in \mathbb{R}^{N \times 2 \times (P/2) \times C_m}$. Figure 3 illustrates an example of patches and odd-even components applied to a univariate time series.(**see Algorithm** 2)

$$X_{\text{emb}} = \text{3D-Embedding}(X_{\text{in}}) \tag{1}$$

## 3.2   3D-BLOCK

**Time dynamic weight:** Previous research, as detailed in Wang et al. (2023), has highlighted both consistencies and discrepancies within time series data, particularly in the phenomenon known as mode drift.In response to these observations and inspired by dynamic convolution work in the video field(Huang et al., 2021), Regarding the 3D-Embedding technique mentioned earlier, we employ a dynamic convolutional method to capture the temporal dynamics within each patch. To clarify, the embedded representation $X_{\text{emb}}$ is envisioned as a collection of individual patches, denoted as $(x_1, x_2, \ldots, x_N)$. The weight associated with the $i$-th patch, denoted as $W_i$, is hypothesized to be decomposable into the product of a constant **time base weight** $\boldsymbol{W_b}$, which is common across all

patches, and a **time varying weight $\alpha_i$**, which is unique to each patch.

$$\tilde{x}_i = \boldsymbol{W_i} \cdot x_i = (\boldsymbol{\alpha_i} \cdot \boldsymbol{W_b}) \cdot x_i, \tag{2}$$

**Time varying weight generation:** To more accurately model the temporal dynamics of patches, the time-varying weight $\alpha_i$ must account for all patches. Based on the above starting point, we design an adaptive Time-varying weight generation that fully considers the interaction between inter-patch and intra-patch.It can be articulated as $\alpha_i = \mathcal{G}(X_{\text{emb}})$. Here, $\mathcal{G}$ denotes the generation function for the time-varying weight.To efficiently handle both inter-patch and intra-patch relationships and to manage pre-training weights effectively, the weights of the 3D-block can be initialized akin to standard convolutional layers. This initialization is realized by setting the weights of $\mathcal{F}(\mathbf{v}_{\text{inter}}) + \mathcal{F}(\mathbf{v}_{\text{intra}})$ and augmenting the formulation with a constant vector of ones(1).The Figure 4 shows the flow chart for Time varing weight generation.

$$\alpha_i = \mathcal{G}(X_{\text{emb}}) = 1 + \mathcal{F}(\mathbf{v}_{\text{inter}}) + \mathcal{F}(\mathbf{v}_{\text{intra}}) \tag{3}$$

In this way, at the initial state,$W_i = 1 \cdot W_b$.

The function $\mathcal{F}(\mathbf{v}_{\text{intra}})$ captures intra-patch features. Initially, we apply 3D Adaptive Average Pooling to the embedded representation $X_{\text{emb}}$ to obtain intra-patch description vectors $\mathbf{v}_{\text{intra}}$, which encapsulate the essential features of each patch. These vectors are of dimension $\mathbb{R}^{C_m \times N}$. This operation is described by the following equation:

$$\mathbf{v}_{\text{intra}} = \text{AdaptiveAvgPool3d}(X_{\text{emb}}) \tag{4}$$

Subsequently, for intra-patch channel modeling, we employ a single-layer Conv1D, denoted as $\mathcal{F}_{\text{intra}}$, on $\mathbf{v}_{\text{intra}}$.

$$\mathcal{F}_{\text{intra}}(\mathbf{v}_{\text{intra}}) = \delta(\text{BN}(\text{Conv1D}^{C \to C}(\mathbf{v}_{\text{intra}}))) \tag{5}$$

Here, $\delta$ represents ReLU activation, and BN denotes Batch Normalization.

The function $\mathcal{F}(\mathbf{v}_{\text{inter}})$ is tasked with capturing features for inter-patches. We employ Adaptive Average Pooling 1D on $\mathbf{v}_{\text{intra}}$ to derive $\mathbf{v}_{\text{inter}}$. This process can be expressed as:

$$\mathbf{v}_{\text{inter}} \in \mathbb{R}^{C_m \times 1} = \text{AdaptiveAvgPool1d}(\mathbf{v}_{\text{intra}}) \tag{6}$$

Subsequently, for inter-patch channel modeling, a single-layer Conv1D, denoted as $\mathcal{F}_{\text{inter}}$, is applied to $\mathbf{v}_{\text{inter}}$:

$$\mathcal{F}_{\text{inter}}(\mathbf{v}_{\text{inter}}) = \delta(\text{Conv1D}^{C \to C}(\mathbf{v}_{\text{inter}})) \tag{7}$$

Here, $\delta$ denotes the ReLU activation function.

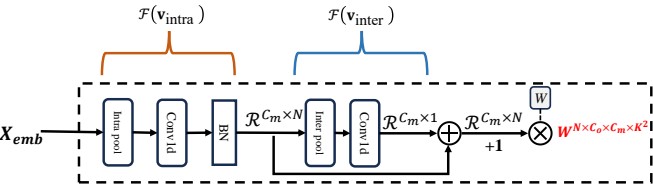

Figure 4: The Time varying weight generation flow chart

## 3.3 OVERALL STRUCTURE

Following the 3D-Embedding process, the embedded representation $X_{\text{emb}}$ can be permuted to the form $X_{\text{emb}} \in \mathbb{R}^{C_m \times N \times 2 \times (P/2)}$, which is then input into the 3D-block architecture. The purpose of this step is to learn effective representations of the time series, denoted as $X^{3D} \in \mathbb{R}^{C_m \times N \times 2 \times (P/2)}$. Subsequently, $X^{3D}$ is reshaped into $X \in \mathbb{R}^{(NP) \times C_m}$ and passed through linear layer, also referred to as the task-linear head, to accommodate the specific requirements of various tasks.

$$X^{3D} = \text{TVNet}(X_{\text{emb}}) \tag{8}$$

TVNet$(\cdot)$ is the stacked 3D-blocks. Each 3D-block is organized in a residual way. The forward process in the i-th 3D block is:

$$X_{i+1}^{3D} = \text{3D-block}(X_i^{3D}) + X_i^{3D} \tag{9}$$

where $X_i^{3D} \in \mathbb{R}^{C_m \times N \times 2 \times (P/2)}$ is the i-th block's input.((**see Algorithm** 1))

---

**Algorithm 1** Training of TVNet.

---

**Require: Input:** Training set $\mathcal{D} = \{(X, Y)\}$ Look back window $X \in \mathbb{R}^{L \times C}$;Embedding dimensions $C_m$;number of 3D-blocks $L$;Patch length$P$ and other training hyperparameter.
**Ensure: Output:** Trained TVNet
 1: Initialize Embedding dimensions and number of 3D-blocks.
 2: Sample $(X, Y)$ from $\mathcal{D}$
 3: (3D-Embdding): $X_{emb} = Algorithm(X)$
 4: **for** for $i$ 1 to $L$ do: **do**
 5:     $\alpha \leftarrow \mathcal{G}(X_{emb})$
 6:     $X_i^{3D} \leftarrow Conv2D(\alpha, W_b, X_{i-1}^{3D})$
 7: **end for**
 8: $\hat{Y} \leftarrow$ Task linear head$(X^{3D})$
 9: Compute loss $\mathcal{L} \leftarrow \mathcal{L}(\hat{Y}, Y)$
10: Update model parameters $\Theta$ via backpropagation
11: until convergence
12: return Trained TVNet

---

# 4 EXPERIMENT

TVNet is evaluated across five common analytical tasks, including both long-term and short-term forecasting, data imputation, classification, and anomaly detection, to demonstrate its adaptability in diverse applications.

**Baselines:** In establishing foundational models for time series analysis, we have extensively incorporated the most recent and advanced models from the time series community as benchmarks. This includes Transformer-based models such as iTransformer, PatchTST, Crossformer, and FEDformer(Liu et al., 2023; Nie et al., 2022; Zhang & Yan, 2023; Zhou et al., 2022); MLP-based models like MTS-Mixer, Dlinear RMLP, and RLinear(Li et al., 2023b; Zeng et al., 2023; Li et al., 2023a); and Convolution-based models including TimesNet, MICN, and ModernTCN(Wu et al., 2022a; Wang et al., 2023; Luo & Wang, 2024). Additionally, we have included state-of-the-art models from each specific task to serve as supplementary benchmarks for a comprehensive comparison. Our objective is to ensure that our model exhibits competitiveness when benchmarked against these advanced models.

**Hyperparameter Setups:** The performance of a model can be influenced by various hyperparameter settings. In this study, the hyperparameter ranges for TVNet are aligned with those reported in Wu et al. (2022a). A detailed discussion on the impact of each hyperparameter is provided in **Appendices C and D.**

**Main results:** As depicted in Figure 5, **TVNet consistently achieves top-tier performance across five pivotal analytical tasks, showcasing enhanced efficiency.** A detailed analysis of the experi-

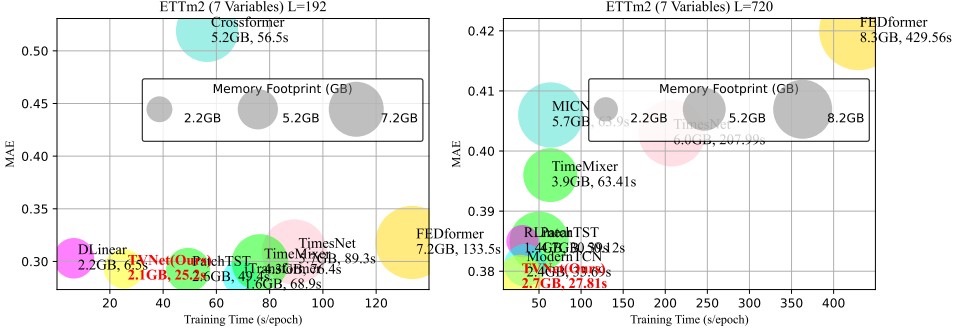

Figure 5: Model efficiency comparison under the setting of $L$(prediction length) =192/720 of ETTm2.

mental results is presented in Section 5.1. The details and outcomes of the experiments for each task are discussed in the following subsections. In the tables, the superior results are highlighted in **bold**, while those ranking just below the top are indicated with underlined.

## 4.1 LONG-TERM FORECASTING

**Datasets and setups:** We performed long-term forecasting experiments on nine well-established real-world benchmarks: Weather(wet), Traffic(pem), Electricity(uci), Exchange(Lai et al., 2018), ILI(cdc), and four ETT datasets(Zhou et al., 2021). To ensure equitable comparison, we re-executed all baseline models with diverse input lengths, opting for the best outcomes to prevent underestimating their performance. Mean Squared Error (MSE) and Mean Absolute Error (MAE) are adopted as the metrics for evaluating multivariate time series forecast.

**Results:** Table 1 illustrates the exceptional performance of TVNet in long-term forecasting. Specifically, TVNet outperformed the majority of existing MLP-based, Transformer-based, and Convolution-based models across all nine datasets. This outcome suggests that our design significantly improves the predictive capabilities of time series forecasting via convolution.

Table 1: Long-term forecasting results are averaged across four prediction lengths: {*24*, *36*, *48*, *60*} for ILI and {*96*, *192*, *336*, *720*} for others. Lower MSE or MAE values indicate superior performance. Refer to Table 28 in the Appendix for comprehensive results.(Hint:Baseline results are derived from their papers)

| Models | | TVNet (Ours) | | PatchTST (2022) | | iTransformer (2023) | | Crossformer (2023) | | RLinear (2023a) | | MTS-Mixer (2023b) | | DLinear (2023b) | | TimesNet (2022a) | | MICN (2023) | | ModernTCN (2024) | | FEDformer (2022) | | RMLP (2023a) | |
|---|---|---|---|---|---|---|---|---|---|---|---|---|---|---|---|---|---|---|---|---|---|---|---|---|---|
| Metrics | | MSE | MAE | MSE | MAE | MSE | MAE | MSE | MAE | MSE | MAE | MSE | MAE | MSE | MAE | MSE | MAE | MSE | MAE | MSE | MAE | MSE | MAE | MSE | MAE |
| ETTm1 | Avg | **0.348** | **0.379** | 0.351 | 0.381 | 0.407 | 0.410 | 0.431 | 0.443 | 0.358 | 0.376 | 0.370 | 0.395 | 0.357 | 0.379 | 0.400 | 0.450 | 0.383 | 0.406 | 0.351 | 0.381 | 0.382 | 0.422 | 0.369 | 0.393 |
| ETTm2 | Avg | **0.251** | **0.311** | 0.255 | 0.315 | 0.288 | 0.332 | 0.632 | 0.578 | 0.256 | 0.314 | 0.277 | 0.325 | 0.267 | 0.332 | 0.291 | 0.333 | 0.277 | 0.336 | 0.253 | 0.314 | 0.292 | 0.343 | 0.268 | 0.322 |
| ETTh1 | Avg | 0.407 | 0.421 | 0.413 | 0.431 | 0.454 | 0.447 | 0.441 | 0.465 | 0.408 | 0.421 | 0.430 | 0.436 | 0.423 | 0.437 | 0.458 | 0.450 | 0.433 | 0.462 | **0.404** | **0.420** | 0.428 | 0.454 | 0.442 | 0.445 |
| ETTh2 | Avg | 0.324 | 0.377 | 0.330 | 0.379 | 0.383 | 0.407 | 0.835 | 0.676 | **0.320** | **0.378** | 0.386 | 0.413 | 0.431 | 0.447 | 0.414 | 0.427 | 0.385 | 0.430 | 0.322 | 0.379 | 0.388 | 0.434 | 0.349 | 0.395 |
| Electricity | Avg | 0.165 | 0.254 | 0.159 | **0.253** | 0.178 | 0.270 | 0.293 | 0.351 | 0.169 | 0.261 | 0.173 | 0.272 | 0.177 | 0.274 | 0.192 | 0.295 | 0.182 | 0.292 | **0.156** | **0.253** | 0.207 | 0.321 | 0.161 | **0.253** |
| Weather | Avg | **0.221** | **0.261** | 0.226 | 0.264 | 0.258 | 0.278 | 0.230 | 0.290 | 0.247 | 0.279 | 0.235 | 0.272 | 0.240 | 0.300 | 0.259 | 0.287 | 0.242 | 0.298 | 0.224 | 0.264 | 0.310 | 0.357 | 0.225 | 0.265 |
| Traffic | Avg | **0.396** | 0.268 | 0.391 | **0.264** | 0.428 | 0.282 | 0.535 | 0.300 | 0.518 | 0.383 | 0.494 | 0.354 | 0.434 | 0.295 | 0.620 | 0.336 | 0.535 | 0.312 | 0.396 | 0.270 | 0.604 | 0.372 | 0.466 | 0.348 |
| Exchange | Avg | **0.298** | 0.367 | 0.387 | 0.419 | 0.360 | 0.403 | 0.701 | 0.633 | 0.345 | 0.394 | 0.373 | 0.407 | **0.297** | 0.378 | 0.416 | 0.443 | 0.315 | 0.404 | 0.302 | **0.366** | 0.478 | 0.478 | 0.345 | 0.394 |
| ILI | Avg | **1.406** | **0.766** | 1.443 | 0.798 | 2.141 | 0.996 | 3.361 | 1.235 | 4.269 | 1.490 | 1.555 | 0.819 | 2.169 | 1.041 | 2.139 | 0.931 | 2.567 | 1.055 | 1.440 | 0.786 | 2.597 | 1.070 | 4.387 | 1.516 |

## 4.2 SHORT-TERM FORECASTING

**Datasets and setups:** Our benchmark for short-term forecasting is the M4 dataset, as introduced by Makridakis (Makridakis et al., 2018). We have set the input sequence length to be twice that of the forecast horizon(Wu et al., 2022a). For evaluating performance, we utilize key metrics including the Symmetric Mean Absolute Percentage Error (SMAPE), Mean Absolute Scaled Error (MASE), and Overall Weighted Average (OWA). To enhance our comparative analysis, we have incorporated models such as U-Mixer (Ma et al., 2024) and N-Hits (Challu et al., 2023).

**Results:** Table 2 presents the findings. The short-term forecasting task on the M4 dataset is notably challenging due to the varied sources and temporal dynamics of the time series data. Nevertheless, TVNet retains its top ranking in this rigorous context, demonstrating its enhanced ability for temporal analysis.

Table 2: The task of short-term forecasting entails evaluating performance across various subdatasets, each with distinct sampling intervals. The results are presented as a weighted average, with lower metric values signifying enhanced performance. Full results are provided in Table 29

| | Models | TVNet (Ours) | PatchTST (2022) | U-Mixer (2024) | Crossformer (2023) | RLinear (2023a) | MTS-Mixer (2023b) | DLinear (2023b) | TimesNet (2022a) | MICN (2023) | ModernTCN (2024) | FEDformer (2022) | N-HiTS (2023) |
|---|---|---|---|---|---|---|---|---|---|---|---|---|---|
| WA | SMAPE | **11.671** | 11.807 | 11.740 | 13.474 | 12.473 | 11.892 | 13.639 | 11.829 | 13.130 | 11.698 | 12.840 | 11.927 |
| | MSE | **1.536** | 1.590 | 1.575 | 1.866 | 1.677 | 1.608 | 2.095 | 1.585 | 1.896 | 11.556 | 1.701 | 1.613 |
| | OWA | **0.832** | 0.851 | 0.845 | 0.985 | 0.898 | 0.859 | 1.051 | 0.851 | 0.980 | 0.838 | 0.918 | 0.861 |

### 4.3 IMPUTATION

**Datasets and setups:** The imputation task is designed to estimate missing values within partially observed time series data. Missing values are prevalent in time series due to unforeseen incidents such as equipment failure or communication errors. Given that missing values can adversely affect the performance of subsequent analyses, the imputation task holds significant practical importance.(Wu et al., 2022a) Subsequently, we concentrate on electricity and weather scenarios, which frequently exhibit data loss issues. We have chosen datasets from these domains as benchmarks, including ETT(Zhou et al., 2021), Electricity (UCI)(uci), and Weather(wet). To assess the model's capacity under varying degrees of missing data, we randomly introduce data occlusions at ratios of {*12.5%*, *25%*, *37.5%*, *50%*}.

**Results:** Table 3 shows TVNet's strong imputation performance despite irregular observations due to missing values. TVNet's top results validate its capability to capture temporal dependencies in complex scenarios. Cross-variable dependencies are key in imputation, aiding in estimating missing values when some variables are observed. Methods ignoring these, like PatchTST(Nie et al., 2022) and DLinear(Zeng et al., 2023), perform poorly in this task.

Table 3: Average results for the imputation task. Randomly masking {*12.5%*, *25%*, *37.5%*, *50%*} time points to compare the model performance under different missing degrees.Full results can be seen in 30

| Models | | **TVNet** (Ours) | | PatchTST (2022) | | SCINet (2022a) | | Crossformer (2023) | | RLinear (2023a) | | MTS-Mixer (2023b) | | DLinear (2023) | | TimesNet (2022a) | | MICN (2023) | | ModernTCN (2024) | | FEDformer (2022) | | RMLP (2023a) | |
|---|---|---|---|---|---|---|---|---|---|---|---|---|---|---|---|---|---|---|---|---|---|---|---|---|---|
| Metrics | | MSE | MAE | MSE | MAE | MSE | MAE | MSE | MAE | MSE | MAE | MSE | MAE | MSE | MAE | MSE | MAE | MSE | MAE | MSE | MAE | MSE | MAE | MSE | MAE |
| ETTm1 | Avg | **0.018** | **0.088** | 0.045 | 0.133 | 0.039 | 0.129 | 0.041 | 0.143 | 0.070 | 0.166 | 0.056 | 0.154 | 0.093 | 0.206 | 0.027 | 0.107 | 0.070 | 0.182 | 0.020 | 0.093 | 0.062 | 0.177 | 0.063 | 0.161 |
| ETTm2 | Avg | 0.022 | 0.086 | 0.028 | 0.098 | 0.027 | 0.102 | 0.046 | 0.149 | 0.032 | 0.108 | 0.032 | 0.107 | 0.096 | 0.208 | 0.022 | 0.088 | 0.144 | 0.249 | 0.019 | 0.082 | 0.101 | 0.215 | 0.032 | 0.109 |
| ETTh1 | Avg | **0.046** | **0.145** | 0.133 | 0.236 | 0.104 | 0.216 | 0.132 | 0.251 | 0.141 | 0.242 | 0.127 | 0.236 | 0.201 | 0.306 | 0.078 | 0.187 | 0.125 | 0.250 | 0.050 | 0.150 | 0.117 | 0.246 | 0.134 | 0.239 |
| ETTh2 | Avg | **0.039** | **0.127** | 0.066 | 0.164 | 0.064 | 0.165 | 0.122 | 0.240 | 0.066 | 0.165 | 0.069 | 0.168 | 0.142 | 0.259 | 0.049 | 0.146 | 0.205 | 0.307 | 0.042 | 0.131 | 0.163 | 0.279 | 0.068 | 0.166 |
| Electricity | Avg | 0.079 | 0.194 | 0.091 | 0.209 | 0.086 | 0.201 | 0.083 | 0.199 | 0.119 | 0.246 | 0.089 | 0.208 | 0.132 | 0.260 | 0.092 | 0.210 | 0.119 | 0.247 | **0.073** | **0.187** | 0.130 | 0.259 | 0.099 | 0.221 |
| Weather | Avg | **0.024** | **0.039** | 0.033 | 0.057 | 0.031 | 0.053 | 0.036 | 0.090 | 0.034 | 0.058 | 0.036 | 0.058 | 0.052 | 0.110 | 0.030 | 0.054 | 0.056 | 0.128 | 0.027 | 0.044 | 0.099 | 0.203 | 0.035 | 0.060 |

### 4.4 CLASSIFICATION AND ANOMALY DETECTION

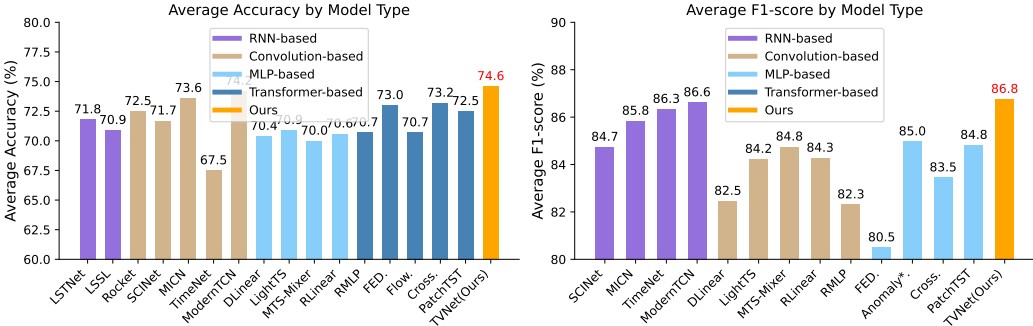

Figure 6: The results are averaged from several datasets.Higher accuracy and F1 score indicate better performance. [*] in the Transformer-based models indicates the name of [*]former. See Table 31 and 32 in Appendix for full results.

**Datasets and setups:** In our classification study, we selected 10 multivariate datasets from the UEA Time Series Classification Archive, as introduced by Bagnall (Bagnall et al., 2018) in 2018. These datasets, which serve as benchmarks, have been standardized using pre-processing practices established by Wu(Wu et al., 2022a). To ensure a comprehensive comparison, we have integrated state-of-the-art methods, including LSTNet (Lai et al., 2018), Rocket (Dempster et al., 2020), and Flowformer (Wu et al., 2022b), into our evaluation framework.

Our analysis covers a range of established benchmarks in the field of anomaly detection, including SMD (Su et al., 2019), SWaT (Mathur & Tippenhauer, 2016), PSM (Abdulaal et al., 2021), MSL,

and SMAP (Hundman et al., 2018). To enhance our comparative analysis, we have incorporated the Anomaly Transformer (Xu et al., 2021) into our baseline models. Our methodology is based on the traditional approach of reconstruction, wherein the deviation from the original, measured by the reconstruction error, is the key indicator for detecting anomalies.

**Results:** The results for time series classification and anomaly detection are depicted in Figure 6. In the classification task, TVNet attained the highest performance, with an average accuracy of 74.6%. For the anomaly detection task, TVNet exhibited competitive results 86.8 relative to the previous state-of-the-art.

## 5 MODEL ANALYSIS

### 5.1 COMPREHENSIVE COMPARISON OF PERFORMANCE AND EFFICIENCY

**Summary of results:** Relative to other task-specific models and prior state-of-the-art benchmarks, TVNet has consistently achieved state-of-the-art performance across five key analytical tasks, highlighting its exceptional versatility for diverse mission types. Moreover, TVNet's efficiency, as illustrated in Figure 5, facilitates a superior balance between efficiency and performance.

**Compared with baselines:** TVNet outperforms both Transformer-based and MLP-based models in terms of performance. Concurrently, it achieves faster training speeds and lower memory usage compared to Transformer-based models, thereby demonstrating superior efficiency. TVNet's comprehensive consideration of the relationships among time series inter-patch, intra-patch, and cross-variables enhances its representational capacity over the lightweight backbone of MLP models. Compared to ModernTCN, MICN, and TimesNet, TVNet employs dynamic convolution and 3D-Embedding techniques, significantly enhancing its memory efficiency and training speed while achieving the best possible results.

**Analysis of complexity:** TVNet exhibits a time complexity of $O(LC_m^2)$ and a space complexity of $O(C_m^2)$, where $C_m$ denotes the embedding dimension. A comparison of time complexity and memory usage during training and inference phases is presented in Table 4. Relative to Transformer-based models, TVNet demonstrates lower time complexity, and in contrast to convolutional models, it maintains independence between space complexity and sequence length.

Table 4: Comparison of different methods in terms of training time and memory complexity.

| Methods | Time Complexity | Space complexity |
|---|---|---|
| TVNet (Ours) | $O(LC_m^2)$ | $O(C_m^2 + LC_m)$ |
| MICN(Wang et al., 2023) | $O(LC_m^2)$ | $O(LC_m^2)$ |
| FEDformer(Zhou et al., 2022) | $O(L)$ | $O(L)$ |
| Autoformer(Wu et al., 2021) | $O(L \log L)$ | $O(L \log L)$ |
| Informer(Zhou et al., 2021) | $O(L \log L)$ | $O(L \log L)$ |
| Transformer(Liu et al., 2023) | $O(L^2)$ | $O(L^2)$ |
| LSTM | $O(L)$ | $O(L)$ |

### 5.2 ABLATION ANALYSIS

**Ablation of inter-pool module:** We conducted ablation experiments on the inter-pool module, and the experimental results are shown in the Table 5. It is found in the table that when inter-pool is removed, the predicted results will decrease, indicating that inter-pool can increase the representation of the relationship between different patches.

**Ablation of dynamic convolution:** Comparing the dynamic convolution weights to their fixed counterparts reveals a significant decline in predictive performance without the former. This observation underscores the effectiveness of the context-aware dynamic convolution proposed in this paper for modeling time series analysis.(Table 6) The dynamic convolution mechanism allows for a more nuanced capture of complex temporal patterns, enhancing the model's ability to adapt to varying data distributions and nonlinear relationships.

Table 5: Ablation of inter-pool module in the TVNet.w/o means without module.

| Models | Metrics | Weather | | | | Electricity | | | | Traffic | | | |
|---|---|---|---|---|---|---|---|---|---|---|---|---|---|
| | | 96 | 192 | 336 | 720 | 96 | 192 | 336 | 720 | 96 | 192 | 336 | 720 |
| TVNet | MSE | 0.147 | 0.194 | 0.235 | 0.308 | 0.142 | 0.165 | 0.164 | 0.190 | 0.367 | 0.381 | 0.395 | 0.442 |
| | MAE | 0.198 | 0.238 | 0.277 | 0.331 | 0.223 | 0.241 | 0.269 | 0.284 | 0.252 | 0.262 | 0.268 | 0.290 |
| w/o inter-pool | MSE | 0.156 | 0.211 | 0.247 | 0.349 | 0.147 | 0.178 | 0.186 | 0.199 | 0.377 | 0.399 | 0.410 | 0.458 |
| | MAE | 0.212 | 0.276 | 0.303 | 0.375 | 0.229 | 0.251 | 0.276 | 0.302 | 0.276 | 0.284 | 0.301 | 0.328 |

Table 6: Ablation of dynamic convolution in the TVNet.w/o means without module.

| Models | Metrics | Weather | | | | Electricity | | | | Traffic | | | |
|---|---|---|---|---|---|---|---|---|---|---|---|---|---|
| | | 96 | 192 | 336 | 720 | 96 | 192 | 336 | 720 | 96 | 192 | 336 | 720 |
| TVNet | MSE | **0.147** | **0.194** | **0.235** | **0.308** | **0.142** | **0.165** | **0.164** | **0.190** | **0.367** | **0.381** | **0.395** | **0.442** |
| | MAE | **0.198** | **0.238** | **0.277** | **0.331** | **0.223** | **0.241** | **0.269** | **0.284** | **0.252** | **0.262** | **0.268** | **0.290** |
| w/o dynamic weight | MSE | 0.251 | 0.257 | 0.291 | 0.341 | 0.185 | 0.201 | 0.214 | 0.268 | 0.391 | 0.423 | 0.440 | 0.507 |
| | MAE | 0.226 | 0.269 | 0.301 | 0.370 | 0.275 | 0.296 | 0.308 | 0.316 | 0.306 | 0.313 | 0.311 | 0.327 |

## 5.3 TRANSFER LEARNING

Figures 7 and 8 illustrate the outcomes of our transfer learning experiments. In both direct prediction and full-tuning approaches, TVNet outperforms the baseline models, underscoring its superior generalization and transfer capabilities. A pivotal advantage of TVNet is its dynamic weighting mechanism, which adeptly captures intricate temporal dynamics across a variety of datasets.

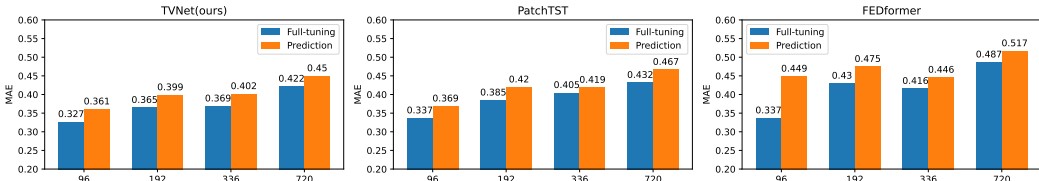

Figure 7: Transfer learning results for ETTh2.

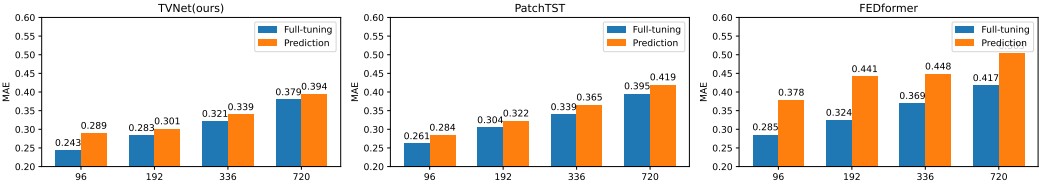

Figure 8: Transfer learning results for ETTm2.

## 6 CONCLUSION AND FUTURE WORK

This paper introduces a 3D variational reshaping and dynamic convolution method for general time series analysis. The experimental results demonstrate that TVNet exhibits excellent versatility across tasks, achieving performance comparable to or surpassing that of the most advanced Transformer-based and MLP-based models. TVNet also preserves the efficiency inherent in Convolution-based models, thus providing an optimal balance between performance and efficiency. Future research will focus on large-scale pre-training methods for time series and multi-scale patches, leveraging TVNet as the foundational architecture, which is anticipated to enhance capabilities across a broad spectrum of downstream tasks.

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

## A TVNet THEORETICAL DISCUSSIONS

**Problem Definition: (Time Series Analysis)** Time series analysis encompasses five primary sub-tasks: long-term forecasting, short-term forecasting, interpolation, classification, and anomaly detection. Mathematically, these tasks are defined with respect to a look back window $X \in \mathbb{R}^{L \times C}$, where $L$ is the window length and $C$ is the number of features. The goal is to develop a deep-learning model $f$ that minimizes the prediction error $e(Y, f(X))$. The target output $Y$ varies by task: - For forecasting: $Y \in \mathbb{R}^{T1 \times C_o}$, where $T1$ is the forecast horizon and $C_o$ is the number of output features. - For imputation and anomaly detection: $Y \in \mathbb{R}^{T2 \times C_o}$, where $T2$ is the data length for imputation or anomaly detection and $C_o$ is the number of output features. - For classification: $Y \in \mathbb{R}^{1 \times C}$, where $C$ represents the number of classification labels.

---

**Algorithm 2** 3D-Embedding applied to general time series analysis.

---

**Require:** Look back window $X \in \mathbb{R}^{L \times C}$, where $L$ is the length of the time series and $C$ is the number of channels.
  1: Embed the input along the feature dimensions $C$ to get $X' \in \mathbb{R}^{L \times C_m}$, where $C_m$ is the embedding dimension.
  2: Use Conv1D (kernel size = $P$) to divide $X'$ into $N$ patches $x_i \in \mathbb{R}^{P \times C_m}$, where $N = \frac{L}{P}$.
  3: **for** $i = 1$ to $N$ **do**
  4:     Split each patch $x_i$ by odd and even indices to get $x_{odd}, x_{even} \in \mathbb{R}^{P/2 \times C_m}$.
  5:     Concatenate $x_{odd}$ and $x_{even}$ to get $x_i \in \mathbb{R}^{2 \times (P/2) \times C_m}$.
  6: **end for**
  7: Concatenate all patches to get $X_{emb} \in \mathbb{R}^{N \times 2 \times (P/2) \times C_m}$.

---

### A.1 THEORETICAL ANALYSIS

**Theorem B.1:** Under appropriate optimization, the dynamic weight model achieves a lower error than the fixed weight model, i.e., $E_d < E_f$.

**Definition:** In the context of time series analysis, we can segment a dataset into $N$ distinct patches, denoted as $x_1, x_2, \ldots, x_N$, each with its corresponding true target value $y_i^*$. Two models are considered for processing these patches:

Fixed Weight Model: This model employs a constant weight $W_f$ to produce the convolution output for each patch $i$. The output is calculated as follows:

$$y_{f,i} = W_f \cdot x_i \tag{10}$$

This model uses variable weights $W_i$ for each patch, defined as $W_i = \alpha_i \cdot W_b$, where $\alpha_i$ is a coefficient that can vary with each patch. The convolution output for patch $i$ is then given by:

$$y_{d,i} = W_i \cdot x_i = (\alpha_i \cdot W_b) \cdot x_i \tag{11}$$

*Proof:*

For the fixed weight model, the total error is given by:

$$E_f = \sum_{i=1}^{N} (W_f \cdot x_i - y_i^*)^2 \tag{12}$$

Expanding the square term, we get:

$$E_f = \sum_{i=1}^{N} \left( (W_f \cdot x_i)^2 - 2W_f \cdot x_i \cdot y_i^* + (y_i^*)^2 \right) \tag{13}$$

For the dynamic weight model, the total error is given by:

$$E_d = \sum_{i=1}^{N} ((\alpha_i \cdot W_b \cdot x_i) - y_i^*)^2 \tag{14}$$

Expanding the square term, we get:

$$E_d = \sum_{i=1}^{N} \left( (\alpha_i \cdot W_b \cdot x_i)^2 - 2(\alpha_i \cdot W_b \cdot x_i) \cdot y_i^* + (y_i^*)^2 \right) \tag{15}$$

In the fixed weight model, the error $E_f$ is minimized by optimizing the single parameter $W_f$:

$$\frac{\partial E_f}{\partial W_f} = 0 \quad \Rightarrow \quad W_f = \text{optimal fixed weight} \tag{16}$$

In the dynamic weight model, the error $E_d$ is minimized by optimizing both $W_b$ (shared across patches) and $\alpha_i$ (specific to each patch). The optimization conditions are:

$$\frac{\partial E_d}{\partial W_b} = 0 \quad \text{and} \quad \frac{\partial E_d}{\partial \alpha_i} = 0, \forall i \tag{17}$$

Solving for $\alpha_i$:

$$\alpha_i = \frac{y_i^*}{W_b \cdot x_i}, \forall i \tag{18}$$

Substituting this back into $E_d$, the error is further reduced compared to the fixed weight model.

The dynamic weight model provides additional flexibility by adjusting $\alpha_i$ for each patch $i$. This allows it to better approximate the target $y_i^*$, resulting in:

$$E_d < E_f \tag{19}$$

## A.2 COMPUTATIONAL ANALYSIS:

Consider the input time series tensor $X^{3D} \in \mathbb{R}^{C_m \times N \times 2 \times \frac{P}{2}}$, where $N \times P = L$ and $L$ represents the time series length. Given that the output dimension of TVNet matches the input dimensions, the computational complexity for 2D convolutions can be described as follows:

$$\text{FLOPs(Conv2D)} = C_m \times C_m \times k^2 \times N \times 2 \times (P/2) = C_m \times C_m \times k^2 \times L$$
$$\text{Params(Conv2d)} = C_m \times C_m \times k^2 \tag{20}$$

$C_m$ denote the embedding dimensions and $k$ represent the kernel size of the 2D convolutions. In the process of generating time-varying weights, the features are initially passed through an intra-pool and subsequently through an inter-pool, which incorporates a non-parameter layer.

$$\text{FLOPs}(\mathcal{G}_{intra}) = C_m \times N \times 2 \times P/2 = C_m \times L$$
$$\text{Params}(\mathcal{G}_{inter}) = C_m \times N \tag{21}$$

In the channel modeling process, two 1D convolutions with a kernel size of $k = 1$ are utilized.

$$\text{FLOPs(channel)} = C_m \times C_m \times k(1) \times N + C_m \times C_m \times k(1)$$
$$\text{Params(channel)} = C_m \times C_m \times k(1) + C_m \times C_m \times k(1) \tag{22}$$

Subsequently, the time-varying weight matrix $\alpha \in \mathbb{R}^{C_m \times N}$ is multiplied by the temporal weight tensor $W \in \mathbb{R}^{C_m \times C_m \times k^2}$ for 2D convolutions.

$$\text{FLOPs(multipy)} = C_m \times C_m \times k^2 \times N \tag{23}$$

Therefore, the total computational complexity and parameter count are as follows:

$$\begin{aligned} \text{FLOPs} =& \text{FLOPs(Conv2D)} + \text{FLOPs}(\mathcal{G}) + \text{FLOPs(channel)} + \text{FLOPs(multiply)} \\ =& C_m \times L + C_m \times L + C_m \times C_m \times k(1) \times N \\ &+ C_m \times C_m \times k(1) + C_m \times C_m \times k^2 \times N \end{aligned} \tag{24}$$

$$\begin{aligned} \text{Params} =& \text{Params(Conv2D)} + \text{Params}(\mathcal{G}) + \text{Params(channel)} \\ =& C_m \times C_m \times k^2 + C_m \times N + C_m \times C_m \times k(1) + C_m \times C_m \times k(1) \end{aligned} \tag{25}$$

Based on the computational analysis, TVNet demonstrates FLOPs of $O(LC_m^2)$ and a parameter count of $O(C_m^2)$.

# B DATASETS

## B.1 LONG-TERM FORECAST AND IMPUTATION DATASETS

Assessments of the long-term forecasting capabilities were conducted using nine widely recognized real-world datasets, encompassing domains such as weather, traffic, electricity, exchange rates, influenza-like illness (ILI), and the four Electricity Transformer Temperature (ETT) datasets (ETTh1, ETTh2, ETTm1, ETTm2). For the imputation task, benchmarks were established using datasets from weather, electricity, and the four ETT datasets. These datasets, extensively utilized in the field, cover various aspects of daily life.

The characteristics of each dataset, including the total number of timesteps, the count of variables, and the sampling frequency, are summarized in Table 7. The datasets are partitioned into training, validation, and testing subsets in chronological order, with the Electricity Transformer Temperature (ETT) dataset employing a 6:2:2 ratio and the remaining datasets using a 7:1:2 ratio. Normalization to a zero mean is applied to the training, validation, and testing subsets based on the mean and standard deviation of the training subset. Each dataset comprises a single, continuous, long-time series, with samples extracted using a sliding window technique.

Further details regarding the datasets are as follows:

1. **Weather**[1] consists of 21 climatic variables, such as humidity and air temperature, recorded in Germany throughout 2020.

2. **Traffic**[2] includes road occupancy rates collected by 862 sensors across San Francisco Bay area highways over a two-year period, provided by the California Department of Transportation.

3. **Electricity**[3] comprises hourly electricity usage data for 321 consumers from 2012 to 2014.

4. **Exchange**[4] encompasses daily exchange rates for eight currencies, observed from 1990 to 2016.

5. **ILI**[5], which stands for Influenza-Like Illness, contains weekly counts of ILI patients in the United States from 2002 to 2021. It includes seven metrics, such as ILI patient counts across various age groups and the proportion of ILI patients relative to the total patient population. The data is provided by the Centers for Disease Control and Prevention of the United States.

6. **ETT**[6], The Electricity Transformer Temperature (ETT) dataset comprises data from seven sensors across two Chinese counties, featuring load and oil temperature metrics. It includes four subsets: 'ETTh1' and 'ETTh2' for hourly data, and 'ETTm1' and 'ETTm2' for 15-minute intervals.

Table 7: Dataset descriptions of long-term forecasting and imputation.

| Dataset | Weather | Traffic | Exchange | Electricity | ILI | ETTh1 | ETTh2 | ETTm1 | ETTm2 |
|---|---|---|---|---|---|---|---|---|---|
| Dataset Size | 52696 | 17544 | 7207 | 26304 | 966 | 17420 | 17420 | 69680 | 69680 |
| Variable Number | 21 | 862 | 8 | 321 | 7 | 7 | 7 | 7 | 7 |
| Sampling Frequency | 10 mins | 1 hour | 1 day | 1 hour | 1 week | 1 hour | 1 hour | 15 mins | 15 mins |

## B.2 SHORT-TERM FORECAST DATASETS

The M4 dataset, which includes 100,000 heterogeneous time series from various domains, poses a unique challenge for short-term forecasting. This is attributed to the diverse origins and distinct

---

[1]https://www.bgc-jena.mpg.de/wetter/

[2]https://pems.dot.ca.gov/

[3]https://archive.ics.uci.edu/dataset/321/electricityloaddiagrams20112014

[4]https://github.com/laiguokun/multivariate-time-series-data

[5]https://github.com/laiguokun/multivariate-time-series-data

[6]https://github.com/zhouhaoyi/ETDataset

temporal characteristics of the data, which contrast with the uniformity typically found in long-term forecasting datasets. A statistical overview of the dataset is provided in Table 8.

Table 8: Dataset descriptions of M4 forecasting

| Dataset | Sample Numbers (train set, test set) | Variable Number | Prediction Length |
|---------|-----------------------------|-----------------|-------------------|
| M4 Yearly | (23000, 23000) | 1 | 6 |
| M4 Quarterly | (24000, 24000) | 1 | 8 |
| M4 Monthly | (48000, 48000) | 1 | 18 |
| M4 Weekly | (359, 359) | 1 | 13 |
| M4 Daily | (4227, 4227) | 1 | 14 |
| M4 Hourly | (414, 414) | 1 | 48 |

### B.3  CLASSIFICATION DATASETS

The UEA dataset comprises a diverse collection of time series samples across various domains, designed for classification tasks. It encompasses a broad spectrum of recognition tasks, including facial, gesture, and action recognition, as well as audio identification. Furthermore, it extends its utility to practical applications in industrial monitoring, health surveillance, and medical diagnostics, with a particular emphasis on the analysis of cardiac data. Typically, the dataset is organized into 10 distinct classes. Table 9 offers a detailed overview of the classification statistics for the UEA datasets, underscoring their extensive applicability across multiple domains.

Table 9: Datasets and mapping details of UEA dataset

| Dataset | Sample Numbers (train set, test set) | Variable Number | Series Length |
|---------|-----------------------------|-----------------|---------------|
| EthanolConcentration | (261, 263) | 3 | 1751 |
| FaceDetection | (5890, 3524) | 144 | 62 |
| Handwriting | (150, 850) | 3 | 152 |
| Heartbeat | (204, 205) | 61 | 405 |
| JapaneseVowels | (270, 370) | 12 | 29 |
| PEMS - SF | (267, 173) | 963 | 144 |
| SelfRegulationSCP1 | (268, 293) | 6 | 896 |
| SelfRegulationSCP2 | (200, 180) | 7 | 1152 |
| SpokenArabicDigits | (6599, 2199) | 13 | 93 |
| UWaveGestureLibrary | (120, 320) | 3 | 315 |

### B.4  ANOMALY DETECTION DATASETS

Our benchmarking process utilizes datasets that cover a range of domains, such as server machinery, spacecraft, and infrastructure systems. These datasets are carefully organized into three distinct phases: training, validation, and testing. Each dataset consists of a single, continuous time series, and samples are extracted using a uniform, fixed-length sliding window method. Table 10 presents a comprehensive statistical summary of these datasets, illustrating their structured arrangement and the systematic approach to sample extraction.

Table 10: Dataset sizes and details for anomaly detection datasets

| Dataset | Dataset sizes (train set, val set, test set) | Variable Number | Sliding Window Length |
|---------|---------------------------------------|-----------------|----------------------|
| SMD | (566724, 141681, 708420) | 38 | 100 |
| MSL | (44653, 11664, 73729) | 55 | 100 |
| SMAP | (108146, 27037, 427617) | 25 | 100 |
| SWaT | (396000, 99000, 449919) | 51 | 100 |
| PSM | (105984, 26497, 87841) | 25 | 100 |

## B.5 DATASET VISUALIZATION

Figure 9,10,11 and 12 shows the pairwise correlation among the variables in ETT dataset.

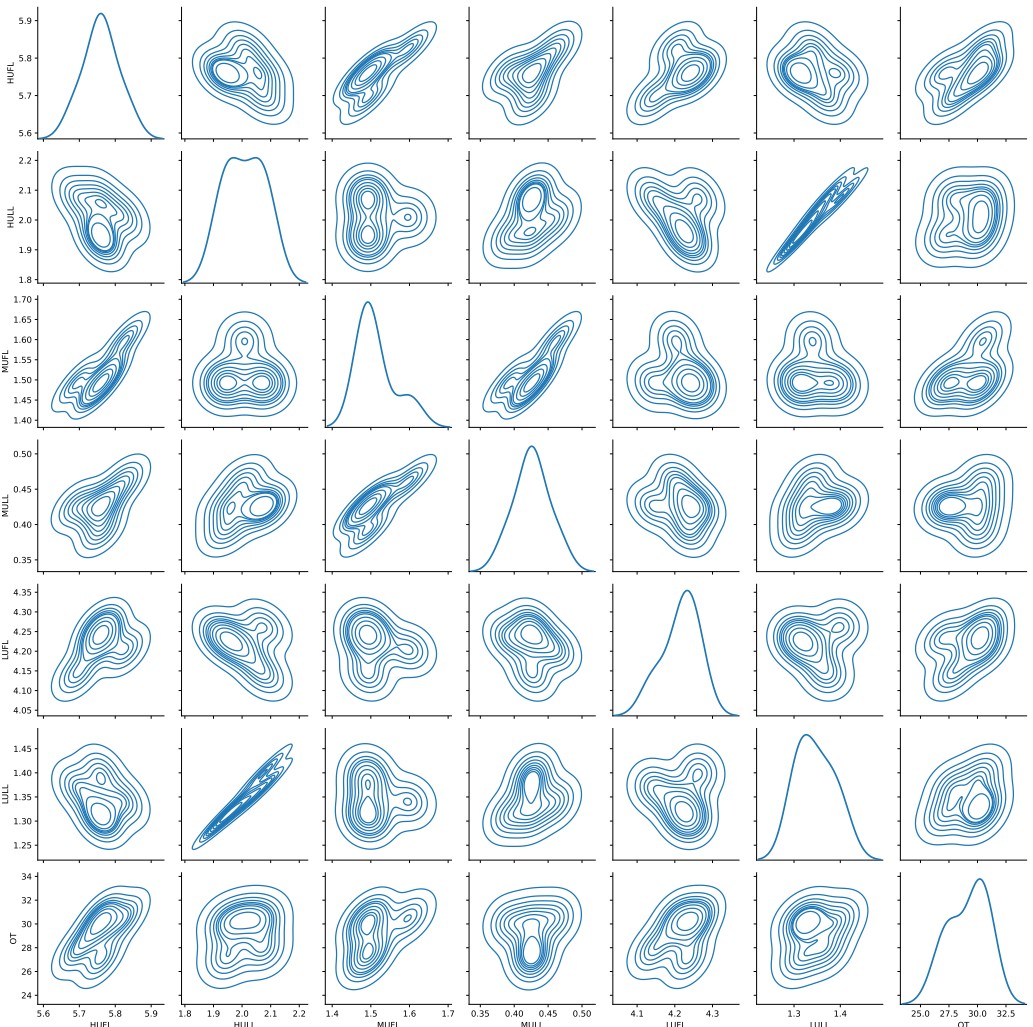

Figure 9: Pairwise correlation among the variables(ETTm1).

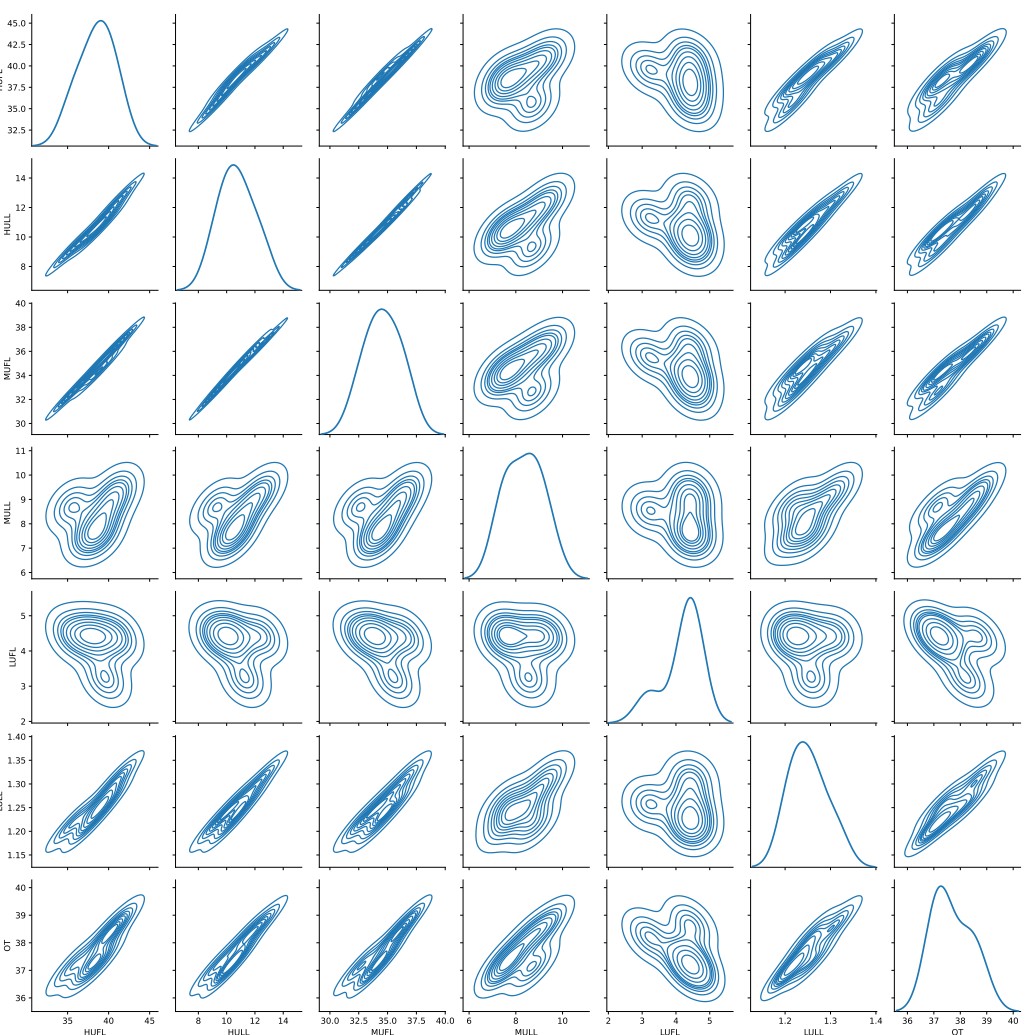

Figure 10: Pairwise correlation among the variables(ETTm2).

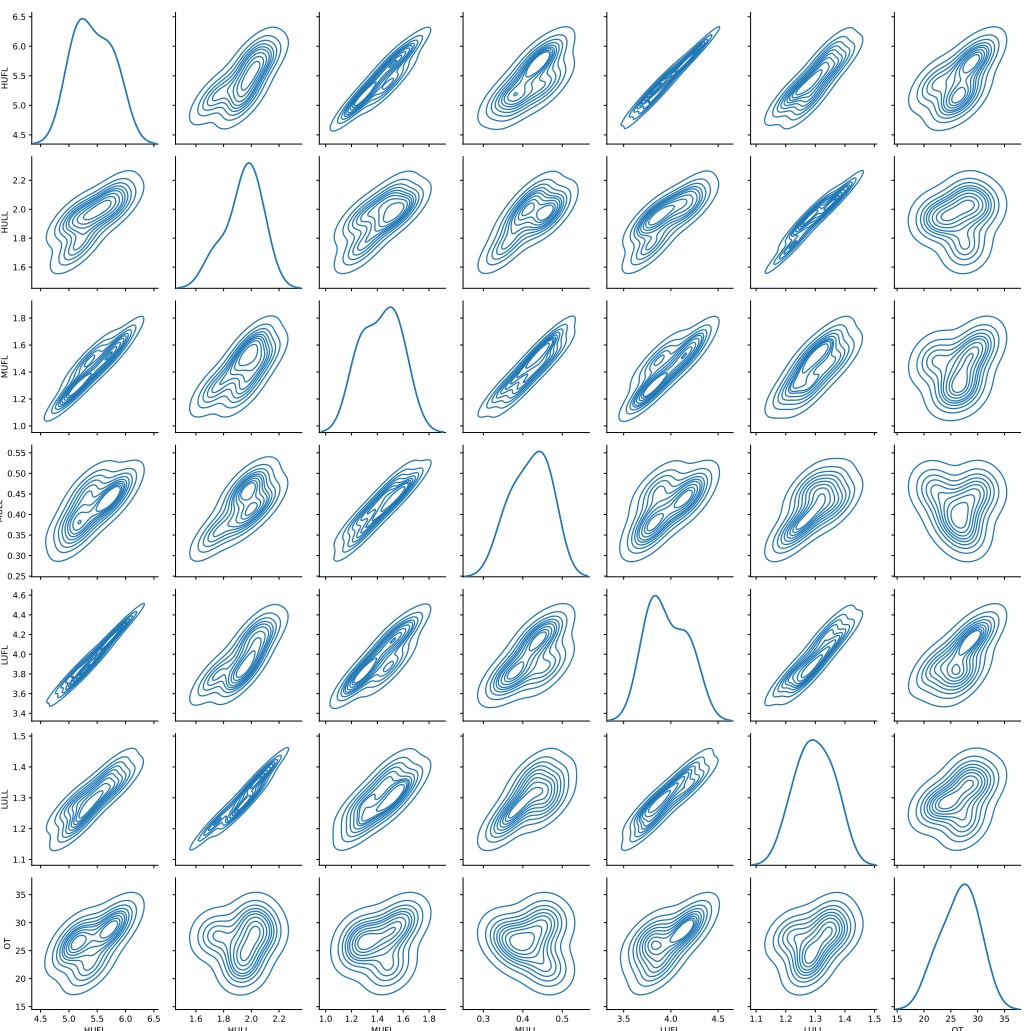

Figure 11: Pairwise correlation among the variables(ETTh1).

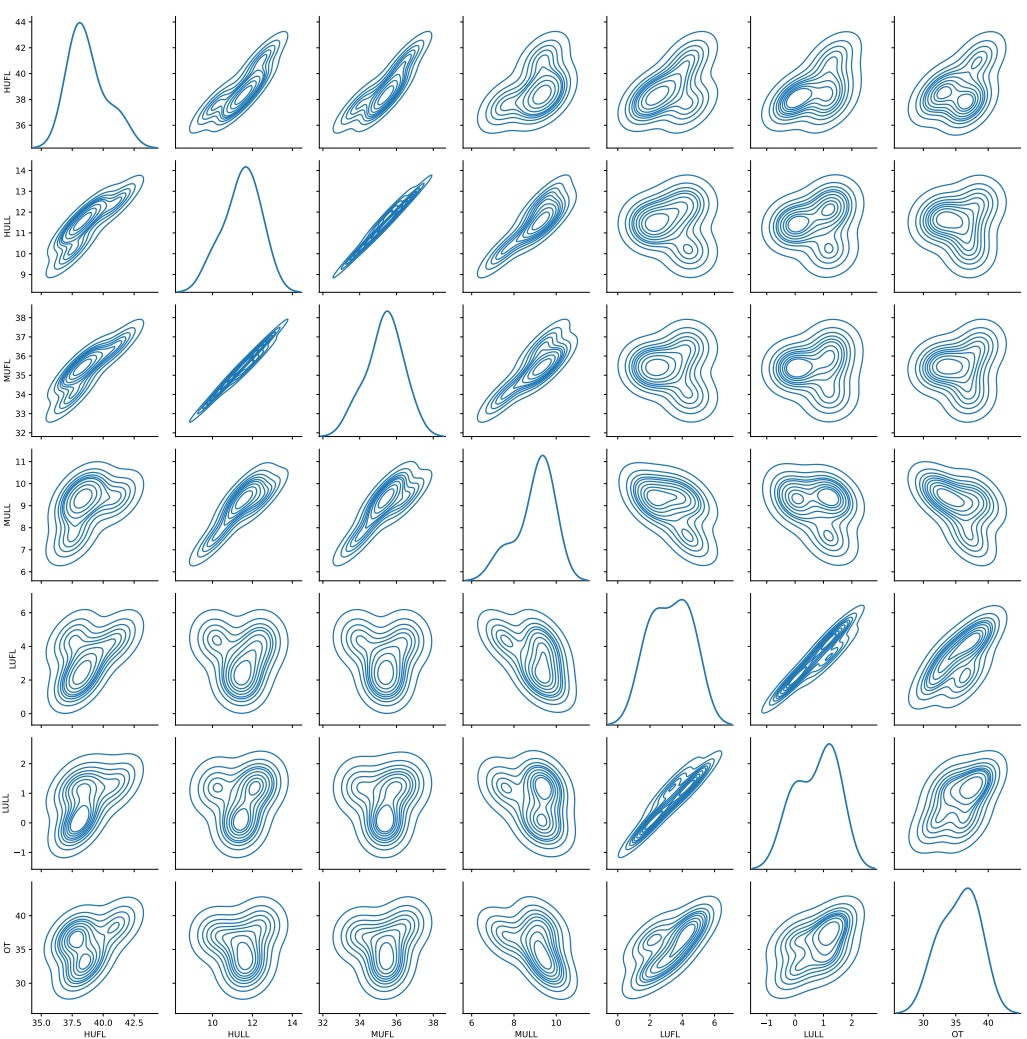

Figure 12: Pairwise correlation among the variables(ETTh2).

## C  EXPERIMENT DETAILS

Table 11: Model Hyper-parameters and Training Process for Different Tasks/Configurations(TVNet)

| Tasks/Configurations | Model Hyper-parameter | | | | Training Process | | | |
|---|---|---|---|---|---|---|---|---|
| | $k$ (kernel size) | Layers | $C_m$ | $P$ (Patch length) | LR | Loss | Batch Size | Epochs |
| Long-term Forecasting | $3x3$ | 3 | 64 | 24 | $10^{-4}$ | MSE | 32 | 10 |
| Short-term Forecasting | $3x3$ | 3 | 64 | 8 | $10^{-3}$ | SMAPE | 16 | 10 |
| Imputation | $3x3$ | 3 | 64 | 1 | $10^{-3}$ | MSE | 16 | 10 |
| Classification | $3x3$ | 3 | Eq31 | 1 | $10^{-3}$ | Cross Entropy | 16 | 30 |
| Anomaly Detection | $3x3$ | 5 | Eq32 | 8 | $10^{-4}$ | MSE | 128 | 10 |

- LR refers to the learning rate, and we have incorporated an early stopping mechanism to enhance the training process.For long-term forecasting task,Input lenght is set to 96Wu et al. (2022a).
- We performed experiments using 5 distinct random seeds: 1111, 333, 2023, 2024, and 2025.

### C.1  LONG-TERM FORECASTING

**Implementation details:** Our method utilizes the mean squared error (MSE) loss function and is optimized using the ADAM optimizer (Kingma, 2014), with an initial learning rate set to $10^{-4}$. The training regimen consists of 10 epochs, augmented by an appropriate early stopping mechanism. We employ the mean squared error (MSE) and mean absolute error (MAE) as evaluation metrics. The deep learning models are implemented using PyTorch (Paszke et al., 2019) and executed on NVIDIA RTX4090 GPU.

**Model parameter:** TVNet comprises 3 3D-blocks, with the embedding dimensions ($C_m$) set to 64. The patch length ($P$) is established at 24, and the kernel size ($k$) for 2D convolutions is $3 \times 3$. The random seed is set to 2024.

**Metrics:** The mean square error (MSE) and mean absolute error (MAE) are adopted for long-term forecasting.

$$MSE = \frac{1}{T} \sum_{i=0}^{T} (\widehat{x}_i - x_i)^2 \tag{26}$$

$$MAE = \frac{1}{T} \sum_{i=0}^{T} |\widehat{x}_i - x_i| \tag{27}$$

where $T$ is the number of observations, $\widehat{x}_i$ is the predicted value, and $x_i$ is the actual value for observation $i$.

### C.2  SHORT-TERM FORECASTING

**Implementation details:** Our method employs the Symmetric Mean Absolute Percentage Error (SMAPE) loss function and is optimized using the ADAM optimizer (Kingma, 2014), with an initial learning rate of $10^{-3}$. The training regimen consists of 10 epochs, augmented by an appropriate early stopping mechanism. We utilize the SMAPE, Mean Absolute Scaled Error (MASE), and Overall Weighted Average (OWA) as evaluation metrics.

**Model parameter:** TVNet consists of 3 3D-blocks, with the embedding dimension $C_m$ configured to 64. The patch length $P$ is set to 8, and the kernel size $k$ for 2D convolutions is $3 \times 3$.The random seed is set to 2024.

**Metrics:** For short-term forecasting, in accordance with (Wu et al., 2022a), we utilize the Symmetric Mean Absolute Percentage Error (SMAPE), Mean Absolute Scaled Error (MASE), and Overall Weighted Average (OWA) as the metrics. These metrics can be computed as follows:

$$\text{SMAPE} = \frac{200}{T} \sum_{i=1}^{T} \frac{|x_i - \widehat{x}_i|}{|x_i| + |\widehat{x}_i|} \tag{28}$$

$$\text{MASE} = \frac{1}{T} \sum_{i=1}^{T} \frac{|x_i - \widehat{x}_i|}{\frac{1}{T-p} \sum_{j=p+1}^{T} |x_j - x_{j-p}|} \tag{29}$$

$$\text{OWA} = \frac{1}{2} \left[ \frac{\text{SMAPE}}{\text{SMAPE}_{\text{Naive2}}} + \frac{\text{MASE}}{\text{MASE}_{\text{Naive2}}} \right] \tag{30}$$

where $p$ is the periodicity of data, $T$ is the number of observations, $\widehat{x}_i$ is the predicted value, and $x_i$ is the actual value for observation $i$.

### C.3 IMPUTATION

**Implementation details:** Our method utilizes the mean squared error (MSE) loss function and is optimized using the ADAM optimizer (Kingma, 2014), with an initial learning rate set to $10^{-3}$. The training regimen consists of 10 epochs, augmented by an appropriate early stopping mechanism. We employ the mean squared error (MSE) and mean absolute error (MAE) as evaluation metrics.

**Model parameter:** TVNet has 3 blocks.The patch length $P$ is set to 1 to avoid mixing the masked and unmasked tokens, with the embedding dimension $C_m$ configured to 64.The random seed is set to 2024

### C.4 CLASSIFICATION

**Implementation details:** Our method employs the Cross-Entropy Loss and is optimized using the ADAM optimizer, initialized with a learning rate of $10^{-3}$. The training regimen consists of 30 epochs, complemented by an appropriate early stopping mechanism. Classification accuracy serves as the metric for evaluation.

**Model parameter:** By default, TVNet comprises 3 blocks. The equation determines the channel number $C_m$.The random seed is set to 2024.

$$C_m = \min \left\{ \max \left( 2^{\lfloor \log_2 M \rfloor}, d_{\min} \right), d_{\max} \right\} \tag{31}$$

where $d_{\min}$ is 32 and $d_{\max}$ is 64, in accordance with the predefined criteria. The patch length $P$ is established at 1.

**Metrics:** For classification, the accuracy is calculated as the metric.

### C.5 ANOMALY DETECTION

**Implementation details:** In the classical reconstruction task, we employ MSE loss for training. The ADAM optimizer is utilized, initialized with a learning rate of $\times 10^{-4}$. The training regimen consists of 10 epochs, complemented by an appropriate early stopping mechanism. The reconstruction error, quantified as the Mean Squared Error (MSE), serves as the criterion for anomaly detection. The F1-Score is adopted as the metric.

**Model parameter:** By default, TVNet comprises 3 blocks. The channel number $C_m$ is determined by the equation

$$C_m = \min \left\{ \max \left( 2^{\lfloor \log_2 M \rfloor}, d_{\min} \right), d_{\max} \right\} \tag{32}$$

where $d_{\min}$ is 32 and $d_{\max}$ is 128, in accordance with the predefined criteria. The patch length $P$ is established at 8.

**Metrics:** For anomaly detection, we adopt the F1-score, which is the harmonic mean of precision and recall.

## D HYPERPARAMETER SENSITIVITY

### D.1 MODEL HYPERPARAMETERS

In this section, we assess the robustness of TVNet by examining its sensitivity to model hyperparameters. We will meticulously assess the impact of hyperparameter tuning on long-term forecasting

models, focusing on the embedding dimensions($C_m$), the number of blocks, and the patch length($P$) and kernel size($k$) to enhance the accuracy of our predictions.

**Embedding Dimensions and Number of blocks**:Figure 13 shows the different effect about DataEmbedding dimension($C_m = 32, 64, 128, 512$) and the number of 3D-blocks($1, 2, 3, 4, 5$) on ETT datasets(Prediction length = 96).Table 12 shows the effects of different $C_m$on the three data sets(ETTh1,Weather and Electricity) for long-term prediction tasks of different prediction lengths.For $Cm$, with the increase of dimension, MSE and MAE show an upward trend on ETTh1 and Weather datasets, but decrease first and then increase on Electricity datasets. In general, we can see that for different $C_m$, although the value of forecasting varies over the long term, the overall volatility is small.

We tested the influence of different 3D-Blocks($L$) on task prediction results in long-term forecasting,short-term forecasting and Anomaly detection.(Table 13,Table 14,Table 15) For long-term forecasting and short-term forecasting, $L = 2, 3, 4$ is tested; for Anomaly detection, $L = 4, 5, 6$ is tested. The selection of $L$is based on Table 11. The results show that as the number of 3D-block layers increases, the accuracy of prediction and anomaly detection generally increases slightly, which indicates that a single 3D-block has good time series characterization ability without relying on deep stacking. According to the above results, the model shows better robustness for DataEmbedding Dimension $C_m$and the number of 3D-Blocks.

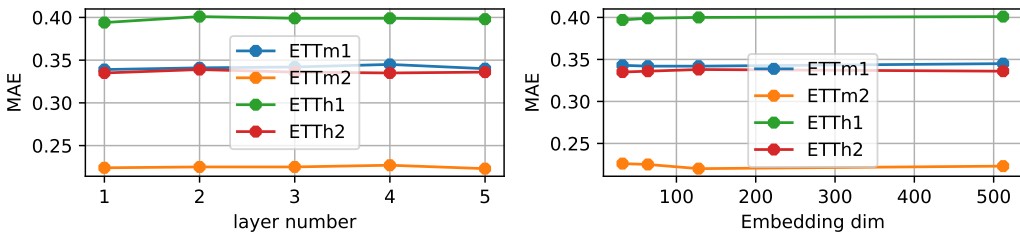

Figure 13: Hyperparameter sensitivity concerning the dimension of DataEmbedding and number of blocks

Table 12: Impact of different DataEmbedding dimension. A lower MSE or MAE indicates a better performance.

| Models | Metrics | ETTh1 | | | | Weather | | | | Electricity | | | |
|---|---|---|---|---|---|---|---|---|---|---|---|---|---|
| | | 96 | 192 | 336 | 720 | 96 | 192 | 336 | 720 | 96 | 192 | 336 | 720 |
| $C_m = 32$ | MSE | 0.369 | 0.399 | 0.403 | 0.460 | 0.149 | 0.192 | 0.235 | 0.306 | 0.139 | 0.168 | 0.162 | 0.189 |
| | MAE | 0.407 | 0.409 | 0.414 | 0.457 | 0.195 | 0.237 | 0.281 | 0.336 | 0.228 | 0.246 | 0.265 | 0.287 |
| $C_m = 64$ | MSE | 0.371 | 0.398 | 0.401 | 0.458 | 0.147 | 0.194 | 0.235 | 0.308 | 0.142 | 0.165 | 0.164 | 0.190 |
| | MAE | 0.408 | 0.409 | 0.409 | 0.459 | 0.198 | 0.238 | 0.277 | 0.331 | 0.223 | 0.241 | 0.269 | 0.284 |
| $C_m = 128$ | MSE | 0.365 | 0.389 | 0.395 | 0.451 | 0.153 | 0.187 | 0.240 | 0.306 | 0.140 | 0.157 | 0.166 | 0.182 |
| | MAE | 0.404 | 0.403 | 0.406 | 0.453 | 0.196 | 0.235 | 0.276 | 0.329 | 0.219 | 0.238 | 0.270 | 0.285 |

Table 13: Impact of different number of 3D-Blocks($L$). A lower MSE or MAE indicates a better performance.

| Models | Metrics | ETTh1 | | | | Weather | | | | Electricity | | | |
|---|---|---|---|---|---|---|---|---|---|---|---|---|---|
| | | 96 | 192 | 336 | 720 | 96 | 192 | 336 | 720 | 96 | 192 | 336 | 720 |
| $L = 2$ | MSE | 0.380 | 0.412 | 0.413 | 0.479 | 0.153 | 0.199 | 0.242 | 0.316 | 0.146 | 0.169 | 0.168 | 0.197 |
| | MAE | 0.410 | 0.419 | 0.417 | 0.475 | 0.202 | 0.244 | 0.285 | 0.339 | 0.230 | 0.247 | 0.275 | 0.291 |
| $L = 3$ | MSE | 0.371 | 0.398 | 0.401 | 0.458 | 0.147 | 0.194 | 0.235 | 0.308 | 0.142 | 0.165 | 0.164 | 0.190 |
| | MAE | 0.408 | 0.409 | 0.409 | 0.459 | 0.198 | 0.238 | 0.277 | 0.331 | 0.223 | 0.241 | 0.269 | 0.284 |
| $L = 4$ | MSE | 0.360 | 0.382 | 0.387 | 0.435 | 0.143 | 0.190 | 0.229 | 0.299 | 0.139 | 0.161 | 0.159 | 0.185 |
| | MAE | 0.399 | 0.402 | 0.399 | 0.443 | 0.194 | 0.232 | 0.275 | 0.321 | 0.216 | 0.235 | 0.269 | 0.277 |

Table 14: Impact of different number of 3D-Blocks($L$) for short-term forecasting

| Models | Yearly | | | Quarterly | | | Monthly | | | Others | | |
|---|---|---|---|---|---|---|---|---|---|---|---|---|
| | SMAPE | MASE | OWA | SMAEP | MASE | OWA | SMAPE | MASE | OWA | SMAPE | MASE | OWA |
| $L=2$ | 13.596 | 3.124 | 0.826 | 10.941 | 1.305 | 1.064 | 12.729 | 0.951 | 0.885 | 4.951 | 3.121 | 1.035 |
| $L=3$ | 13.217 | 2.899 | 0.768 | 9.986 | 1.159 | 0.876 | 12.493 | 0.921 | 0.866 | 4.764 | 2.986 | 0.969 |
| $L=4$ | 13.199 | 2.871 | 0.754 | 9.981 | 1.152 | 0.876 | 12.490 | 0.923 | 0.867 | 4.752 | 2.981 | 0.965 |

Table 15: Impact of different number of 3D-Blocks($L$) for Anomaly detection(F1)

| Dataset | Model | SMD | MSL | SMAP | SWaT | PSM |
|---|---|---|---|---|---|---|
| $L=4$ | Ours | 84.93 | 85.10 | 71.21 | 93.54 | 97.45 |
| $L=5$ | Ours | 85.75 | 85.16 | 71.64 | 93.72 | 97.53 |
| $L=6$ | Ours | 85.79 | 85.20 | 71.59 | 93.81 | 97.67 |

**Patch length and Kernel size:** In this section, we conducted experiments on long-term forecasting tasks by varying the Patch length ($P$) and Kernel size ($k$). We tested four distinct values for Patch length $(8, 12, 24, 32)$ on long-term forecasting.Further, we test different Patch lengths ($P$) in long-term forecasting and short-term forecasting(Table 17,Table 16), and the results show that TVNet is robust when a moderate Patch length is selected. Different $P$ has no significant influence on the result.To maintain the dimensionality of each feature map layer, we employed zero-padding equivalent to (kernel size - 1). The results presented in Table 18 indicate that both extremely small and large convolution kernels negatively impact the prediction accuracy. This study advocates for the adoption of a $3 \times 3$ convolution kernel, which strikes a balance between capturing local features and maintaining computational efficiency.For the Patch length ($P$), our findings indicate that both smaller and larger values can degrade the forecasting performance. In the context of long-term forecasting, we recommend setting the Patch length to 24, as it offers an optimal balance for prediction accuracy. For short term forecasting, 8 is recommended for patch-length.

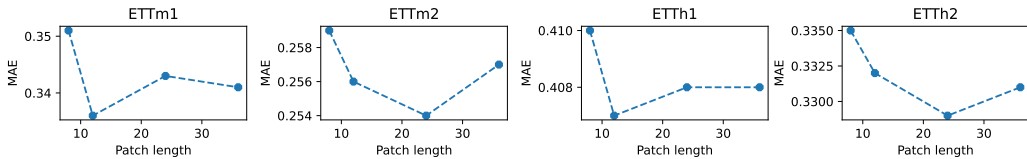

Figure 14: Hyperparameter sensitivity concerning patch length(prediction length=96)

Table 16: Impact of different Patch length.A lower MSE or MAE indicates a better performance.

| Models | Metrics | ETTh1 | | | | Weather | | | | Electricity | | | |
|---|---|---|---|---|---|---|---|---|---|---|---|---|---|
| | | 96 | 192 | 336 | 720 | 96 | 192 | 336 | 720 | 96 | 192 | 336 | 720 |
| $P=8$ | MSE | 0.384 | 0.406 | 0.404 | 0.463 | 0.153 | 0.197 | 0.236 | 0.309 | 0.147 | 0.163 | 0.172 | 0.193 |
| | MAE | 0.413 | 0.417 | 0.412 | 0.463 | 0.204 | 0.251 | 0.286 | 0.334 | 0.226 | 0.247 | 0.272 | 0.289 |
| $P=12$ | MSE | 0.369 | 0.404 | 0.402 | 0.449 | 0.153 | 0.205 | 0.237 | 0.311 | 0.146 | 0.170 | 0.163 | 0.191 |
| | MAE | 0.406 | 0.411 | 0.412 | 0.463 | 0.209 | 0.245 | 0.287 | 0.332 | 0.227 | 0.246 | 0.265 | 0.280 |
| $P=24$ | MSE | 0.371 | 0.398 | 0.401 | 0.458 | 0.147 | 0.194 | 0.235 | 0.308 | 0.142 | 0.165 | 0.164 | 0.190 |
| | MAE | 0.408 | 0.409 | 0.409 | 0.459 | 0.198 | 0.238 | 0.277 | 0.331 | 0.223 | 0.241 | 0.269 | 0.284 |
| $P=32$ | MSE | 0.368 | 0.395 | 0.399 | 0.454 | 0.150 | 0.192 | 0.241 | 0.311 | 0.145 | 0.167 | 0.155 | 0.192 |
| | MAE | 0.401 | 0.407 | 0.420 | 0.457 | 0.205 | 0.239 | 0.277 | 0.332 | 0.227 | 0.242 | 0.252 | 0.288 |

Table 17: Impact of different Patch length($P$) for short-term forecasting

| Models | Metrics | Yearly | | | Quarterly | | | Monthly | | | Others | | |
|---|---|---|---|---|---|---|---|---|---|---|---|---|---|
| | | SMAPE | MASE | OWA | SMAEP | MASE | OWA | SMAPE | MASE | OWA | SMAPE | MASE | OWA |
| $P=4$ | | 13.151 | 2.925 | 0.765 | 10.056 | 1.155 | 0.885 | 12.382 | 0.926 | 0.859 | 4.792 | 2.965 | 0.977 |
| $P=8$ | | 13.217 | 2.899 | 0.768 | 9.986 | 1.159 | 0.876 | 12.493 | 0.921 | 0.866 | 4.764 | 2.986 | 0.969 |
| $P=16$ | | 13.073 | 2.871 | 0.754 | 9.903 | 1.147 | 0.873 | 12.348 | 0.912 | 0.854 | 4.664 | 2.977 | 0.955 |

Table 18: Impact of different kernel sizes. A lower MSE or MAE indicates a better performance.

| Models | Metrics | ETTh1 | | | | ETTm1 | | | | Electricity | | | |
|--------|---------|-------|-------|-------|-------|-------|-------|-------|-------|-------|-------|-------|-------|
| | | 96 | 192 | 336 | 720 | 96 | 192 | 336 | 720 | 96 | 192 | 336 | 720 |
| $k=1$ | MSE | 0.375 | 0.407 | 0.405 | 0.464 | 0.293 | 0.327 | 0.371 | 0.412 | 0.143 | 0.171 | 0.169 | 0.196 |
| | MAE | 0.413 | 0.417 | 0.415 | 0.464 | 0.358 | 0.371 | 0.392 | 0.419 | 0.221 | 0.249 | 0.265 | 0.302 |
| $k=3$ | MSE | 0.371 | 0.398 | 0.401 | 0.458 | 0.288 | 0.326 | 0.365 | 0.412 | 0.142 | 0.165 | 0.164 | 0.190 |
| | MAE | 0.408 | 0.409 | 0.409 | 0.459 | 0.343 | 0.367 | 0.391 | 0.413 | 0.223 | 0.241 | 0.269 | 0.284 |
| $k=5$ | MSE | 0.369 | 0.396 | 0.399 | 0.454 | 0.289 | 0.324 | 0.367 | 0.417 | 0.146 | 0.168 | 0.165 | 0.187 |
| | MAE | 0.405 | 0.408 | 0.409 | 0.457 | 0.339 | 0.362 | 0.390 | 0.415 | 0.220 | 0.240 | 0.264 | 0.279 |
| $k=7$ | MSE | 0.370 | 0.399 | 0.398 | 0.450 | 0.291 | 0.325 | 0.368 | 0.420 | 0.147 | 0.167 | 0.164 | 0.189 |
| | MAE | 0.406 | 0.409 | 0.409 | 0.462 | 0.341 | 0.360 | 0.391 | 0.426 | 0.225 | 0.239 | 0.267 | 0.281 |
| $k=9$ | MSE | 0.386 | 0.413 | 0.407 | 0.474 | 0.302 | 0.349 | 0.403 | 0.419 | 0.157 | 0.189 | 0.194 | 0.212 |
| | MAE | 0.437 | 0.444 | 0.439 | 0.485 | 0.372 | 0.396 | 0.410 | 0.432 | 0.230 | 0.271 | 0.285 | 0.314 |

## D.2 TRANING PROCESS HYPERPARAMETERS

Furthermore, acknowledging that hyperparameters such as learning rate and batch size influence the results, we present experimental results inclusive of standard deviation. We explore batch sizes ranging from 32 to 512, learning rates from $10^{-5}$ to 0.05, five seeds(1111,333,2023,2024,2025), and training epochs from 10 to 100. The findings are summarized in Table 19,Table 20 and Table21 for long-term forecasting,short-term forecasting and Anomaly detection.From the above results, we can find that for different training process Settings, the standard deviation of the results of different time series tasks is smaller, indicating that the model shows high robustness for different training Settings.

Table 19: Detailed performance of TVNet. We report MSE/MAE and standard deviation of different forecasting horizons $\{24, 36, 48, 60\}$ for ILI and $\{96, 192, 336, 720\}$ for others.$H1, H2, H3, H4$ means the horizons.

| Horizon | Electricity | | ETTh2 | | Exchange | |
|---|---|---|---|---|---|---|
| | MSE | MAE | MSE | MAE | MSE | MAE |
| $H1$ | $0.142 \pm 0.002$ | $0.223 \pm 0.003$ | $0.263 \pm 0.003$ | $0.329 \pm 0.003$ | $0.080 \pm 0.002$ | $0.195 \pm 0.003$ |
| $H2$ | $0.165 \pm 0.002$ | $0.241 \pm 0.006$ | $0.319 \pm 0.003$ | $0.372 \pm 0.002$ | $0.163 \pm 0.004$ | $0.285 \pm 0.002$ |
| $H3$ | $0.164 \pm 0.003$ | $0.269 \pm 0.002$ | $0.311 \pm 0.004$ | $0.373 \pm 0.002$ | $0.291 \pm 0.003$ | $0.394 \pm 0.003$ |
| $H4$ | $0.190 \pm 0.005$ | $0.284 \pm 0.002$ | $0.401 \pm 0.003$ | $0.434 \pm 0.004$ | $0.658 \pm 0.007$ | $0.594 \pm 0.002$ |
| Horizon | ILI | | Traffic | | Weather | |
| | MSE | MAE | MSE | MAE | MSE | MAE |
| $H1$ | $1.324 \pm 0.006$ | $0.712 \pm 0.004$ | $0.367 \pm 0.003$ | $0.252 \pm 0.003$ | $0.147 \pm 0.005$ | $0.198 \pm 0.004$ |
| $H2$ | $1.190 \pm 0.020$ | $0.772 \pm 0.018$ | $0.381 \pm 0.003$ | $0.262 \pm 0.004$ | $0.194 \pm 0.003$ | $0.238 \pm 0.003$ |
| $H3$ | $1.456 \pm 0.015$ | $0.782 \pm 0.004$ | $0.395 \pm 0.004$ | $0.268 \pm 0.003$ | $0.235 \pm 0.005$ | $0.277 \pm 0.005$ |
| $H4$ | $1.652 \pm 0.022$ | $0.796 \pm 0.005$ | $0.442 \pm 0.004$ | $0.290 \pm 0.006$ | $0.308 \pm 0.003$ | $0.331 \pm 0.003$ |

Table 20: Detailed performance of TVNet. We report standard deviation on short-term forecasting

| Models | Metrics | Yearly | | | Quarterly | | | Monthly | | | Others | | |
|---|---|---|---|---|---|---|---|---|---|---|---|---|---|
| | | SMAPE | MASE | OWA | SMAEP | MASE | OWA | SMAPE | MASE | OWA | SMAPE | MASE | OWA |
| TVNet | | $13.217 \pm 0.005$ | $2.899 \pm 0.001$ | $0.768 \pm 0.004$ | $9.986 \pm 0.005$ | $1.159 \pm 0.002$ | $0.876 \pm 0.003$ | $12.493 \pm 0.003$ | $0.921 \pm 0.001$ | $0.866 \pm 0.002$ | $4.764 \pm 0.002$ | $2.986 \pm 0.002$ | $0.969 \pm 0.001$ |

Table 21: Detailed performance of TVNet. We report standard deviation on Anomaly detection(F1)

| Dataset | Model | SMD | MSL | SMAP | SWaT | PSM |
|---|---|---|---|---|---|---|
| Anomaly Detection(F1) | **Ours** | $85.75 \pm 0.06$ | $85.16 \pm 0.04$ | $71.64 \pm 00.03$ | $93.72 \pm 0.05$ | $97.53 \pm 0.09$ |
| Anomaly Detection(R) | **Ours** | $83.49 \pm 0.02$ | $86.47 \pm 0.03$ | $58.26 \pm 0.02$ | $96.33 \pm 0.05$ | $96.76 \pm 0.04$ |
| Anomaly Detection(P) | **Ours** | $88.03 \pm 0.03$ | $83.91 \pm 0.05$ | $92.64 \pm 0.06$ | $91.26 \pm 0.04$ | $98.30 \pm 0.04$ |

# E MORE ABLATION EXPERIMENTS

## E.1 INTER-POOL MODULE

In this section, we further conducted ablation studies on the tasks of short-term forecasting and time series anomaly detection, and presented the results.(Table 23)

## E.2 RESHAPE REPRESENTATION

The existing representations for time series include 1D, 2D (inter-pool, cross-dimensions), and 3D (inter-pool, intra-pool, and cross-dimensions). To demonstrate the effectiveness of the 3D-embedding(③) proposed in this paper, 1D-embedding$X_{emb} \in \mathcal{R}^{L \times C_m}$(label ① )and 2D-embedding $X_{emb} \in \mathcal{R}^{N \times P \times C_m}$(label ②) are designed following the same rationale.For the time series analysis task using **fixed weight convolution** for different methods, long-term forecasting(Prediction length =192) is tested.Table 24 shows the result.It can be seen from the results that the error is smaller than that of 1D-Embedding and 2D-Embedding.3D-Embedding, indicating that 3D-Embedding can better represent time series.

Table 22: Ablation of Inter-pool Module.(Anomaly detection(F1) and Short-term forecasting(OWA))

| Dataset | Model | SMD | MSL | SMAP | SWaT | PSM |
|---|---|---|---|---|---|---|
| Anomaly Detection | **Ours** | 85.75 | 85.16 | 71.64 | 93.72 | 97.53 |
| | w/o | 78.61 | 67.34 | 59.82 | 82.74 | 86.27 |
| Dataset | Model | Yearly | Quarterly | Monthly | Others | |
| Short-term forecasting | **Ours** | 0.768 | 0.876 | 0.866 | 0.969 | |
| | w/o | 0.809 | 1.231 | 0.929 | 1.205 | |

Table 23: Different Embedding ways for Anomaly-detection and short-term forecasting.

| Dataset | Model | SMD | MSL | SMAP | SWaT | PSM |
|---|---|---|---|---|---|---|
| Anomaly Detection | ① | 67.38 | 70.29 | 65.43 | 75.41 | 85.77 |
| | ② | 70.60 | 65.95 | 52.09 | 76.54 | 82.05 |
| | ③ | 80.21 | 83.24 | 69.41 | 87.65 | 92.34 |
| Dataset | Model | Yearly | Quarterly | Monthly | Others | |
| Short-term forecasting | ① | 1.043 | 1.576 | 1.132 | 1.574 | |
| | ② | 0.904 | 1.589 | 1.076 | 1.322 | |
| | ③ | 0.839 | 1.375 | 1.034 | 1.323 | |

Table 24: Different Embedding ways for long-term forecasting.

| Embedding | | Electricity | | ETTh2 | | Exchange | |
|---|---|---|---|---|---|---|---|
| | | MSE | MAE | MSE | MAE | MSE | MAE |
| ① | | 0.214 | 0.318 | 0.347 | 0.391 | 0.185 | 0.320 |
| ② | | 0.209 | 0.304 | 0.333 | 0.386 | 0.171 | 0.312 |
| ③ | | **0.201** | **0.296** | **0.330** | **0.381** | **0.164** | **0.296** |
| Embedding | | ETTm1 | | Traffic | | Weather | |
| | | MSE | MAE | MSE | MAE | MSE | MAE |
| ① | | 0.357 | 0.392 | 0.440 | 0.349 | 0.271 | 0.292 |
| ② | | 0.332 | 0.383 | 0.427 | 0.320 | 0.262 | 0.271 |
| ③ | | **0.336** | **0.385** | **0.423** | **0.313** | **0.257** | **0.269** |

### E.3 TRAINING SPEED AND MEMORY

Considering that the hyperparameters set during different training processes and the look-back window can affect the number of model parameters and running speed, to make a fairer comparison of the parameters and training speed of different models, we selected three typical models: PatchTST (Transformer-based), Dlinear (MLP-based), and ModernTCN (CNN-based), as well as the model proposed in this paper for comparison. We unified the training hyperparameters as shown in Table 11. The results for different input lengths on ETTm2 (prediction length fixed at 96) are shown in Figure 15. From the figure, it can be observed that in terms of running speed, as the input length increases, the training time of PatchTST increases, and the memory usage of ModernTCN increases significantly. In contrast, the training time and memory usage of the TVNet proposed in this paper increase slowly with the increase of input length, proving the superiority of the proposed model in terms of efficiency and effectiveness.

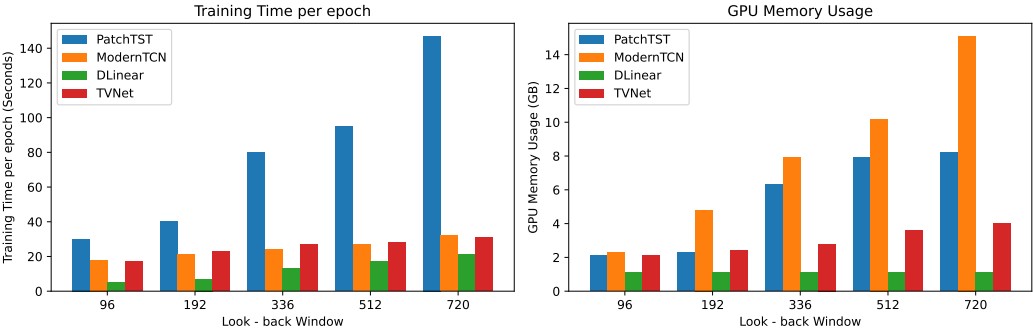

Figure 15: Params and GPU Memory for different input length(ETTm2)

# F MORE EXTRA EXPERIMENTS STUDIES

We have conducted additional experiments and analyses.

## F.1 INPUT LENGTH

**Impact of input length:** In time series forecasting, the input length dictates the extent of historical data available for the algorithm's analysis. Typically, models that excel at capturing long-term dependencies tend to exhibit superior performance with increasing input lengths. To substantiate this, we evaluated our model across a range of input lengths while maintaining constant prediction lengths. Figure 16 illustrates that the performance of transformer-based models diminishes with longer inputs due to the prevalence of repetitive short-term patterns. Conversely, TVNet's predictive accuracy consistently improves with extended input lengths, suggesting its proficiency in capturing long-term dependencies and extracting meaningful information effectively.

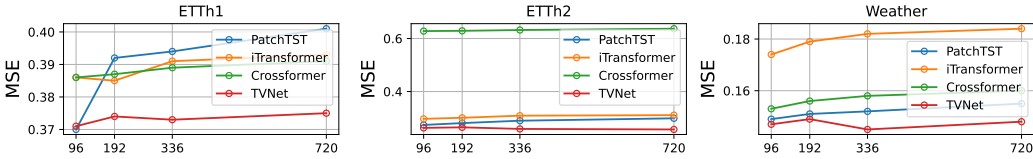

Figure 16: The MSE results with different input lengths and same prediction lengths(ETTh1,ETTh2 and Weather) prediction length($L$)=192

## F.2 MODEL ROBUSTNESS

We employ a straightforward noise injection technique to ascertain the robustness of our model. Specifically, we randomly select a proportion $\epsilon$ of data from the original input sequence and introduce random perturbations within the range $[-2X_i, 2X_i]$, where $X_i$ represents the original data values. The perturbed data is subsequently utilized for training, and the mean squared error (MSE) and mean absolute error (MAE) metrics are documented. The findings are presented in Table 25. An increase in the perturbation ratio $\epsilon$ leads to a marginal rise in both MSE and MAE, indicating that TVNet demonstrates commendable robustness against mild noise (up to 10%) and excels at handling significant data anomalies, such as those caused by equipment malfunctions in power data.

Table 25: Different $\epsilon$ indicates different proportions of noise injection.

| Models | Metrics | Weather | | | | Electricity | | | | Traffic | | | |
|---|---|---|---|---|---|---|---|---|---|---|---|---|---|
| | | 96 | 192 | 336 | 720 | 96 | 192 | 336 | 720 | 96 | 192 | 336 | 720 |
| TVNet | MSE | 0.147 | 0.194 | 0.235 | 0.308 | 0.142 | 0.165 | 0.164 | 0.190 | 0.367 | 0.381 | 0.395 | 0.442 |
| | MAE | 0.198 | 0.238 | 0.277 | 0.331 | 0.223 | 0.241 | 0.269 | 0.284 | 0.252 | 0.262 | 0.268 | 0.290 |
| $\epsilon = 1\%$ | MSE | 0.146 | 0.192 | 0.238 | 0.305 | 0.142 | 0.165 | 0.167 | 0.191 | 0.365 | 0.382 | 0.396 | 0.444 |
| | MAE | 0.197 | 0.236 | 0.280 | 0.332 | 0.225 | 0.244 | 0.271 | 0.286 | 0.255 | 0.260 | 0.267 | 0.292 |
| $\epsilon = 5\%$ | MSE | 0.150 | 0.196 | 0.238 | 0.307 | 0.148 | 0.170 | 0.166 | 0.191 | 0.369 | 0.380 | 0.399 | 0.445 |
| | MAE | 0.195 | 0.239 | 0.281 | 0.335 | 0.227 | 0.248 | 0.271 | 0.283 | 0.257 | 0.260 | 0.273 | 0.290 |
| $\epsilon = 10\%$ | MSE | 0.159 | 0.207 | 0.245 | 0.310 | 0.149 | 0.176 | 0.177 | 0.194 | 0.379 | 0.386 | 0.405 | 0.451 |
| | MAE | 0.205 | 0.247 | 0.289 | 0.336 | 0.229 | 0.24 5 | 0.279 | 0.283 | 0.265 | 0.271 | 0.278 | 0.294 |

## F.3 UNIVARIATE LONG-TERM FORECASTING RESULTS

Table 26 shows the Univariate long-term forecasting results.

Table 26: Following PatchTS(Nie et al., 2022) input length is fixed as 336 and prediction lengths are $T \in \{96, 192, 336, 720\}$. The best results are in **bold**.

| Models | Metrics | TVNet | | PatchTST | | DLinear | | FEDformer | | Autoformer | | Informer | | LogTrans | |
|---|---|---|---|---|---|---|---|---|---|---|---|---|---|---|---|
| | | MSE | MAE | MSE | MAE | MSE | MAE | MSE | MAE | MSE | MAE | MSE | MAE | MSE | MAE |
| ETTh1 | 96 | **0.053** | **0.179** | 0.055 | 0.179 | 0.056 | 0.180 | 0.079 | 0.215 | 0.071 | 0.206 | 0.193 | 0.377 | 0.283 | 0.468 |
| | 192 | **0.067** | **0.202** | 0.071 | 0.205 | 0.071 | 0.204 | 0.104 | 0.245 | 0.114 | 0.262 | 0.217 | 0.295 | 0.234 | 0.409 |
| | 336 | **0.071** | **0.210** | 0.076 | 0.220 | 0.098 | 0.244 | 0.119 | 0.270 | 0.107 | 0.258 | 0.202 | 0.381 | 0.286 | 0.546 |
| | 720 | **0.083** | **0.229** | 0.087 | 0.236 | 0.087 | 0.359 | 0.142 | 0.299 | 0.126 | 0.283 | 0.183 | 0.355 | 0.475 | 0.629 |
| ETTh2 | 96 | **0.121** | **0.268** | 0.129 | 0.282 | 0.131 | 0.279 | 0.128 | 0.271 | 0.153 | 0.306 | 0.213 | 0.373 | 0.217 | 0.379 |
| | 192 | **0.164** | **0.320** | 0.168 | 0.328 | 0.176 | 0.329 | 0.185 | 0.330 | 0.204 | 0.351 | 0.227 | 0.387 | 0.281 | 0.429 |
| | 336 | **0.171** | **0.333** | 0.171 | 0.336 | 0.209 | 0.367 | 0.231 | 0.378 | 0.246 | 0.389 | 0.242 | 0.408 | 0.293 | 0.437 |
| | 720 | **0.219** | **0.380** | 0.223 | 0.380 | 0.276 | 0.426 | 0.278 | 0.420 | 0.268 | 0.409 | 0.291 | 0.439 | 0.218 | 0.387 |
| ETTm1 | 96 | **0.024** | **0.120** | 0.026 | 0.121 | 0.028 | 0.123 | 0.033 | 0.140 | 0.056 | 0.183 | 0.109 | 0.277 | 0.218 | 0.387 |
| | 192 | **0.038** | **0.150** | 0.039 | 0.150 | 0.045 | 0.156 | 0.058 | 0.186 | 0.081 | 0.216 | 0.151 | 0.310 | 0.157 | 0.317 |
| | 336 | **0.053** | **0.173** | 0.053 | 0.173 | 0.061 | 0.182 | 0.084 | 0.231 | 0.076 | 0.218 | 0.427 | 0.591 | 0.289 | 0.459 |
| | 720 | **0.072** | **0.203** | 0.073 | 0.206 | 0.080 | 0.210 | 0.102 | 0.250 | 0.110 | 0.267 | 0.438 | 0.586 | 0.430 | 0.579 |
| ETTm2 | 96 | **0.063** | **0.183** | 0.065 | 0.186 | 0.063 | 0.183 | 0.067 | 0.198 | 0.065 | 0.189 | 0.088 | 0.225 | 0.075 | 0.208 |
| | 192 | **0.089** | **0.227** | 0.093 | 0.231 | 0.092 | 0.227 | 0.102 | 0.245 | 0.118 | 0.256 | 0.132 | 0.283 | 0.129 | 0.275 |
| | 336 | **0.119** | **0.261** | 0.120 | 0.265 | 0.119 | 0.261 | 0.130 | 0.279 | 0.154 | 0.305 | 0.180 | 0.336 | 0.154 | 0.302 |
| | 720 | **0.170** | **0.322** | 0.171 | 0.322 | 0.175 | 0.320 | 0.178 | 0.325 | 0.182 | 0.335 | 0.300 | 0.435 | 0.160 | 0.321 |

## F.4 LLM AND MORE BASELINES

For the long-term forecasting task, we have added some new Baselines (LLM-Based and TMixer). Table 27 shows the results, from which we can see that TVNet has shown good performance in most of the forecasting results.

Table 27: The complete results for **long-term forecasting** are detailed(More baselines focused on LLM), demonstrating a thorough comparison among various competitive models across four distinct forecast horizons.The Avg metric indicates the average performance across these forecast horizons.

| Models | | TVNet (Ours) | | S2IP-LLM (2024) | | TimeLLM (2023) | | GPT4TS (2024) | | TimeMixer (2024b) | |
|---|---|---|---|---|---|---|---|---|---|---|---|
| Metrics | | MSE | MAE | MSE | MAE | MSE | MAE | MSE | MAE | MSE | MAE |
| ETTm1 | 96 | 0.288 | 0.343 | 0.288 | 0.346 | 0.311 | 0.365 | 0.292 | 0.346 | 0.320 | 0.357 |
| | 192 | 0.326 | 0.367 | 0.323 | 0.365 | 0.364 | 0.395 | 0.332 | 0.372 | 0.361 | 0.381 |
| | 336 | 0.365 | 0.391 | 0.359 | 0.390 | 0.369 | 0.398 | 0.366 | 0.394 | 0.390 | 0.404 |
| | 720 | 0.412 | 0.413 | 0.403 | 0.418 | 0.416 | 0.425 | 0.417 | 0.421 | 0.454 | 0.441 |
| | Avg | 0.348 | 0.379 | 0.343 | 0.379 | 0.365 | 0.395 | 0.388 | 0.403 | 0.381 | 0.395 |
| ETTm2 | 96 | 0.161 | 0.254 | 0.165 | 0.257 | 0.170 | 0.262 | 0.173 | 0.258 | 0.175 | 0.258 |
| | 192 | 0.220 | 0.293 | 0.222 | 0.299 | 0.229 | 0.303 | 0.229 | 0.301 | 0.237 | 0.299 |
| | 336 | 0.272 | 0.316 | 0.277 | 0.330 | 0.281 | 0.335 | 0.286 | 0.341 | 0.298 | 0.340 |
| | 720 | 0.349 | 0.379 | 0.363 | 0.390 | 0.379 | 0.403 | 0.378 | 0.401 | 0.391 | 0.396 |
| | Avg | 0.251 | 0.311 | 0.257 | 0.319 | 0.264 | 0.325 | 0.284 | 0.339 | 0.275 | 0.323 |
| ETTh1 | 96 | 0.371 | 0.408 | 0.366 | 0.396 | 0.380 | 0.406 | 0.376 | 0.397 | 0.375 | 0.400 |
| | 192 | 0.398 | 0.409 | 0.401 | 0.420 | 0.426 | 0.438 | 0.416 | 0.418 | 0.429 | 0.421 |
| | 336 | 0.401 | 0.409 | 0.412 | 0.431 | 0.437 | 0.451 | 0.442 | 0.433 | 0.484 | 0.458 |
| | 720 | 0.458 | 0.459 | 0.440 | 0.458 | 0.515 | 0.509 | 0.477 | 0.456 | 0.498 | 0.482 |
| | Avg | 0.407 | 0.421 | 0.406 | 0.427 | 0.439 | 0.451 | 0.465 | 0.455 | 0.447 | 0.440 |
| ETTh2 | 96 | 0.263 | 0.329 | 0.278 | 0.340 | 0.306 | 0.362 | 0.285 | 0.342 | 0.289 | 0.341 |
| | 192 | 0.319 | 0.372 | 0.346 | 0.385 | 0.346 | 0.385 | 0.354 | 0.389 | 0.372 | 0.392 |
| | 336 | 0.311 | 0.373 | 0.367 | 0.406 | 0.393 | 0.422 | 0.373 | 0.407 | 0.386 | 0.414 |
| | 720 | 0.401 | 0.434 | 0.400 | 0.436 | 0.397 | 0.433 | 0.406 | 0.441 | 0.412 | 0.434 |
| | Avg | 0.324 | 0.377 | 0.347 | 0.391 | 0.360 | 0.400 | 0.381 | 0.412 | 0.364 | 0.395 |
| Electricity | 96 | 0.142 | 0.223 | 0.135 | 0.230 | 0.140 | 0.246 | 0.139 | 0.238 | 0.153 | 0.247 |
| | 192 | 0.165 | 0.241 | 0.149 | 0.247 | 0.155 | 0.253 | 0.153 | 0.251 | 0.166 | 0.256 |
| | 336 | 0.164 | 0.269 | 0.167 | 0.266 | 0.175 | 0.279 | 0.169 | 0.266 | 0.185 | 0.277 |
| | 720 | 0.190 | 0.284 | 0.200 | 0.287 | 0.204 | 0.305 | 0.206 | 0.297 | 0.225 | 0.310 |
| | Avg | 0.165 | 0.254 | 0.161 | 0.257 | 0.168 | 0.270 | 0.167 | 0.263 | 0.182 | 0.272 |
| Weather | 96 | 0.147 | 0.198 | 0.145 | 0.195 | 0.148 | 0.197 | 0.162 | 0.212 | 0.163 | 0.209 |
| | 192 | 0.194 | 0.238 | 0.190 | 0.235 | 0.194 | 0.246 | 0.204 | 0.248 | 0.208 | 0.250 |
| | 336 | 0.235 | 0.277 | 0.243 | 0.280 | 0.248 | 0.285 | 0.254 | 0.286 | 0.251 | 0.287 |
| | 720 | 0.308 | 0.331 | 0.312 | 0.326 | 0.317 | 0.332 | 0.326 | 0.337 | 0.339 | 0.341 |
| | Avg | 0.221 | 0.261 | 0.222 | 0.259 | 0.226 | 0.265 | 0.237 | 0.270 | 0.240 | 0.271 |
| Traffic | 96 | 0.367 | 0.252 | 0.379 | 0.274 | 0.383 | 0.280 | 0.388 | 0.282 | 0.462 | 0.285 |
| | 192 | 0.381 | 0.262 | 0.397 | 0.282 | 0.399 | 0.294 | 0.407 | 0.290 | 0.473 | 0.296 |
| | 336 | 0.395 | 0.268 | 0.407 | 0.289 | 0.411 | 0.306 | 0.412 | 0.294 | 0.498 | 0.296 |
| | 720 | 0.442 | 0.290 | 0.440 | 0.301 | 0.448 | 0.319 | 0.450 | 0.312 | 0.506 | 0.313 |
| | Avg | 0.396 | 0.268 | 0.405 | 0.286 | 0.440 | 0.301 | 0.414 | 0.294 | 0.484 | 0.297 |

## F.5    EXOGENOUS VARIABLE FORECASTING

As discussed in Timexer (Wang et al., 2024b), forecasting exogenous variables is crucial in various domains, including load forecasting. Our experiments on the ETT dataset also focused on predicting such variables. Figures 17 demonstrate that TVNet consistently outperforms the optimal baseline model in exogenous variable forecasting.

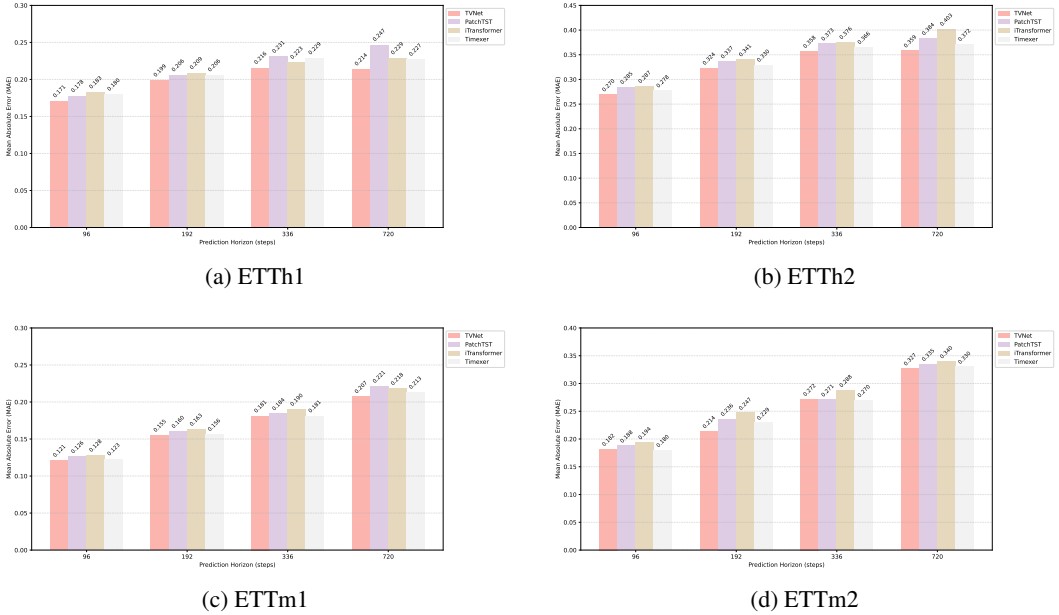

(a) ETTh1

(b) ETTh2

(c) ETTm1

(d) ETTm2

Figure 17: Exogenous variable learning results for ETT dataset.

# G   FULL RESULTS

## G.1   LONG-TERM FORECASTING

Table 28: The complete results for **long-term forecasting** are detailed, demonstrating a thorough comparison among various competitive models across four distinct forecast horizons.The Avg metric indicates the average performance across these forecast horizons.

| Models | | TVNet (Ours) | | PatchTST (2022) | | iTransformer (2023) | | Crossformer (2023) | | RLinear (2023a) | | MTS-Mixer (2023b) | | DLinear (2023b) | | TimesNet (2022a) | | MICN (2023) | | ModernTCN (2024) | | FEDformer (2022) | | RMLP (2023a) | |
|---|---|---|---|---|---|---|---|---|---|---|---|---|---|---|---|---|---|---|---|---|---|---|---|---|---|
| Metrics | | MSE | MAE | MSE | MAE | MSE | MAE | MSE | MAE | MSE | MAE | MSE | MAE | MSE | MAE | MSE | MAE | MSE | MAE | MSE | MAE | MSE | MAE | MSE | MAE |
| ETTm1 | 96 | **0.288** | **0.343** | 0.290 | 0.342 | 0.334 | 0.368 | 0.316 | 0.373 | 0.301 | 0.342 | 0.314 | 0.358 | 0.299 | 0.343 | 0.338 | 0.375 | 0.314 | 0.360 | 0.292 | 0.346 | 0.326 | 0.390 | 0.298 | 0.345 |
| | 192 | **0.326** | **0.367** | 0.332 | 0.369 | 0.377 | 0.391 | 0.377 | 0.411 | 0.355 | 0.363 | 0.354 | 0.386 | 0.335 | 0.365 | 0.371 | 0.387 | 0.359 | 0.387 | 0.332 | 0.368 | 0.365 | 0.415 | 0.344 | 0.375 |
| | 336 | **0.365** | **0.391** | 0.366 | 0.392 | 0.426 | 0.420 | 0.431 | 0.442 | 0.370 | 0.383 | 0.384 | 0.405 | 0.369 | 0.386 | 0.410 | 0.411 | 0.398 | 0.413 | 0.365 | 0.391 | 0.392 | 0.425 | 0.390 | 0.410 |
| | 720 | **0.412** | **0.413** | 0.416 | 0.420 | 0.491 | 0.459 | 0.600 | 0.547 | 0.425 | 0.414 | 0.427 | 0.432 | 0.425 | 0.421 | 0.478 | 0.450 | 0.459 | 0.464 | 0.416 | 0.417 | 0.446 | 0.458 | 0.445 | 0.441 |
| | Avg | **0.348** | **0.379** | 0.351 | 0.381 | 0.407 | 0.410 | 0.431 | 0.443 | 0.358 | 0.376 | 0.370 | 0.395 | 0.357 | 0.379 | 0.400 | 0.450 | 0.383 | 0.406 | 0.351 | 0.381 | 0.382 | 0.422 | 0.369 | 0.393 |
| ETTm2 | 96 | **0.161** | **0.254** | 0.165 | 0.255 | 0.180 | 0.264 | 0.421 | 0.461 | 0.164 | 0.253 | 0.177 | 0.259 | 0.167 | 0.260 | 0.187 | 0.267 | 0.178 | 0.273 | 0.166 | 0.256 | 0.180 | 0.271 | 0.174 | 0.259 |
| | 192 | 0.220 | 0.293 | 0.220 | 0.292 | 0.250 | 0.309 | 0.503 | 0.519 | **0.219** | **0.290** | 0.241 | 0.303 | 0.224 | 0.303 | 0.249 | 0.309 | 0.245 | 0.316 | 0.222 | 0.293 | 0.252 | 0.318 | 0.236 | 0.303 |
| | 336 | **0.272** | **0.316** | 0.274 | 0.329 | 0.311 | 0.348 | 0.611 | 0.580 | 0.273 | 0.326 | 0.297 | 0.338 | 0.281 | 0.342 | 0.312 | 0.351 | 0.295 | 0.350 | 0.272 | 0.324 | 0.324 | 0.364 | 0.291 | 0.338 |
| | 720 | 0.349 | 0.379 | 0.362 | 0.385 | 0.412 | 0.407 | 0.996 | 0.750 | 0.366 | 0.385 | 0.396 | 0.398 | 0.397 | 0.421 | 0.497 | 0.403 | 0.389 | 0.406 | 0.351 | 0.381 | 0.410 | 0.420 | 0.371 | 0.391 |
| | Avg | **0.251** | **0.311** | 0.255 | 0.315 | 0.288 | 0.332 | 0.632 | 0.578 | 0.256 | 0.314 | 0.277 | 0.325 | 0.267 | 0.332 | 0.291 | 0.333 | 0.277 | 0.336 | 0.253 | 0.314 | 0.292 | 0.343 | 0.268 | 0.322 |
| ETTh1 | 96 | 0.371 | 0.408 | 0.370 | 0.399 | 0.386 | 0.405 | 0.386 | 0.429 | 0.366 | 0.391 | 0.372 | 0.395 | 0.375 | 0.399 | 0.384 | 0.402 | 0.396 | 0.427 | 0.368 | 0.394 | 0.376 | 0.415 | 0.390 | 0.410 |
| | 192 | **0.398** | **0.409** | 0.413 | 0.421 | 0.441 | 0.436 | 0.419 | 0.444 | 0.404 | 0.412 | 0.416 | 0.426 | 0.405 | 0.416 | 0.557 | 0.436 | 0.430 | 0.453 | 0.405 | 0.413 | 0.423 | 0.446 | 0.430 | 0.432 |
| | 336 | 0.401 | 0.409 | 0.422 | 0.436 | 0.487 | 0.458 | 0.440 | 0.461 | 0.420 | 0.423 | 0.455 | 0.449 | 0.439 | 0.443 | 0.491 | 0.469 | 0.433 | 0.458 | 0.391 | 0.412 | 0.444 | 0.462 | 0.441 | 0.441 |
| | 720 | 0.458 | 0.459 | 0.447 | 0.466 | 0.503 | 0.491 | 0.519 | 0.524 | 0.442 | 0.456 | 0.475 | 0.472 | 0.472 | 0.490 | 0.521 | 0.500 | 0.474 | 0.508 | 0.450 | 0.461 | 0.469 | 0.492 | 0.506 | 0.495 |
| | Avg | 0.407 | 0.421 | 0.413 | 0.431 | 0.454 | 0.447 | 0.441 | 0.465 | 0.408 | 0.421 | 0.430 | 0.436 | 0.423 | 0.437 | 0.458 | 0.450 | 0.433 | 0.462 | 0.404 | 0.420 | 0.428 | 0.454 | 0.442 | 0.445 |
| ETTh2 | 96 | 0.263 | 0.329 | 0.274 | 0.336 | 0.297 | 0.349 | 0.628 | 0.563 | **0.262** | 0.331 | 0.307 | 0.354 | 0.289 | 0.353 | 0.340 | 0.374 | 0.289 | 0.357 | 0.263 | 0.332 | 0.332 | 0.374 | 0.288 | 0.352 |
| | 192 | **0.319** | **0.372** | 0.339 | 0.379 | 0.380 | 0.400 | 0.703 | 0.624 | 0.320 | 0.374 | 0.374 | 0.399 | 0.383 | 0.418 | 0.402 | 0.414 | 0.409 | 0.438 | 0.320 | 0.374 | 0.407 | 0.446 | 0.343 | 0.387 |
| | 336 | **0.311** | **0.373** | 0.329 | 0.384 | 0.428 | 0.432 | 0.827 | 0.675 | 0.325 | 0.386 | 0.398 | 0.432 | 0.448 | 0.465 | 0.452 | 0.452 | 0.417 | 0.452 | 0.313 | 0.376 | 0.400 | 0.447 | 0.353 | 0.402 |
| | 720 | 0.401 | 0.434 | 0.379 | 0.422 | 0.427 | 0.445 | 1.181 | 0.840 | **0.372** | **0.421** | 0.463 | 0.465 | 0.605 | 0.551 | 0.462 | 0.468 | 0.426 | 0.473 | 0.392 | 0.433 | 0.412 | 0.469 | 0.410 | 0.440 |
| | Avg | 0.324 | 0.377 | 0.330 | 0.379 | 0.383 | 0.407 | 0.835 | 0.676 | **0.320** | **0.378** | 0.386 | 0.413 | 0.431 | 0.447 | 0.414 | 0.427 | 0.385 | 0.430 | 0.322 | 0.379 | 0.388 | 0.434 | 0.349 | 0.395 |
| Electricity | 96 | 0.142 | 0.223 | **0.129** | **0.222** | 0.148 | 0.240 | 0.187 | 0.283 | 0.140 | 0.235 | 0.141 | 0.243 | 0.153 | 0.237 | 0.168 | 0.272 | 0.159 | 0.267 | 0.129 | 0.226 | 0.186 | 0.302 | 0.129 | 0.224 |
| | 192 | 0.165 | 0.241 | 0.147 | 0.240 | 0.162 | 0.253 | 0.258 | 0.330 | 0.154 | 0.248 | 0.163 | 0.261 | 0.152 | 0.249 | 0.184 | 0.289 | 0.168 | 0.279 | **0.143** | **0.239** | 0.197 | 0.311 | 0.147 | 0.240 |
| | 336 | 0.164 | 0.269 | 0.163 | 0.259 | 0.178 | 0.269 | 0.323 | 0.369 | 0.171 | 0.264 | 0.176 | 0.277 | 0.169 | 0.267 | 0.198 | 0.300 | 0.196 | 0.308 | **0.161** | **0.259** | 0.213 | 0.328 | 0.164 | 0.257 |
| | 720 | **0.190** | **0.284** | 0.197 | 0.290 | 0.225 | 0.317 | 0.404 | 0.423 | 0.209 | 0.297 | 0.212 | 0.308 | 0.233 | 0.344 | 0.220 | 0.320 | 0.203 | 0.312 | 0.191 | 0.286 | 0.233 | 0.344 | 0.203 | 0.291 |
| | Avg | 0.165 | 0.254 | 0.159 | 0.253 | 0.178 | 0.270 | 0.293 | 0.351 | 0.169 | 0.261 | 0.173 | 0.272 | 0.177 | 0.274 | 0.192 | 0.295 | 0.182 | 0.292 | **0.156** | **0.253** | 0.207 | 0.321 | 0.161 | **0.253** |
| Weather | 96 | **0.147** | **0.198** | 0.149 | 0.198 | 0.174 | 0.214 | 0.153 | 0.217 | 0.175 | 0.225 | 0.156 | 0.206 | 0.152 | 0.237 | 0.172 | 0.220 | 0.161 | 0.226 | 0.149 | 0.200 | 0.238 | 0.314 | 0.149 | 0.202 |
| | 192 | **0.194** | **0.238** | 0.194 | 0.241 | 0.221 | 0.254 | 0.197 | 0.269 | 0.218 | 0.260 | 0.199 | 0.248 | 0.220 | 0.282 | 0.219 | 0.261 | 0.220 | 0.283 | 0.196 | 0.245 | 0.275 | 0.329 | 0.194 | 0.242 |
| | 336 | **0.235** | **0.277** | 0.245 | 0.282 | 0.278 | 0.296 | 0.252 | 0.311 | 0.265 | 0.294 | 0.249 | 0.291 | 0.265 | 0.319 | 0.280 | 0.306 | 0.275 | 0.328 | 0.238 | 0.277 | 0.339 | 0.377 | 0.243 | 0.282 |
| | 720 | **0.308** | **0.331** | 0.314 | 0.334 | 0.358 | 0.347 | 0.318 | 0.363 | 0.329 | 0.339 | 0.336 | 0.343 | 0.323 | 0.362 | 0.365 | 0.359 | 0.311 | 0.356 | 0.314 | 0.334 | 0.389 | 0.409 | 0.316 | 0.333 |
| | Avg | **0.221** | **0.261** | 0.226 | 0.264 | 0.258 | 0.278 | 0.230 | 0.290 | 0.247 | 0.279 | 0.235 | 0.272 | 0.240 | 0.300 | 0.259 | 0.287 | 0.242 | 0.298 | 0.224 | 0.264 | 0.310 | 0.357 | 0.225 | 0.265 |
| Traffic | 96 | 0.367 | 0.252 | **0.360** | **0.249** | 0.395 | 0.268 | 0.512 | 0.290 | 0.496 | 0.375 | 0.462 | 0.332 | 0.410 | 0.282 | 0.593 | 0.321 | 0.508 | 0.301 | 0.368 | 0.253 | 0.576 | 0.359 | 0.430 | 0.327 |
| | 192 | 0.381 | 0.262 | **0.379** | **0.256** | 0.417 | 0.276 | 0.523 | 0.297 | 0.503 | 0.377 | 0.488 | 0.354 | 0.423 | 0.287 | 0.617 | 0.336 | 0.536 | 0.315 | 0.379 | 0.261 | 0.610 | 0.380 | 0.451 | 0.340 |
| | 336 | 0.395 | 0.268 | **0.392** | **0.264** | 0.433 | 0.283 | 0.530 | 0.300 | 0.517 | 0.382 | 0.498 | 0.360 | 0.436 | 0.296 | 0.629 | 0.336 | 0.525 | 0.310 | 0.397 | 0.270 | 0.608 | 0.375 | 0.470 | 0.351 |
| | 720 | 0.442 | 0.290 | **0.432** | **0.286** | 0.467 | 0.302 | 0.573 | 0.313 | 0.555 | 0.398 | 0.529 | 0.370 | 0.466 | 0.315 | 0.640 | 0.350 | 0.571 | 0.323 | 0.440 | 0.296 | 0.621 | 0.375 | 0.513 | 0.372 |
| | Avg | 0.396 | 0.268 | **0.391** | **0.264** | 0.428 | 0.282 | 0.535 | 0.300 | 0.518 | 0.383 | 0.494 | 0.354 | 0.434 | 0.295 | 0.620 | 0.336 | 0.535 | 0.312 | 0.396 | 0.270 | 0.604 | 0.372 | 0.466 | 0.348 |
| Exchange | 96 | **0.080** | **0.195** | 0.093 | 0.214 | 0.086 | 0.206 | 0.186 | 0.346 | 0.083 | 0.301 | 0.083 | 0.020 | 0.081 | 0.203 | 0.107 | 0.234 | 0.102 | 0.235 | 0.080 | 0.196 | 0.139 | 0.276 | 0.083 | 0.201 |
| | 192 | 0.163 | **0.285** | 0.192 | 0.312 | 0.177 | 0.299 | 0.467 | 0.522 | 0.170 | 0.293 | 0.174 | 0.296 | **0.157** | 0.293 | 0.226 | 0.344 | 0.172 | 0.316 | 0.166 | 0.288 | 0.256 | 0.369 | 0.170 | 0.292 |
| | 336 | 0.291 | 0.394 | 0.350 | 0.432 | 0.331 | 0.417 | 0.783 | 0.721 | 0.309 | 0.401 | 0.336 | 0.417 | 0.305 | 0.414 | 0.367 | 0.448 | **0.272** | 0.407 | 0.307 | 0.398 | 0.426 | 0.464 | 0.309 | 0.401 |
| | 720 | 0.658 | 0.594 | 0.911 | 0.716 | 0.847 | 0.691 | 1.367 | 0.943 | 0.817 | 0.680 | 0.900 | 0.715 | **0.643** | 0.601 | 0.964 | 0.746 | 0.714 | 0.658 | 0.656 | **0.582** | 1.090 | 0.800 | 0.816 | 0.680 |
| | Avg | 0.298 | 0.367 | 0.387 | 0.419 | 0.360 | 0.403 | 0.701 | 0.633 | 0.345 | 0.394 | 0.373 | 0.407 | **0.297** | 0.378 | 0.416 | 0.443 | 0.315 | 0.404 | 0.302 | **0.366** | 0.478 | 0.478 | 0.345 | 0.394 |
| ILI | 24 | 1.324 | **0.712** | **1.319** | 0.754 | 2.207 | 1.032 | 3.040 | 1.186 | 4.337 | 1.507 | 1.472 | 0.798 | 2.215 | 1.081 | 2.317 | 0.934 | 2.684 | 1.112 | 1.347 | 0.717 | 2.624 | 1.095 | 4.445 | 1.536 |
| | 36 | **1.190** | **0.772** | 1.430 | 0.834 | 1.934 | 0.951 | 3.356 | 1.230 | 4.205 | 1.481 | 1.435 | 0.745 | 1.963 | 0.963 | 1.972 | 0.920 | 2.507 | 1.013 | 1.250 | 0.778 | 2.516 | 1.021 | 4.409 | 1.159 |
| | 48 | **1.456** | **0.782** | 1.553 | 0.815 | 2.127 | 1.004 | 3.441 | 1.223 | 4.257 | 1.484 | 1.474 | 0.822 | 2.130 | 1.024 | 2.238 | 0.940 | 2.423 | 1.012 | 1.388 | 0.781 | 2.505 | 1.041 | 4.388 | 1.507 |
| | 60 | 1.652 | 0.796 | **1.470** | **0.788** | 2.298 | 0.998 | 3.608 | 1.302 | 4.278 | 1.487 | 1.839 | 0.912 | 2.368 | 1.096 | 2.027 | 0.928 | 2.653 | 1.085 | 1.774 | 0.868 | 2.742 | 1.122 | 4.306 | 1.502 |
| | Avg | **1.406** | **0.766** | 1.443 | 0.798 | 2.141 | 0.996 | 3.361 | 1.235 | 4.269 | 1.490 | 1.555 | 0.819 | 2.169 | 1.041 | 2.139 | 0.931 | 2.567 | 1.055 | 1.440 | 0.786 | 2.597 | 1.070 | 4.387 | 1.516 |
| 1st count | | **21** | **24** | 9 | 9 | 0 | 0 | 0 | 0 | 4 | 3 | 0 | 0 | 3 | 0 | 0 | 0 | 1 | 0 | 11 | 12 | 0 | 0 | 0 | 1 |

## G.2 Short-term Forecasting

Table 29: The complete results for the **short-term forecasting** task in the M4 dataset are presented, utilizing the Weighted Average (WA) as the metric.

| | Models | TVNet (Ours) | PatchTST (2022) | U-Mixer (2024) | Crossformer (2023) | RLinear (2023a) | MTS-Mixer (2023b) | DLinear (2023) | TimesNet (2022a) | MICN 2023 | ModernTCN (2024) | FEDformer (2022) | N-HiTS (2023) |
|---|---|---|---|---|---|---|---|---|---|---|---|---|---|
| Yearly | SMAPE | 13.217 | 13.258 | 13.317 | 13.392 | 13.944 | 13.548 | 16.965 | 13.387 | 14.935 | 13.226 | 13.728 | 13.418 |
| | MASE | 2.899 | 2.985 | 3.006 | 3.001 | 3.015 | 3.091 | 4.283 | 2.996 | 3.523 | 2.957 | 3.048 | 3.045 |
| | OWA | 0.768 | 0.781 | 0.786 | 0.787 | 0.807 | 0.803 | 1.058 | 0.786 | 0.900 | 0.777 | 0.803 | 0.793 |
| Quarterly | SMAPE | 9.986 | 10.197 | 9.956 | 16.317 | 10.702 | 10.128 | 12.145 | 10.100 | 11.452 | 9.971 | 10.792 | 10.202 |
| | MASE | 1.159 | 0.803 | 1.156 | 2.197 | 1.299 | 1.196 | 1.520 | 1.182 | 1.389 | 1.167 | 1.283 | 1.194 |
| | OWA | 0.876 | 0.803 | 0.873 | 1.542 | 0.959 | 0.896 | 1.106 | 0.890 | 1.026 | 0.878 | 0.958 | 0.899 |
| Monthly | SMAPE | 12.493 | 12.641 | 13.057 | 12.924 | 13.363 | 12.717 | 13.514 | 12.670 | 13.773 | 12.556 | 14.260 | 12.791 |
| | MASE | 0.921 | 0.930 | 1.067 | 0.966 | 1.014 | 0.931 | 1.037 | 0.933 | 1.076 | 0.917 | 1.102 | 0.969 |
| | OWA | 0.866 | 0.876 | 0.976 | 0.902 | 0.940 | 0.879 | 0.956 | 0.878 | 0.983 | 0.866 | 1.012 | 0.899 |
| Others | SMAPE | 4.764 | 4.946 | 4.858 | 5.493 | 5.437 | 4.817 | 6.709 | 4.891 | 6.716 | 4.715 | 4.954 | 5.061 |
| | MASE | 2.986 | 2.985 | 3.195 | 3.690 | 3.706 | 3.255 | 4.953 | 3.302 | 4.717 | 3.107 | 3.264 | 3.216 |
| | OWA | 0.969 | 1.044 | 1.015 | 1.160 | 1.157 | 1.02 | 1.487 | 1.035 | 1.451 | 0.986 | 1.036 | 1.040 |
| WA | SMAPE | 11.671 | 11.807 | 11.740 | 13.474 | 12.473 | 11.892 | 13.639 | 11.829 | 13.130 | 11.698 | 12.840 | 11.927 |
| | MASE | 1.536 | 1.590 | 1.575 | 1.866 | 1.677 | 1.608 | 2.095 | 1.585 | 1.896 | 11.556 | 1.701 | 1.613 |
| | OWA | 0.832 | 0.851 | 0.845 | 0.985 | 0.898 | 0.859 | 1.051 | 0.851 | 0.980 | 0.838 | 0.918 | 0.861 |

## G.3 Imputation

Table 30: The complete results for the **imputation task** are as follows: To assess model performance under varying degrees of missing data, we randomly masked time points at rates of 12.5%, 25%, 37.5%, and 50%.

| | Models | TVNet (Ours) | | PatchTST (2022) | | SCINet (2022a) | | Crossformer (2023) | | RLinear (2023a) | | MTS-Mixer (2023b) | | DLinear (2023) | | TimesNet (2022a) | | MICN 2023 | | ModernTCN (2024) | | FEDformer (2022) | | RMLP (2023a) | |
|---|---|---|---|---|---|---|---|---|---|---|---|---|---|---|---|---|---|---|---|---|---|---|---|---|---|
| | Metrics | MSE | MAE | MSE | MAE | MSE | MAE | MSE | MAE | MSE | MAE | MSE | MAE | MSE | MAE | MSE | MAE | MSE | MAE | MSE | MAE | MSE | MAE | MSE | MAE |
| ETTm1 | 12.5% | 0.014 | 0.078 | 0.041 | 0.128 | 0.031 | 0.116 | 0.037 | 0.137 | 0.047 | 0.137 | 0.043 | 0.134 | 0.058 | 0.162 | 0.019 | 0.092 | 0.039 | 0.137 | 0.015 | 0.082 | 0.035 | 0.135 | 0.049 | 0.139 |
| | 25% | 0.016 | 0.083 | 0.043 | 0.130 | 0.036 | 0.124 | 0.038 | 0.141 | 0.061 | 0.157 | 0.051 | 0.147 | 0.080 | 0.193 | 0.023 | 0.101 | 0.059 | 0.170 | 0.018 | 0.088 | 0.052 | 0.166 | 0.057 | 0.154 |
| | 37.5% | 0.019 | 0.090 | 0.044 | 0.133 | 0.041 | 0.134 | 0.041 | 0.142 | 0.077 | 0.175 | 0.060 | 0.160 | 0.103 | 0.219 | 0.029 | 0.111 | 0.080 | 0.199 | 0.021 | 0.095 | 0.069 | 0.191 | 0.067 | 0.168 |
| | 50% | 0.023 | 0.101 | 0.050 | 0.142 | 0.049 | 0.143 | 0.047 | 0.152 | 0.096 | 0.195 | 0.070 | 0.174 | 0.132 | 0.248 | 0.036 | 0.124 | 0.103 | 0.221 | 0.026 | 0.105 | 0.089 | 0.218 | 0.079 | 0.183 |
| | Avg | 0.018 | 0.088 | 0.045 | 0.133 | 0.039 | 0.129 | 0.041 | 0.143 | 0.070 | 0.166 | 0.056 | 0.154 | 0.093 | 0.206 | 0.027 | 0.107 | 0.070 | 0.182 | 0.020 | 0.093 | 0.062 | 0.177 | 0.063 | 0.161 |
| ETTm2 | 12.5% | 0.018 | 0.081 | 0.025 | 0.092 | 0.023 | 0.093 | 0.044 | 0.148 | 0.026 | 0.093 | 0.026 | 0.096 | 0.062 | 0.166 | 0.018 | 0.080 | 0.060 | 0.165 | 0.017 | 0.076 | 0.056 | 0.159 | 0.026 | 0.096 |
| | 25% | 0.021 | 0.081 | 0.027 | 0.095 | 0.026 | 0.100 | 0.047 | 0.151 | 0.030 | 0.103 | 0.030 | 0.103 | 0.085 | 0.196 | 0.020 | 0.085 | 0.100 | 0.216 | 0.018 | 0.080 | 0.080 | 0.195 | 0.030 | 0.106 |
| | 37.5% | 0.025 | 0.089 | 0.029 | 0.099 | 0.028 | 0.105 | 0.044 | 0.145 | 0.034 | 0.113 | 0.033 | 0.110 | 0.106 | 0.222 | 0.023 | 0.091 | 0.163 | 0.273 | 0.020 | 0.084 | 0.110 | 0.231 | 0.034 | 0.113 |
| | 50% | 0.024 | 0.091 | 0.032 | 0.106 | 0.031 | 0.111 | 0.047 | 0.150 | 0.039 | 0.123 | 0.037 | 0.118 | 0.131 | 0.247 | 0.026 | 0.098 | 0.254 | 0.342 | 0.022 | 0.090 | 0.156 | 0.276 | 0.039 | 0.121 |
| | Avg | 0.022 | 0.086 | 0.028 | 0.098 | 0.027 | 0.102 | 0.046 | 0.149 | 0.032 | 0.108 | 0.032 | 0.107 | 0.096 | 0.208 | 0.022 | 0.088 | 0.144 | 0.249 | 0.019 | 0.082 | 0.101 | 0.215 | 0.032 | 0.109 |
| ETTh1 | 12.5% | 0.032 | 0.119 | 0.094 | 0.199 | 0.089 | 0.202 | 0.099 | 0.218 | 0.098 | 0.206 | 0.097 | 0.209 | 0.151 | 0.267 | 0.057 | 0.159 | 0.072 | 0.192 | 0.035 | 0.128 | 0.070 | 0.190 | 0.096 | 0.205 |
| | 25% | 0.039 | 0.137 | 0.119 | 0.225 | 0.099 | 0.211 | 0.125 | 0.243 | 0.123 | 0.229 | 0.115 | 0.226 | 0.180 | 0.292 | 0.069 | 0.178 | 0.105 | 0.232 | 0.042 | 0.140 | 0.106 | 0.236 | 0.120 | 0.228 |
| | 37.5% | 0.051 | 0.156 | 0.145 | 0.248 | 0.107 | 0.218 | 0.146 | 0.263 | 0.153 | 0.253 | 0.135 | 0.244 | 0.215 | 0.318 | 0.084 | 0.196 | 0.139 | 0.267 | 0.054 | 0.157 | 0.124 | 0.258 | 0.145 | 0.250 |
| | 50% | 0.064 | 0.168 | 0.173 | 0.271 | 0.120 | 0.231 | 0.158 | 0.281 | 0.188 | 0.278 | 0.160 | 0.263 | 0.257 | 0.347 | 0.102 | 0.215 | 0.185 | 0.310 | 0.067 | 0.174 | 0.165 | 0.299 | 0.176 | 0.274 |
| | Avg | 0.046 | 0.145 | 0.133 | 0.236 | 0.104 | 0.216 | 0.132 | 0.251 | 0.141 | 0.242 | 0.127 | 0.236 | 0.201 | 0.306 | 0.078 | 0.187 | 0.125 | 0.250 | 0.050 | 0.150 | 0.117 | 0.246 | 0.134 | 0.239 |
| ETTh2 | 12.5% | 0.036 | 0.119 | 0.057 | 0.150 | 0.061 | 0.161 | 0.103 | 0.220 | 0.057 | 0.152 | 0.061 | 0.157 | 0.100 | 0.216 | 0.040 | 0.130 | 0.106 | 0.223 | 0.037 | 0.121 | 0.095 | 0.212 | 0.058 | 0.153 |
| | 25% | 0.038 | 0.123 | 0.062 | 0.158 | 0.062 | 0.162 | 0.110 | 0.229 | 0.062 | 0.160 | 0.065 | 0.163 | 0.127 | 0.247 | 0.046 | 0.141 | 0.151 | 0.271 | 0.040 | 0.127 | 0.137 | 0.258 | 0.064 | 0.161 |
| | 37.5% | 0.040 | 0.126 | 0.068 | 0.168 | 0.065 | 0.166 | 0.129 | 0.246 | 0.068 | 0.168 | 0.070 | 0.171 | 0.158 | 0.276 | 0.052 | 0.151 | 0.229 | 0.332 | 0.043 | 0.134 | 0.187 | 0.304 | 0.070 | 0.171 |
| | 50% | 0.044 | 0.143 | 0.076 | 0.179 | 0.069 | 0.172 | 0.148 | 0.265 | 0.076 | 0.179 | 0.078 | 0.181 | 0.183 | 0.299 | 0.060 | 0.162 | 0.334 | 0.403 | 0.048 | 0.143 | 0.232 | 0.341 | 0.078 | 0.181 |
| | Avg | 0.039 | 0.127 | 0.066 | 0.164 | 0.064 | 0.165 | 0.122 | 0.240 | 0.066 | 0.165 | 0.069 | 0.168 | 0.142 | 0.259 | 0.049 | 0.146 | 0.205 | 0.307 | 0.042 | 0.131 | 0.163 | 0.279 | 0.068 | 0.166 |
| Electricity | 12.5% | 0.065 | 0.180 | 0.073 | 0.188 | 0.073 | 0.185 | 0.068 | 0.181 | 0.079 | 0.199 | 0.069 | 0.182 | 0.092 | 0.214 | 0.085 | 0.202 | 0.090 | 0.216 | 0.059 | 0.171 | 0.107 | 0.237 | 0.073 | 0.188 |
| | 25% | 0.075 | 0.192 | 0.082 | 0.200 | 0.081 | 0.198 | 0.079 | 0.198 | 0.105 | 0.233 | 0.083 | 0.202 | 0.118 | 0.247 | 0.089 | 0.206 | 0.108 | 0.236 | 0.071 | 0.188 | 0.120 | 0.251 | 0.090 | 0.211 |
| | 37.5% | 0.085 | 0.196 | 0.097 | 0.217 | 0.090 | 0.207 | 0.087 | 0.203 | 0.131 | 0.262 | 0.097 | 0.218 | 0.144 | 0.276 | 0.094 | 0.213 | 0.128 | 0.257 | 0.077 | 0.190 | 0.136 | 0.266 | 0.107 | 0.231 |
| | 50% | 0.091 | 0.210 | 0.110 | 0.232 | 0.099 | 0.214 | 0.096 | 0.212 | 0.160 | 0.291 | 0.108 | 0.231 | 0.175 | 0.305 | 0.100 | 0.221 | 0.151 | 0.278 | 0.085 | 0.200 | 0.158 | 0.284 | 0.125 | 0.252 |
| | Avg | 0.079 | 0.194 | 0.091 | 0.209 | 0.086 | 0.201 | 0.083 | 0.199 | 0.119 | 0.246 | 0.089 | 0.208 | 0.132 | 0.260 | 0.092 | 0.210 | 0.119 | 0.247 | 0.073 | 0.187 | 0.130 | 0.259 | 0.099 | 0.221 |
| Weather | 12.5% | 0.021 | 0.037 | 0.029 | 0.049 | 0.028 | 0.047 | 0.036 | 0.092 | 0.029 | 0.048 | 0.033 | 0.052 | 0.039 | 0.084 | 0.025 | 0.045 | 0.036 | 0.088 | 0.023 | 0.038 | 0.041 | 0.107 | 0.030 | 0.051 |
| | 25% | 0.024 | 0.037 | 0.031 | 0.053 | 0.029 | 0.050 | 0.035 | 0.088 | 0.032 | 0.055 | 0.034 | 0.056 | 0.048 | 0.103 | 0.029 | 0.052 | 0.047 | 0.115 | 0.025 | 0.041 | 0.064 | 0.163 | 0.033 | 0.057 |
| | 37.5% | 0.024 | 0.040 | 0.034 | 0.058 | 0.031 | 0.055 | 0.035 | 0.088 | 0.036 | 0.062 | 0.037 | 0.060 | 0.057 | 0.117 | 0.031 | 0.057 | 0.062 | 0.141 | 0.027 | 0.046 | 0.107 | 0.229 | 0.036 | 0.062 |
| | 50% | 0.028 | 0.041 | 0.039 | 0.066 | 0.034 | 0.059 | 0.038 | 0.092 | 0.040 | 0.067 | 0.041 | 0.066 | 0.066 | 0.134 | 0.034 | 0.062 | 0.080 | 0.168 | 0.031 | 0.051 | 0.183 | 0.312 | 0.040 | 0.068 |
| | Avg | 0.024 | 0.039 | 0.033 | 0.057 | 0.031 | 0.053 | 0.036 | 0.090 | 0.034 | 0.058 | 0.036 | 0.058 | 0.052 | 0.110 | 0.030 | 0.054 | 0.056 | 0.128 | 0.027 | 0.044 | 0.099 | 0.203 | 0.035 | 0.060 |

## G.4 CLASSIFICATION

Table 31: Performance comparison of various models on different datasets with accuracy metrics for **classification**.

| Datasets / Models | RNN-based | | | Convolution-based | | | | MLP-based | | | Transformer-based | | | | **TVNet** |
|---|---|---|---|---|---|---|---|---|---|---|---|---|---|---|---|
| | LSTNet (2018) | LSSL (2021) | Rocket (2020) | SCINet (2022a) | TimesNet 2022a | MICN (2023) | ModernTCN (2024) | DLinear (2023) | LightTS (2022) | MTS-Mixer (2023b) | FED. (2022) | PatchTST (2022) | Cross. (2023) | Flow. (2022b) | (Ours) |
| EthanolConcentration | 39.9 | 31.1 | 45.2 | 34.4 | 35.3 | 35.7 | 36.3 | 36.2 | 29.7 | 33.8 | 31.2 | 32.8 | 38.0 | 33.8 | 35.6 |
| FaceDetection | 65.7 | 66.7 | 64.7 | 68.9 | 65.2 | 68.6 | 70.8 | 68.0 | 67.5 | 70.2 | 66.0 | 68.3 | 68.7 | 67.6 | 71.2 |
| Handwriting | 25.8 | 24.6 | 58.8 | 23.6 | 25.5 | 32.1 | 30.6 | 27.0 | 26.1 | 26.0 | 28.0 | 29.6 | 28.8 | 33.8 | 32.7 |
| Heartbeat | 77.1 | 72.7 | 75.6 | 77.5 | 74.7 | 78.0 | 77.2 | 75.1 | 75.1 | 77.1 | 73.7 | 74.9 | 77.6 | 77.6 | 78.1 |
| JapaneseVowels | 98.1 | 98.4 | 96.2 | 96.0 | 94.6 | 98.4 | 98.8 | 96.2 | 96.2 | 94.3 | 98.4 | 97.5 | 99.1 | 98.9 | 98.9 |
| PEMS-SF | 86.7 | 86.1 | 75.1 | 83.8 | 85.5 | 89.6 | 89.1 | 75.1 | 88.4 | 80.9 | 80.9 | 89.3 | 85.9 | 86.0 | 88.9 |
| SelfRegulationSCP1 | 84.0 | 90.8 | 90.8 | 92.5 | 86.0 | 91.8 | 93.4 | 87.3 | 89.8 | 91.7 | 88.7 | 90.7 | 92.1 | 92.5 | 93.7 |
| SelfRegulationSCP2 | 52.8 | 52.2 | 53.3 | 57.2 | 53.6 | 57.2 | 60.3 | 50.5 | 51.1 | 55.0 | 54.4 | 57.8 | 58.3 | 56.1 | 60.5 |
| SpokenArabicDigits | 100.0 | 100.0 | 71.2 | 98.1 | 97.1 | 99.0 | 98.7 | 81.4 | 100.0 | 97.4 | 100.0 | 98/3 | 97.9 | 98.8 | 99.4 |
| UWaveGestureLibrary | 87.8 | 85.9 | 94.4 | 85.1 | 82.8 | 85.3 | 86.7 | 82.1 | 80.3 | 82.3 | 85.3 | 85.8 | 85.3 | 86.6 | 86.6 |
| Average Accuracy | 71.8 | 70.9 | 72.5 | 71.7 | 70.0 | 73.6 | 74.2 | 67.5 | 70.4 | 70.9 | 70.7 | 72.5 | 73.2 | 73.0 | **74.6** |

## G.5 ANOMALY DETECTION

Table 32: Performance comparison of various models on different datasets with Precision (P), Recall (R), and F1-score (F1) metrics for **anomaly detection**.

| Models | Datasets | SMD | | | MSL | | | SMAP | | | SWaT | | | PSM | | | Avg F1 |
|---|---|---|---|---|---|---|---|---|---|---|---|---|---|---|---|---|---|
| | Metrics | P | R | F1 | P | R | F1 | P | R | F1 | P | R | F1 | P | R | F1 | (%) |
| SCINet | (2022a) | 85.97 | 82.57 | 84.24 | 84.16 | 82.61 | 83.38 | 93.12 | 54.81 | 69.00 | 87.53 | 94.95 | 91.09 | 97.93 | 94.15 | 96.00 | 84.74 |
| MICN | (2023) | 88.45 | 83.47 | 85.89 | 83.02 | 83.67 | 83.34 | 90.65 | 61.42 | 73.23 | 91.87 | 95.08 | 93.45 | 98.40 | 88.69 | 93.29 | 85.84 |
| TimesNet | (2022a) | 88.66 | 83.14 | 85.81 | 83.92 | 86.32 | 85.15 | 92.52 | 58.29 | 71.52 | 86.76 | 97.32 | 91.74 | 98.19 | 96.76 | 97.47 | 86.34 |
| DLinear | (2023) | 83.62 | 71.52 | 77.10 | 84.34 | 84.52 | 84.88 | 92.32 | 55.41 | 69.26 | 80.91 | 95.30 | 87.52 | 98.28 | 89.26 | 93.55 | 82.46 |
| LightTS | (2022) | 87.10 | 78.42 | 82.53 | 82.40 | 75.78 | 78.95 | 92.58 | 55.27 | 69.21 | 91.98 | 94.72 | 93.33 | 98.37 | 95.97 | 97.15 | 84.23 |
| MTS-Mixer | (2023b) | 88.60 | 82.92 | 85.67 | 85.35 | 84.13 | 84.74 | 92.13 | 58.01 | 71.19 | 84.49 | 93.81 | 88.91 | 98.41 | 88.63 | 93.26 | 84.75 |
| RLLinear | (2023a) | 87.79 | 79.98 | 83.70 | 89.26 | 74.39 | 81.49 | 89.94 | 54.01 | 67.49 | 92.27 | 93.18 | 92.73 | 98.47 | 94.28 | 96.33 | 84.28 |
| RMLP | (2023a) | 87.35 | 78.10 | 82.46 | 86.67 | 65.30 | 74.48 | 90.62 | 52.22 | 66.26 | 92.32 | 93.20 | 92.76 | 98.01 | 93.25 | 95.57 | 82.31 |
| Reformer | (2021) | 82.58 | 69.24 | 75.32 | 85.51 | 83.31 | 84.40 | 90.91 | 57.44 | 70.40 | 72.50 | 96.53 | 82.80 | 59.93 | 95.38 | 73.61 | 77.31 |
| Informer | (2021) | 86.60 | 77.23 | 81.65 | 81.77 | 86.48 | 84.06 | 90.11 | 57.13 | 69.92 | 70.29 | 96.75 | 81.43 | 64.27 | 96.33 | 77.10 | 78.83 |
| Anomaly* | (2021) | 88.91 | 82.23 | 85.49 | 79.61 | 87.37 | 83.31 | 91.85 | 58.11 | 71.18 | 72.51 | 97.32 | 83.10 | 68.35 | 94.72 | 79.40 | 80.50 |
| Pyraformer | (2021) | 85.61 | 80.61 | 83.04 | 83.81 | 85.93 | 84.86 | 92.54 | 57.71 | 71.09 | 87.92 | 96.00 | 91.78 | 71/67 | 96.02 | 82.08 | 82.57 |
| Autoformer | (2021) | 88.06 | 82.35 | 85.11 | 77.27 | 80.92 | 79.05 | 90.40 | 58.62 | 71.12 | 89.85 | 95.81 | 92.74 | 99.08 | 88.15 | 93.29 | 84.26 |
| Stationary | (2022b) | 88.33 | 81.21 | 84.62 | 68.55 | 89.14 | 77.50 | 89.37 | 59.02 | 71.09 | 68.03 | 96.75 | 79.88 | 97.82 | 96.76 | 97.29 | 82.08 |
| FEDformer | (2022) | 87.95 | 82.39 | 85.08 | 77.14 | 80.07 | 78.57 | 90.47 | 58.10 | 70.76 | 90.17 | 96.42 | 93.17 | 97.31 | 97.16 | 97.23 | 84.97 |
| Crossformer | (2023) | 83.06 | 76.61 | 79.70 | 84.68 | 83.71 | 84.19 | 92.04 | 55.37 | 69.14 | 88.49 | 93.48 | 90.92 | 97.16 | 89.73 | 93.30 | 83.45 |
| PatchTST | (2022) | 87.42 | 81.65 | 84.44 | 84.07 | 86.23 | 85.14 | 92.43 | 57.51 | 70.91 | 80.70 | 94.43 | 87.24 | 98.87 | 93.99 | 96.37 | 84.82 |
| ModernTCN | (2024) | 87.86 | 83.85 | 85.81 | 83.94 | 85.93 | 84.92 | 95.17 | 57.69 | 71.26 | 91.83 | 95.98 | 93.86 | 98.09 | 96.38 | 97.23 | 86.62 |
| TVNet | (Ours) | 88.03 | 83.49 | 85.75 | 83.91 | 86.47 | 85.16 | 92.94 | 58.26 | 71.64 | 91.26 | 96.33 | 93.72 | 98.30 | 96.76 | 97.53 | **86.76** |

# H SHOWCASES

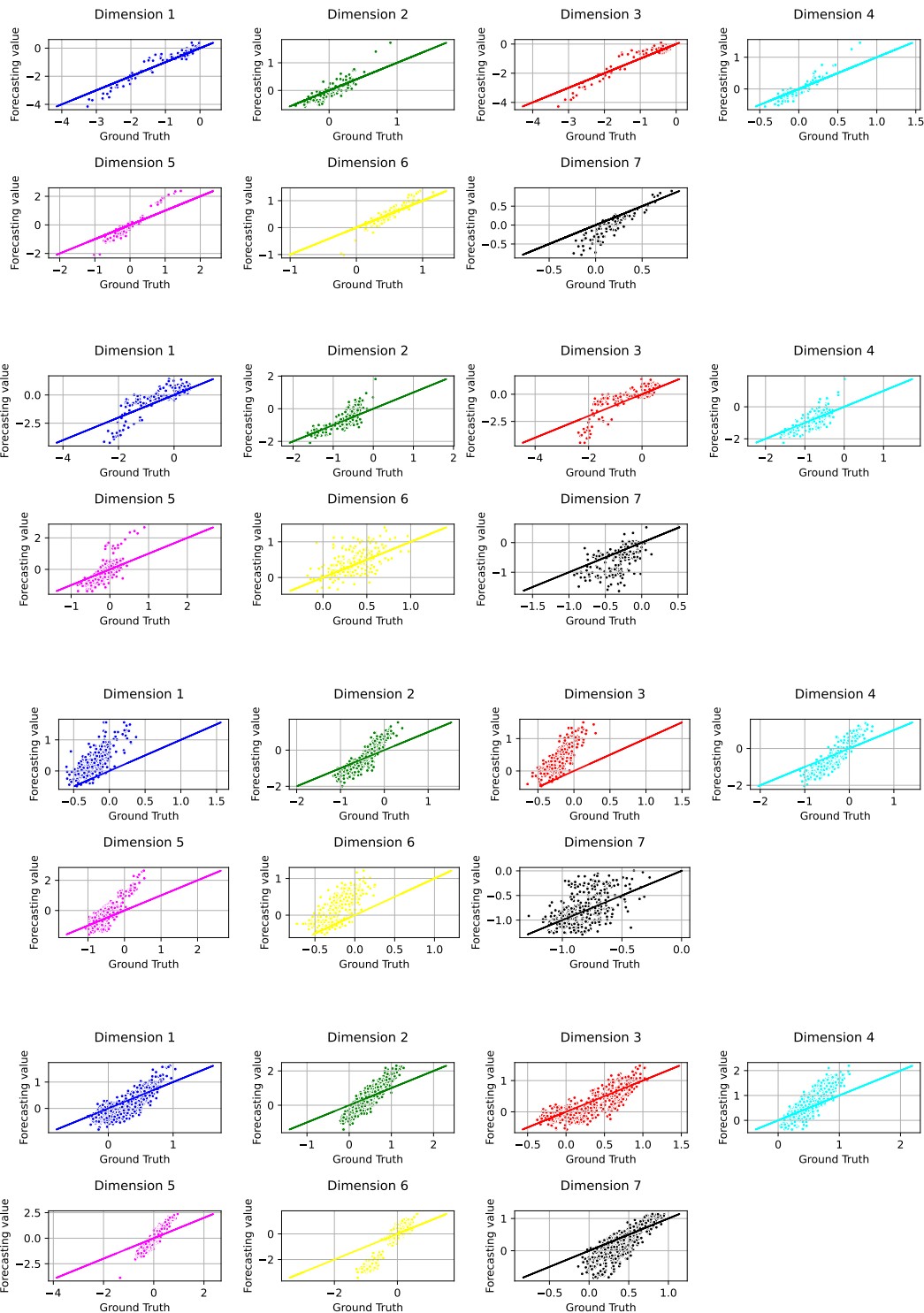

Figure 18: Visualization of ETTh1 Multivariate forecasting results

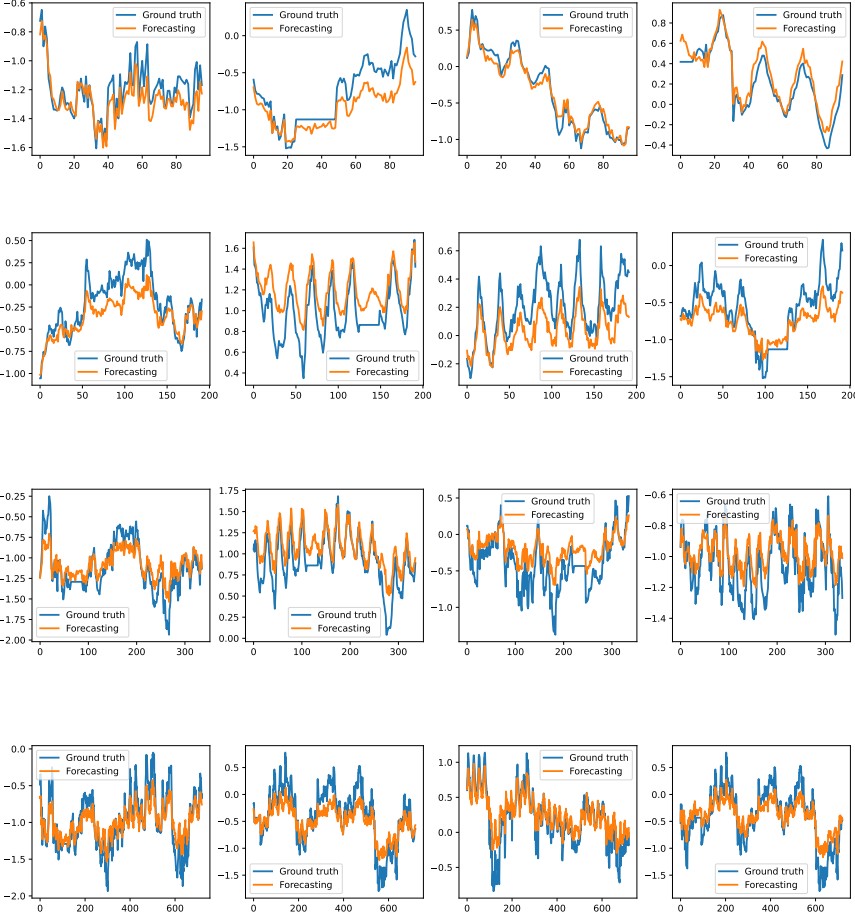

Figure 19: Visualization of ETTh1 Univariate forecasting results

