# OpenReview forum: "TVNet: A Novel Time Series Analysis Method Based on Dynamic Convolution and 3D-Variation"
_ICLR.cc/2025/Conference — ICLR 2025 Poster_

### Official Review · Reviewer_ZoZa · 2024-10-21

**Soundness:** 3
**Presentation:** 3
**Contribution:** 3
**Rating:** 8
**Confidence:** 2

**Summary:**

This paper propose a method called TVNet. TVNet can capture intra-patch,inter-patch and cross-variables features by converting 1D time series data into 3D shape tensor.

**Strengths:**

To sum up, this is a good paper. They proposed TVNet.
TVNet captures intra-patch,inter-patch and cross-variables features by converting 1D time series data into 3D shape tensor.
TVNet implements consistent state-of-the-art performance time series analysis tasks across multiple mainstreams, demonstrating excellent task generalization.

**Weaknesses:**

1. 3D-EMBEDDING
This is your innovation, but you did not tell it  clearly. For example:
"stride S to divide into N patches($X_{emb} ∈ R^{N×P ×C_m}$)"
It is hard for me to know how do you achieve this. Almost the whole process in this part is confused. You should give explanations.

**Questions:**

The same to Weaknesses.

---

> ### Author Response · Authors · 2024-11-19
>
> Dear Reviewer **ZoZa**:
>
> We appreciate you taking the time to review our paper and provide valuable feedback. We think your comments are very valuable for our paper(TVNet)'s improvement and have revised the paper in response to the questions you raised. Please find responses to your questions below:
>
> **Response to W1**:
>
> For 3D-Embedding, firstly, we use DataEmbedding(like Transformer); after that, the time series can be written as $Xemb \in R^{L \times Cm}$ .  Then we use **Conv1D (kernel size=P, stride=S )** to cut patches for Xemb. Then, we will get N patches $x_{i} \in R^{P \times Cm}$ (length equals kernel size P).For each patch, we use the odd-even split to let  $x_{i} \in R^{2 \times P/2 \times Cm}$, then concat all patches get $Xemb \in R^{N \times 2 \times P/2 \times Cm}$. You can also find more details in **Section 3** and **Appendix A**.
>
> Based on your suggestion, we adjusted the narrative order and added some content **(to be marked in blue)** in the third section of the paper. At the same time, we add **Algorithm 1(more details about 3D-Embedding)** and **Algorithm 2(more details about the training process about)** in **Appendix A** of the paper TVNet) so that readers can better understand our proposed model.
>
> Once again, thank you for reviewing the paper. We think we have solved your problem as much as possible in rebuttal, If you have any further questions, please do not hesitate to contact us. **If possible, we would like to thank you for reconsidering to improve your confidence.**

---

> > ### Comment · Reviewer_ZoZa · 2024-11-20
> > **With respect to confidence**
> >
> > I awarded 8 points because, based on the articles I have readed, your submission largely meets the criteria and effectively tells your own story. I do my best to give you the highest score.
> >
> > The reason for only giving 2 points in Confidence is that I have seen many superior submissions get rejected, leaving me uncertain about what exactly constitutes an acceptable article.
> >
> > Additionally, the metrics presented in your paper do not surpass the current state-of-the-art; several models have demonstrated better performance, such as the Periodicity Decoupling Framework for Long-term Series Forecasting. Therefore, your paper is essentially a well-told story. Its acceptance might simply reflect good fortune, having encountered a reviewer willing to give high marks. Conversely, rejection would also be justified, given that many equally or even better papers have faced the same fate.

---

> > > ### Author Response · Authors · 2024-11-20
> > >
> > > Thank you very much for your reply. **As for the effect of Periodicity Decoupling Framework for long-term forecasting, it needs to be clarified that PDF adopts input length =336. However, input length =96[2] is used in this paper. Different input lengths may cause different prediction effects [1]**, and **we think this comparison may not be fair.** We have tried our best to demonstrate the validity of the framework proposed in this paper through experiment and theory.At the same time, **we found directions that were not attempted in the previous research on 3D-reshape and dynamic convolution.** **Thank you again for your contribution to the review work and your suggestions for improving our work.**
> > >
> > > [1] Wang, H., Peng, J., Huang, F., Wang, J., Chen, J., & Xiao, Y. (2023). Micn: Multi-scale local and global context modeling for long-term series forecasting. In The eleventh international conference on learning representations.
> > >
> > > [2] Wu, H., Hu, T., Liu, Y., Zhou, H., Wang, J., & Long, M. (2022). Timesnet: Temporal 2d-variation modeling for general time series analysis. arXiv preprint arXiv:2210.02186.

---

### Official Review · Reviewer_PJvc · 2024-11-01

**Soundness:** 2
**Presentation:** 2
**Contribution:** 2
**Rating:** 6
**Confidence:** 4

**Summary:**

This paper presents TVNet, a dynamic convolutional network for time series analysis, utilizing a 3D-Embedding technique and dynamic convolution to capture intra-patch, inter-patch, and cross-variable dependencies. TVNet is evaluated on five tasks and shows state-of-the-art performance, better efficiency than some models, and good transferability and robustness.

**Strengths:**

1. The proposed 3D-Embedding technique and the consideration of different types of dependencies (intra-patch, inter-patch, cross-variable) are novel and well-motivated.
2. The experimental evaluation is comprehensive, covering multiple tasks and comparing with a wide range of state-of-the-art models. The consistent top performance across these tasks is a significant strength.

**Weaknesses:**

1. Some parts of the method description could be made clearer. For example, the generation function for the time-varying weight could be explained in more detail.
2. The paper does not compare with some of the latest models such as TimeMixer[1] and recent models combined with large language models (e.g., S2 IP-LLM[2], TimeLLM[3]). This omission may lead to an incomplete understanding of the relative performance and novelty of TVNet in the context of the most recent research trends.

[1]Wang S, Wu H, Shi X, et al. Timemixer: Decomposable multiscale mixing for time series forecasting[J]. arXiv preprint arXiv:2405.14616, 2024.
[2]Pan Z, Jiang Y, Garg S, et al. $ S^ 2$ IP-LLM: Semantic Space Informed Prompt Learning with LLM for Time Series Forecasting[C]//Forty-first International Conference on Machine Learning. 2024.
[3]Jin M, Wang S, Ma L, et al. Time-llm: Time series forecasting by reprogramming large language models[J]. arXiv preprint arXiv:2310.01728, 2023.

**Questions:**

1. Can the authors provide more details about the choice of hyperparameters and their impact on performance?
2. Why was there no comparison with some of the latest models such as TimeMixer and models combined with large language models (e.g., S2 IP-LLM, TimeLLM)? What could be the potential implications of such comparisons for the evaluation of TVNet?

---

> ### Author Response · Authors · 2024-11-19
>
> Dear Reviewer **PJvc**:
>
> We appreciate you taking the time to review our paper and provide valuable feedback. We think your comments are very valuable for our paper(TVNet)'s improvement and have revised the paper in response to the questions you raised. Please find responses to your questions below:
>
> **Response to W1:**
>
> Thank you very much for your suggestions on the method description. We have added some content and adjusted the description order of the generation function for time-varying weight. At the same time, two Algorithm descriptions are added, **Algorithm 1** and **Algorithm 2**, in **Appendix A(labeled as blue)**.
>
> **Response to Q1:**
>
> Thank you very much for your suggestions on the Hyperparameter. Hyperparameters have a certain influence on the model effect. In order to ensure the reproducibility of the model, we provided the hyperparameter selection in the first draft, but did not emphasize it. **With your advice, We add Table 11 to Appendix C. Table 11 shows hyperparameters in time series analysis (long-term forecasting, short-term forecasting, imputation, classification, classification, etc.) and anomaly detection).**  At the same time, for each task, we also provide the hyperparameters of the experimental results in Appendix C.1-C.5. (labeled as red). **In order to better study the influence of hyperparameters on the model effect, We conducted relevant experiments in Appendix D (The highlighted part is the newly added content), and in Appendix D.1 we studied the Embedding dim, 3D-block number, kernel size and patch length Impact of the four hyperparameters on the long-term forecasting task.**  It can be seen from the results that although the model prediction results are different for different Settings of the four hyperparameters, they remain stable in general, indicating that the proposed model is robust. **In Appendix D.2, we studied the Training process hyperparameter (learning rate, epochs, and batch size); the standard deviation of forecasting is provided for long-term forecasting and short-term forecasting (Table 15, Table 16).** It can be seen that TVNet is also robust for different training process hyperparameters.

---

> ### Author Response · Authors · 2024-11-19
>
> **Response to W2 and Q2:**
>
> Thank you very much for your suggestion on adding the new baseline. **We have shown the results of the new baseline in Table 12 of Appendix F.4.(highlight) As can be seen from the results, TVNet proposed in this paper still shows good results on most data sets of long-term forecast (baseline data comes from paper).** In view of your proposal that baseline is mainly LLM-based mode. In the initial draft, we also noted the development of LLM-based for Ts, but we had several concerns regarding LLM for Ts as a baseline: **1.LLM training is likely to have the problem of future data leakage.** Existing studies based on time series prediction are all based on training set-test set segmentation, but this cannot be guaranteed in the LLM training process. **2. The LLM prediction effect is not stable.** At present, the LLM prediction especially relies on prompt, which makes it not roboust in the time series analysis problem. **3.LLM occupies much larger memory and parameter quantity than other time series baselines.** In short, thank you for your response on baseline, and we also share a little personal opinion on LLM4TS to explain why LLM-based was not considered in the first draft.
>
>
> | ETTh2 | TVNet | S2IP-LLM | Time-LLM | TimeMixer|
> | :-----| ----: | :----: |:----: |:----: |
> | Avg(MAE) | 0.377 | 0.391 | 0.400 | 0.395|
> | Avg(MSE) | 0.324 | 0.347 | 0.360 | 0.364|

---

> > ### Author Response · Authors · 2024-11-19
> >
> > We noticed that you marked our article as having Ethics Review, and we would like to ask you about the specific reason. **It should be clarified that the GitHub site in Appendix B.1 is the site where the public dataset is available and is not the site where our article code** Once again, thank you for reviewing the paper. We think we have solved your problem as much as possible in rebuttal, If you have any further questions, please do not hesitate to contact us. **If possible, we would like to thank you for reconsidering to improve your score.**

---

> > > ### Author Response · Authors · 2024-11-24
> > >
> > > Dear Reviewer **PJvc:**
> > >
> > > Thank you very much for your valuable comments on our article. We have revised and responded to the article according to your comments. We fully understand that you are busy, but we still hope that you can take time out of your busy schedule to respond to our review comments. We think we have solved your problem as much as possible in rebuttal, If you have any further questions, please do not hesitate to contact us. **If possible, we would like to thank you for reconsidering to improve your score.**

---

> ### Comment · Reviewer_PJvc · 2024-11-25
> **Response for author rebuttal**
>
> 1. Thanks for providing Algorithms 1 and 2, supporting the technical clarity. However, this part is still heavily related to the Appendix.
> 2. Thanks for the provided hyperparameters. Please further enhance this issue.
> 3. The comparisons with TimeLLM, S2IP-LLM, and TimeMixer are limited to a single dataset (ETTh2). More datasets are required to validate the proposed approach.
> The reviewer will prefer to reconsider the final score after addressing the mentioned issues.

---

> ### Author Response · Authors · 2024-11-25
>
> Dear Reviewer **PJvc**:
>
> Thank you very much for your valuable suggestions on our article. In response to your suggestions, we have tried our best to make the following modifications.
>
> **Response to Q1:**
>
> In order to make our article on algorithms (for example, Time-varying weight generation) better understood in the main text, We will add The Time-varying weight generation flow chart(**Figure 4**) and Training of TVNet(**Algorithm 1**) in the main body. (**3.2 Figure 4 and 3.3 Algorithm 1**) At the same time, the algorithm content order is changed to make readers better understand(**3.1 3D-Embedding,3.2 3D-block**).Given the 10-page limit, I believe we have done our best to have TVNet(our proposed) better understand the main body.

---

> ### Author Response · Authors · 2024-11-25
>
> **Response to Q2:**
>
> As for the influence of hyperparameters, due to the limited main content of the text, we mainly focus on the selection of hyperparameters in **Appendix C and D**, the specific content includes the following: (**All results regarding the new addition are highlighted in Appendix C and D**)
>
> **Table 11 in Appendix C** is the main content of this paper on the task of time series (long-term forecasting short-term forecasting, imputation, Classification and anomaly detection) experimental results of hyperparameter selection. Hyperparameters are divided into two categories. One is that the model itself needs to select hyperparameters (for example: kernel size and block numbers), and the other is the parameters needed for model training (for example, epochs). **At the same time, we give the details of different task parameter selection in C.1-C.5.(containing random seed)**
>
> For example:**Short-term forecasting:** TVNet consists of 3 3D-blocks, with the embedding dimension $C_m$ configured to 64. The patch length $P$ is set to 8, and the kernel size $k$ for 2D convolutions is $3 \times 3$.The random seed is set to 2024.(see in **Appendix C.2**,label as red)
>
> For the influence of different hyperparameters on the experimental effect, **the details of content is in Appendix D**. Appendix D has the following structure. In D.1, we mainly discuss the influence of four different model parameters (**embedding dim, block numbers, kernel size and patch length**) on the experimental results through the control variable method.
>
> Specifically, **Table 12** is about the impact of different Embedding dim on different data sets in long term forecasting. It is found that although different Embedding dim make the prediction result fluctuate, the overall fluctuation is small. **Meanwhile, Given the larger Embedding dim will make the model occupy larger memory,  wei think $Cm =32,64,128$ is  suitable for TVNet and it also proves the robustness of the Embedding dim parameter model.**
>
> **Table 13-15 is mainly about different blocks number.** We conducted experiements on three tasks: **long-term forecasting, short-term forecasting and Anomaly detection.** The test found that with the more blocks number, the effects are improved, but we found that in fact, increase the blocks number, the effect is also limited. **This shows that a single 3D-block proposed in this paper also has good model characterization ability and robustness of blocks number .**
>
> Similarly, **Table 16,17,18** is mainly about the influence of different patch length and kernel size on model prediction effect, and the results show that the model proposed in this paper still shows strong robustness within a certain range.
>
> **In Appendix D.2**, we mainly observe the influence of different training parameters on the model effect while fixing the model's own parameters. Due to the large number of model training parameters, A more common practice is to traverse these different parameters(for example epochs 10 to 100) to give the standard deviation of results [1].  **Table 19,20 and 21 show the results of standard deviation in long-term forecasting, short-term forcasting and Anomaly detection**. The robustness of the proposed method can be seen in this paper about hyperparameter.
>
> Given the limited time available and main-body limitation, We have to put some details in the appendix, please understand, **we have done our best to fully evaluate the impact of hyperparameters on our models**, and we are committed to their reproducibility. Thank you again for your comments.
>
> [1] Wang, S., Wu, H., Shi, X., Hu, T., Luo, H., Ma, L., ... & Zhou, J. (2024). Timemixer: Decomposable multiscale mixing for time series forecasting. arXiv preprint arXiv:2405.14616.

---

> ### Author Response · Authors · 2024-11-25
>
> **Response to Q3:**
>
> **What we need to clarify is that we have added the comparison of the new baseline you mentioned in the original text (Appendix F.4, label as highlight), you can refer to the detailed data.** We'll give you more examples.
>
> More details in Appendix F.4.
>
> | ETTh2 | TVNet | S2IP-LLM | Time-LLM | TimeMixer|
> | :-----| ----: | :----: |:----: |:----: |
> | Avg(MAE) | **0.377** | 0.391 | 0.400 | 0.395|
> | Avg(MSE) | **0.324** | 0.347 | 0.360 | 0.364|
>
> | ETTh1 | TVNet | S2IP-LLM | Time-LLM | TimeMixer|
> | :-----| ----: | :----: |:----: |:----: |
> | Avg(MAE) | **0.421** | 0.427 | 0.451 | 0.440|
> | Avg(MSE) | 0.407 | **0.406** | 0.439 | 0.447|
>
> | ETTm1 | TVNet | S2IP-LLM | Time-LLM | TimeMixer|
> | :-----| ----: | :----: |:----: |:----: |
> | Avg(MAE) | **0.379** | 0.379 | 0.395 | 0.395|
> | Avg(MSE) | 0.348 | **0.343** | 0.365 | 0.381|
>
> | ETTm2 | TVNet | S2IP-LLM | Time-LLM | TimeMixer|
> | :-----| ----: | :----: |:----: |:----: |
> | Avg(MAE) | **0.311** | 0.319 | 0.325 | 0.323 |
> | Avg(MSE) | **0.251** | 0.257 | 0.264 | 0.275 |
>
> | Weather | TVNet | S2IP-LLM | Time-LLM | TimeMixer|
> | :-----| ----: | :----: |:----: |:----: |
> | Avg(MAE) | 0.261 | **0.259** | 0.265 | 0.271 |
> | Avg(MSE) | **0.221** | 0.222 | 0.226 | 0.240 |
>
> | Traffic | TVNet | S2IP-LLM | Time-LLM | TimeMixer|
> | :-----| ----: | :----: |:----: |:----: |
> | Avg(MAE) | **0.268** | 0.286 | 0.301 | 0.297 |
> | Avg(MSE) | **0.396** | 0405  | 0.440 | 0.484 |
>
> | Electricity | TVNet | S2IP-LLM | Time-LLM | TimeMixer|
> | :-----| ----: | :----: |:----: |:----: |
> | Avg(MAE) | **0.254** | 0.257 | 0.270 | 0.272 |
> | Avg(MSE) | 0.165 | **0.161**  | 0.168 | 0.182 |
>
> Considering the huge memory consumption and training data amount of LLM-based and the results mentioned above, we believe that TVNet proposed in this paper has advantages in time series tasks.

---

> > ### Author Response · Authors · 2024-11-25
> >
> > Thank you for your valuable comments on our work, **we believe we have done our best to address your concerns and would appreciate it if you could reconsider the score.**

---

> ### Author Response · Authors · 2024-11-28
>
> Thank you again for your valuable comments on our article, regarding your three questions, **We have made modifications in the text (3.1,3.2, Appendix C,D, and Appendix F.4 Table 27 labeled as blue and highlighted in the paper)**, we believe we have done our best to address your concerns and would appreciate it if you could reconsider the score.

---

> > ### Author Response · Authors · 2024-12-01
> >
> > Thank you again for your valuable comments on our article, regarding your three questions, **We have made modifications in the text (3.1,3.2, Appendix C,D, and Appendix F.4 Table 27 labeled as blue and highlighted in the paper)**, we believe we have done our best to address your concerns and would appreciate it if you could reconsider the score.

---

### Official Review · Reviewer_X5hZ · 2024-11-02

**Soundness:** 2
**Presentation:** 1
**Contribution:** 2
**Rating:** 5
**Confidence:** 4

**Summary:**

The manuscript introduces TVNet, a dynamic convolutional network for time series analysis, employing 3D embedding and convolution mechanisms to handle inter-patch, intra-patch, and cross-variable dependencies. TVNet aims to improve CNN performance in time series tasks and demonstrates its utility across multiple tasks, including long-term and short-term forecasting, classification, anomaly detection, and data imputation.

**Strengths:**

- The modular design of TVNet is scalable and easily extensible
- Maintains computational efficiency comparable to CNNs while achieving superior results.
- Comprehensive experiments covering many tasks, baselines and analysis

**Weaknesses:**

- **Lack of Strong Motivation**: The paper's motivation could be strengthened. While it aims to “enhance the representational capacity of CNNs for time series analysis,” it does not adequately explain **why** this enhancement is necessary, given the strong performance of RNNs and Transformers for such tasks. It is well understood that different network architectures are suited for different input modalities, based on the inductive biases they introduce [1]. Therefore, further justification is needed to clarify the specific gaps that CNNs can fill in time series analysis.
- **Insufficient Emphasis on Novelty**: Although the introduction of **3D-embedding** is a core idea, the rationale for choosing the specific three embeddings (inter-patch, intra-patch, and cross-variable) is not adequately explained. A deeper discussion is necessary to understand why these dimensions were prioritized and how they contribute to improved performance. This would help highlight the **novelty** and distinctiveness of the proposed method.

[1] Chiyuan Zhang, Samy Bengio, Moritz Hardt, Michael C. Mozer, Yoram Singer: Identity Crisis: Memorization and Generalization Under Extreme Overparameterization. ICLR 2020

**Questions:**

- **Motivation**:
    - Can the authors elaborate on the specific limitations of RNNs and Transformers in time series analysis that CNNs aim to address? Given the well-known success of Transformers and RNNs, what makes CNNs uniquely suitable for time series tasks?
- **Rationale Behind 3D-Embedding**:
    - Why were the inter-patch, intra-patch, and cross-variable embeddings selected? Are there theoretical or empirical justifications for these choices? Could alternative embeddings have been explored, and how would the performance be affected by such variations?

---

> ### Author Response · Authors · 2024-11-20
>
> Dear **X5hZ**:
>
> We appreciate you taking the time to review our paper and provide valuable feedback. We think your comments are very valuable for our paper(TVNet)'s improvement and have revised the paper in response to the questions you raised. Please find responses to your questions below:
>
> **Response to W1 and Q1:**
> First of all, we think that "given the strong performance of RNNs and Transformers for such tasks." is not very accurate, Common Transformers time series research focuses on **long-term forecasting**. At the same time, **the secondary complexity of the Transformer is also a problem that cannot be ignored.** We believe that this may be related to the Transformer's support for long input sequences.[1] **Research on RNN time series mainly focuses on short-term forecasting.[2]. RNNS do not perform well in the face of long-term forecasting.[3]**
>
> In the paper you mentioned[4], a “vanilla CNN” is used. However, in our work, due to the introduction of patches and 3D embeddings, the actual paths of information flow differ significantly from those in a “vanilla CNN.” Therefore, equating the inductive bias of a “vanilla CNN” to that of TVNet is inaccurate. Specifically, for (multivariate) time series data, 3D embeddings enhance the extraction of relationships between inter-patch, intra-patch, and cross-variable embeddings from the perspective of information flow.
>
> Consider that a typical time series analysis task consists of five sub-tasks (long-term forecasting, short-term forecasting, Classification problem, imputation and Anomaly detection), **we wanted to design a CNN-based architecture to represent the five sub-tasks with good results, which was also verified in our experiments. As indicated in the article you mentioned [4]**, CNN has good generalization ability and memory ability, which inspired us to design TVNet model for general time series analysis tasks by enhancing information flow in specific directions on the basis of CNN.
>
> [1] Wu, H., Xu, J., Wang, J., & Long, M. (2021). Autoformer: Decomposition transformers with auto-correlation for long-term series forecasting. Advances in neural information processing systems, 34, 22419-22430.
>
> [2]  Aseeri, A. O. (2023). Effective RNN-based forecasting methodology design for improving short-term power load forecasts: Application to large-scale power-grid time series. Journal of Computational Science, 68, 101984.
>
> [3] Zhao, J., Huang, F., Lv, J., Duan, Y., Qin, Z., Li, G., & Tian, G. (2020, November). Do RNN and LSTM have long memory?. In International Conference on Machine Learning (pp. 11365-11375). PMLR.
>
> [4] Zhang, C., Bengio, S., Hardt, M., Mozer, M. C., & Singer, Y. (2019). Identity crisis: Memorization and generalization under extreme overparameterization. arXiv preprint arXiv:1902.04698.

---

> ### Author Response · Authors · 2024-11-20
>
> **Response to W2 and Q2:**
>
> The main motivation of TVNet is that we observed the correlation between different patches and patches in each time series in Figure 3. The choice of inter-patch, intra-patch, and cross-variable embeddings stems from our prior knowledge of time series data. Inter-patch represents the periodicity or autocorrelation of time in nature, intra-patch represents the sequentiality and stationarity of time series, and cross-variable represents the multivariate correlations within the time series. These three embeddings collectively capture almost all the essential properties of time series data.
> It should be clarified that 3D-Embedding is a time series representation method, as follows:( For 3D-Embedding, firstly, we use DataEmbedding(like Transformer); after that, the time series can be written as $Xemb \in R^{L \times Cm}$ .  Then we use **Conv1D (kernel size=P, stride=S )** to cut patches for Xemb. Then, we will get N patches $x_{i} \in R^{P \times Cm}$ (length equals kernel size P).For each patch, we use the odd-even split to let  $x_{i} \in R^{2 \times P/2 \times Cm}$, then concat all patches get $Xemb \in R^{N \times 2 \times P/2 \times Cm}$. You can also find more details in **Section 3** and **Appendix A**.)
> We also recognize that time series contains various features (such as Fourier transform and wavelet transform), **but they all rely on complex feature extraction techniques and network design**, while Patch features are direct. [1][2], we adopt CNN-based to achieve both good prediction effect and operational efficiency
> Ultimately, we think you provide us with a good perspective on how CNN can better adapt to time series tasks. We believe that TVNet is sufficient in motivation and experimental results, and it also provides a new idea for designing CNN-based for Time series.
>
> [1] Chen, P., Zhang, Y., Cheng, Y., Shu, Y., Wang, Y., Wen, Q., ... & Guo, C. (2024). Pathformer: Multi-scale transformers with adaptive pathways for time series forecasting. arXiv preprint arXiv:2402.05956.
> [2] Nie, Y., Nguyen, N. H., Sinthong, P., & Kalagnanam, J. (2022). A time series is worth 64 words: Long-term forecasting with transformers. arXiv preprint arXiv:2211.14730.

---

### Official Review · Reviewer_K1xh · 2024-11-03

**Soundness:** 3
**Presentation:** 3
**Contribution:** 2
**Rating:** 6
**Confidence:** 4

**Summary:**

The paper introduces TVNet, a novel time series analysis method that addresses the limitations of CNNs in capturing complex temporal dynamics by converting 1D time series data into 3D tensors and applying dynamic convolution. This approach achieves state-of-the-art performance across various time series tasks while maintaining efficiency.

**Strengths:**

1. The paper introduces TVNet, a novel method for time series analysis that leverages dynamic convolution and 3D transformation, offering a fresh perspective in the field by converting 1D time series data into 3D tensors through a 3D-Embedding technique.
2. The paper demonstrates the capacity of the model across five critical time series analysis tasks, including long-term and short-term forecasting, imputation, classification, and anomaly detection, showcasing the model's generalization capabilities.
3. The paper is well-organized, with a logical flow , making it easy to follow and understand.

**Weaknesses:**

1. Although the paper mentions the efficiency of the model, it does not discuss in detail the consumption of computational resources when running on datasets of different scales. Additionally, different methods have different training strategies, making the training time a poor reflection of the true time consumption of the model. It is recommended to present the inference time and computational load of the model under the same settings.
2. The experimental results show long-term forecasting, but it seems that the prediction lengths = {24, 36, 48, 60} do not reflect the setting of 'long-term.' It is suggested to follow the same settings as the original dataset, such as: {96, 192, 336, 720}.
3. In Table 1, it is a very serious error that the values for weather and traffic are identical.
4. The article's claim of "achieves top-tier performance across five pivotal analytical tasks" is overclaimed because the results on long sequences show that the method did not achieve state-of-the-art (SOTA) on multiple datasets, and the improvement in results that did achieve SOTA is very limited (approximately 0.85% to 2.36% on MSE).
5. Although the paper proposes a powerful model, more explanation may be needed to justify its complexity, especially in the experimental analysis, where more demonstration of its performance in terms of inter-patch, intra-patch, and cross-variable interpretability is needed.

**Questions:**

1. It is recommended to present the inference time and computational load of the model under the same settings.
2. To effectively demonstrate the capability of long-term forecasting, it is recommended to follow the same settings as the original dataset, for instance: {96, 192, 336, 720}.
3. More explanation may be needed to justify its complexity, especially in the experimental analysis, where more demonstration of its performance in terms of inter-patch, intra-patch, and cross-variable interpretability is needed.

---

> ### Author Response · Authors · 2024-11-19
>
> Dear Reviewer **K1xh**
>
> We appreciate you taking the time to review our paper and provide valuable feedback. We think your comments are very valuable for our paper(TVNet)'s improvement and have revised the paper in response to the questions you raised. Please find responses to your questions below:
>
> **Response to W1 and Q1:**
>
> Thank you very much for your comments about model runtime and memory usage. We recognize that different models in different training hyperparameters (such as Batch size) will affect the model running speed and memory usage, in order to ensure a fair comparison, **Figure 2 reports that different baselines memory and epoch speed maintain the same training hyperparameter Settings (see Appendix C Table 11), while at ETTm2(prediction length =192/720,Hint input) length is fixed 96)** . This comparison is also widely used in other time series papers. [1] [2]
>
> At the same time, regarding the impact of different data set sizes on the model, input lenth is a more direct factor in the field of time series. **We have added the comparison of running speed and memory of four advanced models with different input lengths to Appendix E.3 Figure 15. (other settings are the same, ETTm2 prediction =96), it can be found that TVNet memory and running time grow slowly with input length (only slightly higher than Dlinear since Dlinear is the simplest MLP structure)**, which can further support Figure 2. At the same time, we calculate the time and space complexity of different models in Table 4.
>
> [1]. Liu, Y., Li, C., Wang, J., & Long, M. (2024). Koopa: Learning non-stationary time series dynamics with Koopman predictors. Advances in Neural Information Processing Systems, 36.
>
> [2]. Luo, D., & Wang, X. (2024). Moderntcn: A modern pure convolution structure for general time series analysis. In The Twelfth International Conference on Learning Representations.

---

> ### Author Response · Authors · 2024-11-19
>
> **Response to W2 and Q2:**
>
> Thank you very much for your questions about data sets. **For other data sets except ILI, the prediction length is set to {96, 192, 336, 720} in our long-term forecast (Appendix G.1(Table 23)).**  ILI is set as {24, 36, 48, 60} because the sample frequency of this dataset is 1week(Appendix B.1(Table 7)). **In all other studies on time series, the long-term prediction length of this data(ILI) is set as {24, 36, 48, 60}. [1] [2] [3]**
>
> **Response to W3:**
>
> We are very sorry for this error. We have fixed this error, and the full result can be found in **Appendix G.1(Table 23)**
>
> [1] Wu, H., Hu, T., Liu, Y., Zhou, H., Wang, J., & Long, M. (2022). Timesnet: Temporal 2d-variation modeling for general time series analysis. arXiv preprint arXiv:2210.02186.
>
> [2] Wu, H., Xu, J., Wang, J., & Long, M. (2021). Autoformer: Decomposition transformers with auto-correlation for long-term series forecasting. Advances in neural information processing systems, 34, 22419-22430.
>
> [3] Luo, D., & Wang, X. (2024). Moderntcn: A modern pure convolution structure for general time series analysis. In The Twelfth International Conference on Learning Representations.

---

> ### Author Response · Authors · 2024-11-19
>
> **Response to W4:**
>
> Thank you very much for your advice on long-term forecasting. **It should be clarified that all the results reported in long-term forecasting are faithfully derived from the original model.** **In fact, in many models, the setting of hyperparameters is not reported. We found a difference in the effect when we reproduced it**. In our article, we have given the hyperparameter Settings of the results of this paper **(Appendix C)**, which means that all effects are consistent with the hyperparameter Settings as far as possible without reporting the best results. **And even in that case, we're still better than a lot of baseline results. Below, we provide some results(ETTm1, ETTh1, Weather, Traffic) on the same hyperparameter Settings**; you can find that TVNet achieved a greater degree of performance improvement. We also want to emphasize that even with the best results reported using the baseline model, we achieved SOTA results on the vast majority of tasks.
>
>
> | ETTm1 | TVNet | ModernTCN | PatchTST | improve |
> | :-----| ----: | :----: |:----: |:----:
> | Avg(MAE) | 0.379 | 0.404 | 0.396 | 4.29% |
> | Avg(MSE) | 0.348 | 0.364 | 0.359 | 3.06% |
>
> | ETTh1 | TVNet | ModernTCN | PatchTST | improve |
> | :-----| ----: | :----: |:----: |:----:
> | Avg(MAE) | 0.421 | 0.448 | 0.443 | 4.96%|
> | Avg(MSE) | 0.407 | 0.416 | 0.415| 1.92% |
>
> | Weather | TVNet | ModernTCN | PatchTST | improve|
> | :-----| ----: | :----: |:----: |:----:
> | Avg(MAE) | 0.261 | 0.296 | 0.286 |  8.74% |
> | Avg(MSE) | 0.221 | 0.245 | 0.250 |  9.79% |
>
> | Traffic | TVNet | ModernTCN | PatchTST | improve |
> | :-----| ----: | :----: |:----: |:----:
> | Avg(MAE) | 0.268 | 0.283 | 0.275 | 2.54% |
> | Avg(MSE) | 0.396 | 0.414 | 0.407 | 2.70%|
>
>
> **Response to Q3 and W5:**
>
> Thank you very much for your suggestion. Our motive is to better extract inter-patch, intra-patch and cross-variables dependencies. Figure 2 also shows that such dependencies do exist. Further, we use dynamic convolution to extract the dependencies. In response to your suggestions, **we first added the proof in Appendix A that dynamic convolution can better extract time series features than fixed convolution (labeled as blue)**. **In addition, we added ablation experiments to Appendix E.1(Table 17), and compared the effects of 1D,2D and 3D representation methods on different tasks in Appendix E.2.** The inter-patch, intra-patch and cross-variables dependencies are proved.

---

> > ### Author Response · Authors · 2024-11-19
> >
> > Once again, thank you for reviewing the paper. We think we have solved your problem as much as possible in rebuttal, If you have any further questions, please do not hesitate to contact us. **If possible, we would like to thank you for reconsidering to improve your score.**

---

> > ### Comment · Reviewer_K1xh · 2024-11-26
> >
> > Dear Authors,
> >
> > Thank you for your efforts in addressing my concerns. I’m not quite sure what you mean by "the same hyperparameter settings." The optimal parameters for different models should not be the same. Are the parameters you chose beneficial for TVNET?

---

> > > ### Author Response · Authors · 2024-11-28
> > >
> > > Dear Reviewer **K1xh**：
> > >
> > > Thank you very much for your reply. First of all, we need to declare that the hyperparameters of time series models are mainly divided into two categories. The first type is the **model hyperparameters **(such as the number of modules stacked), which are related to the model structure. The second type of hyperparameter is the **Training process hyperparameter**(such as epochs, learning rate, and random seed).
> > >
> > > Regarding the concern about optimal parameter tuning, **we first clarify that we still use the results about different baselines from their papers if existing.*8 But sometimes, studies have shown that rigorous and fair comparison is ensured by following a consistent experimental pipeline.[1,2]. So, we also do some experiments with the same experimental settings as TimesNet[1]. For scenarios where the settings differ or tasks were not implemented, we reproduced the baselines using the benchmark framework from the time series library [3], which is widely adopted in existing studies [1,2] and ensures high consistency. That is the “same setting.” Meanwhile, we also marked the TVNet hyperparameter selection of models in **Appendix C Table 11**.
> > >
> > > We fully understand your concerns in response to the problem, "Are the parameters you chose beneficial for TVNeT?" . But what we want to emphasize is this in the revised paper Appendix C.1-C.5, you can see that we target each broad class task (for example, long-term). The hyperparameter Settings are provided, meaning that the hyperparameters are consistent across the class of tasks for different data sets and different forecast lengths. In addition, **Appendix D provides the influence of different hyperparameters on our model as much as possible to demonstrate the robustness of TVNet for different hyperparameters and the rationality of selecting hyperparameters. And we promise repeatability.**
> > >
> > > [1] Wu, H., Hu, T., Liu, Y., Zhou, H., Wang, J., & Long, M. (2022). Timesnet: Temporal 2d-variation modeling for general time series analysis. arXiv preprint arXiv:2210.02186.
> > >
> > > [2] Wang, S., Wu, H., Shi, X., Hu, T., Luo, H., Ma, L., ... & Zhou, J. (2024). Timemixer: Decomposable multiscale mixing for time series forecasting. arXiv preprint arXiv:2405.14616.
> > >
> > > [3] https://github.com/thuml/Time-Series-Library

---

> ### Author Response · Authors · 2024-11-28
>
> Once again, thank you for reviewing the paper. We think we have solved your problem as much as possible in rebuttal, If you have any further questions, please do not hesitate to contact us. **If possible, we would like to thank you for reconsidering to improve your score.**

---

> > ### Author Response · Authors · 2024-12-01
> >
> > Once again, thank you for reviewing the paper. **We think we have solved your problem as much as possible in rebuttal, If you have any further questions, please do not hesitate to contact us. If possible, we would like to thank you for reconsidering to improve your score.**

---

> > > ### Author Response · Authors · 2024-12-01
> > >
> > > Once again, thank you for reviewing the paper. **We think we have solved your problem as much as possible in rebuttal, If you have any further questions, please do not hesitate to contact us. If possible, we would like to thank you for reconsidering to improve your score.**

---

### Meta-Review · Area_Chair_Mda8 · 2024-12-22

**Metareview:**

Summary: TVNet is a novel time series analysis method that converts 1D time series data into 3D tensors and applies dynamic convolution to capture complex temporal dynamics. It achieves state-of-the-art performance across various time series tasks, including forecasting, classification, anomaly detection, and data imputation, while maintaining efficiency.

Strengths:

TVNet's 3D-Embedding technique and dynamic convolution enable the capture of intra-patch, inter-patch, and cross-variable dependencies, leading to improved performance in multiple time series analysis tasks.

The model demonstrates excellent generalization capabilities, achieving state-of-the-art performance across a range of mainstream time series analysis tasks.

TVNet maintains computational efficiency comparable to CNNs while delivering superior results, making it a valuable addition to the field of time series analysis.

Drawback:

The paper could provide clearer explanations of the 3D-Embedding process and its impact on performance, as some reviewers found this part of the method description confusing and in need of further elaboration.

Accepting this work, as it offers a significant contribution to time series analysis with its innovative approach and consistent performance across multiple tasks.

**Additional Comments On Reviewer Discussion:**

Most of the concerns have been well-addressed.

---

### Decision · Program_Chairs · 2025-01-22

Accept (Poster)